# UNREAL: UNLABELED NODES RETRIEVAL AND LABELING FOR HEAVILY-IMBALANCED NODE CLASSIFICATION

## ABSTRACT

Extremely skewed label distributions are common in real-world node classification tasks. If not dealt with appropriately, it significantly hurts the performance of GNNs on minority classes. Due to the practical importance, there have been a series of recent researches devoted to this challenge. Existing over-sampling techniques smooth the label distribution by generating "fake" minority nodes and synthesize their features and local topology, which largely ignore the rich information of unlabeled nodes on graphs. On the other hand, methods based on loss function modification re-weight different samples or change classification margins. Representative methods in this category need to use label information to estimate the distance of each node to its class center, which is unavailable on unlabeled nodes. In this paper, we propose UNREAL, an iterative over-sampling method. The first key difference is that we only add unlabeled nodes instead of synthetic nodes, which eliminates the challenge of feature and neighborhood generation. To select which unlabeled nodes to add, we propose geometric ranking to rank unlabeled nodes. Geometric ranking exploits unsupervised learning in the node embedding space to effectively calibrates pseudo-label assignment. Finally, we identify the issue of geometric imbalance in the embedding space and provide a simple metric to filter out geometrically imbalanced nodes. Extensive experiments on real-world benchmark datasets are conducted, and the empirical results show that our method significantly outperforms current state-of-the-art methods consistent on different datasets with different imbalance ratios.

## 1 INTRODUCTION

Node classification is ubiquitous in real-world applications, ranging from malicious account detection (Mohammadrezaei et al., 2018) to fake news detection (Monti et al., 2019). Many real-world data comes with an imbalanced class distribution (Mohammadrezaei et al., 2018; Wang et al., 2020b). For instance, the proportion of malicious accounts in social networks is usually very rare. A model trained using an imbalanced dataset is prone to be sub-optimal on under-represented classes. While GNNs have achieved superior performance on node classification, training a fair GNN model for handling highly-imbalanced class distributions remains a challenging task. For the application of malicious account detection, GNN models would easily overfit the samples from the rare class of malicious accounts (Liu et al., 2018; Zhao et al., 2021). The message passing scheme of GNN models make the problem even more complex, as here the samples cannot be treated as i.i.d. samples. Moreover, quantity imbalanced is often coupled with topology imbalance (Chen et al., 2021), and thus it is difficult to extend existing techniques for handling i.i.d. data to relational data.

Given its importance and unique characteristics, a group of recent studies has been devoted to solving the imbalanced node classification problem (Zhao et al., 2021; Shi et al., 2020; Chen et al., 2021; Park et al., 2021; Song et al., 2022). Over-sampling strategies are simple and effective for handling data imbalance. However, it is a non-trivial task to adapt them to graph data since the topological information of newly synthesized nodes is not provided. GraphSMOTE (Zhao et al., 2021) extends the synthetic minority over-sampling technique (SMOTE) to graph data by synthesizing nodes in the embedding space and generating relation information using link prediction. Shi et al. (2020) uses a generative model to generate nodes to smooth the label distribution. GraphENS (Park et al., 2021)

synthesizes the whole ego network of a new sample by combining two different ego networks based on their similarity.

It is empirically observed in (Song et al., 2022) that the performance of existing over-sampling approaches is easily affected by (synthetic) minority nodes with high connectivity to other classes. To alleviate this issue, Song et al. (2022) modify the loss function (thus the classification margin) based on various statistics of the true label distributions of target nodes and classes. Chen et al. (2021) coin this phenomenon by *topology imbalance*, and propose to re-weight the samples according to their distance to the classification boundary, in which the distance is inferred via the structural similarity and label information. As we note, both methods rely on ground truth label information, which is not available for most nodes. Song et al. (2022) first train the model with the original training set and use the model predictions to modify the loss function, while Chen et al. (2021) only uses label information in the training set when computing the topology imbalance metric. However, the training set is highly-skewed in the first place, so information derived from them is less reliable, and the bias could spread to later building blocks, which hurts the overall performance.

In this paper, we propose a novel imbalanced node classification method: **u**nlabeled **n**ode **re**trieval **a**nd **l**abeling (UNREAL). At a high level, UNREAL is an over-sampling based approach, however several distinct features make our method differ from existing over-sampling techniques significantly. First, motivated by the observation that abundant unlabeled nodes are available in a node classification scenario, instead of synthesizing new minority nodes, which brings in both additional noise and large computational burden, we only add "real" nodes to the training set. Adding unlabeled nodes (together with their pseudo-labels) to the training set is a commonly used technique for semi-supervised node classification, which is proved to be highly effective for dealing with label sparseness (Li et al., 2018; Zhou et al., 2019; Sun et al., 2020; Wang et al., 2021c). Self-Training (ST) trains GNN on existing labeled data, then selects samples with high prediction confidence for each class from unlabeled data, and adds them to the training set. However, in imbalanced scenarios, ST cannot achieve satisfactory performance due to the bias in the original training set: using the predictions from a classifier trained on the imbalanced training set may be highly biased and contains a large portion incorrect pseudo-labels. This drawback of ST is empirically verified in our experiments (see Section 3). Thus, we propose a series of techniques to overcome this challenge, and our experimental results show these techniques are highly effective and outperforms baselines by a large margin.

Similar to (Chen et al., 2021), we try to add nodes that are close to class centers to alleviate topology imbalance. To identify such good nodes, we train the model with the training set and use the prediction confidence as the selecting criteria, which we call *confidence ranking*. However, the bias in the original training set results in unreliable predictions (Song et al., 2022), which inevitably hurts the performance. Therefore, we introduce a key building block which utilizes the geometric structure in the embedding space to calibrate the bias in the prediction confidence. This idea is partially inspired by the work of Kang et al. (2019), where they hypothesize and verify empirically that the classifier is the only under-performed component in the model when trained on an imbalanced training set. Thus, after the preliminary training step, we retrieve node embeddings from the output layer (before the classification layer) and use unsupervised clustering methods to rank the closeness of nodes to their class centers, which we call *geometric ranking*. Also, given the two rankings, we apply information retrieval techniques to select the best-unlabeled nodes to add. In practice, this procedure will be applied iteratively for multiple rounds.

We summarize our contribution as follows: 1) As far as we know, UNREAL is the first method to use unlabeled nodes rather than synthetic ones in over-sampling approaches to deal with class imbalanced node classification; 2) for unlabeled node selection, UNREAL is also the first to apply unsupervised methods in the embedding space to get complementary and less biased label predictions; 3) we introduce geometric ranking, which ranks nodes according to the closeness of each node to its class center in the embedding space; 4) given confidence and geometric rankings, information retrieval techniques is used to effectively select high-quality new samples; 5) We identify the Geometric Imbalance (GI) issue in the embedding space, and propose a metric to measure GI and discard imbalanced nodes.

We conduct comprehensive experiments on multiple benchmarks, including citation networks (Sen et al., 2008), an Amazon product co-purchasing network (Sen et al., 2008), and Flickr (Zeng et al., 2019). We also test the performance of UNREAL on several mainstream GNN architectures namely

GCN (Kipf & Welling, 2016), GAT (Veličković et al., 2017), and GraphSAGE (Hamilton et al., 2017). Experimental results demonstrate the superiority of the proposal as UNREAL consistently outperforms existing state-of-the-art approaches by a large margin.

## 2 PRELIMINARIES

### 2.1 NOTATION AND DEFINITIONS

In this work, we mainly focus on the ubiquitous semi-supervised node classification setup. Given an undirected and unweighted graph $\mathcal{G} = (\mathcal{V}, \mathcal{E}, \mathcal{L})$. Here, $\mathcal{V}$ is the node set and $\mathcal{E}$ is the edge set, $\mathcal{L} \subset \mathcal{V}$ denote the set of labeled nodes, so the set of unlabeled nodes is $\mathcal{U} = \mathcal{V} - \mathcal{L}$, and $\mathcal{X} \in \mathbb{R}^{n \times f}$ is the feature matrix (where $n = |\mathcal{V}|$ is the node size and $f$ is the node feature dimension). We use $A \in \{0, 1\}^{n \times n}$ to denote the adjacency matrix and $\mathcal{N}(v)$ the set of 1-hop neighbors for node $v$. The labeled sets for all classes are denoted by $(\mathcal{C}_1, \mathcal{C}_2, \cdots, \mathcal{C}_k)$, where $k$ is the number of different classes. We use imbalance ratio, defined as $\rho := \frac{\max_i(|\mathcal{C}_i|)}{\min_i(|\mathcal{C}_i|)}$, to measure the level of imbalance in a dataset. We summarize the notation in a table in Appendix A.

### 2.2 MESSAGE PASSING NEURAL NETWORK FOR NODE CLASSIFICATION

In this section, we briefly introduce message passing neural networks (MPNNs). A standard MPNNs consists of three components, a message function $m_l$, an information aggregation function $\theta_l$, and a node feature update function $\psi_l$. The feature of each node is updated iteratively. Let $h_v^l$ be the feature of node $v$ in the $l$-th layer, then in the $(l+1)$-th layer the feature is updated as:

$$h_v^{(l+1)} = \psi_l\left(h_v^{(l)}, \theta_l\left(\left\{m_l\left(h_v^{(l)}, h_u^{(l)}, e_{v,u}\right) \mid u \in \mathcal{N}(v)\right\}\right)\right), \tag{1}$$

where $e_{v,u}$ is the edge weight between $v$ and $u$. For the classic GCN model (Kipf & Welling, 2016), $h_v^{(l+1)}$ is computed as: $h_v^{(l+1)} = \Phi^l \sum_{u \in \mathcal{N}(v) \cup \{v\}} \frac{e_{v,u}}{\sqrt{\hat{d}_u \hat{d}_v}} h_u^{(l)}$, where $\Phi^l$ is the parameter matrix of the $l$-th layer and $\hat{d}_v = 1 + \sum_{u \in \mathcal{N}(v)} e_{v,u}$. For node classification, a classification layer is concatenated after the last layer of a GNN.

## 3 PSEUDO-LABEL MISJUDGMENT AUGMENTATION PROBLEM IN IMBALANCED LEARNING

Since self-training adds pseudo-labels to the training set and trains the model iteratively, misjudgements in the early stages will cause the method to fail badly. We extensively investigate this issue of ST in imbalanced learning. Conventional ST-based methods are generally exploited to deal with sparsely label distribution to improve the performance of the model. However, the problem of classifier bias that often occurs in imbalanced scenarios has not received attention if we apply these methods straightly to imbalanced learning. Here, we hypothesize that as the imbalance ratio of the dataset becomes larger, the pseudo-labels obtained by ST-based methods are less credible. At the same time, the prediction confidence of unlabeled nodes is no longer reliable. We conduct comprehensive experimental studies to very this hypothesis. Due to space constraints, we elaborate the experimental details and conclusions in Appendix B.

## 4 UNREAL

In this section, we provide the details of the proposed method. UNREAL iteratively adds unlabeled nodes (with predicted labels) to the training set and retrains the model. We propose three complementary techniques to enhance the unlabeled node selection and labeling. More specifically, in Section 4.1, we describe Dual Pseudo-tag Alignment Mechanism (DPAM) for effective node filtering, the key idea of which is to use unsupervised clustering in the embedding space to obtain a node ranking. In Section 4.2, we show how to combine geometric rank from DPAM and confidence ranking to reorder unlabeled nodes according to their closeness to the class centers (Node-reordering). Finally, in Section 4.3, we identify the issue of geometric node imbalance (GI) and define a new

metric to measure GI, which is then used to filter out nodes with high GI. The overall pipeline of UNREAL is illustrated in Figure 1. Our full algorithm is also provided in the Appendix G (Algorithm 1).

## 4.1 DUAL PSEUDO-TAG ALIGNMENT MECHANISM FOR NODE FILTERING

UNREAL iteratively adds unlabeled nodes to the training set. In each iteration, we first train the GNN model using the current training set. In the early stages the training set remains imbalanced, so the model is likely to generate biased predictions. According to Kang et al. (2019), the embeddings learned by the model are still of high quality, even if it is trained on imbalanced data. Therefore, DPAM exploits the geometric structure in the embedding space and produce a candidate set of new samples.

Let $d$ be the embedding dimension. We use $H^L \in \mathbb{R}^{|\mathcal{L}| \times d}$ and $H^U \in \mathbb{R}^{|\mathcal{U}| \times d}$ to denote the embedding matrix of labeled and unlabeled nodes respectively. Each row of the embedding matrix is the embedding of a node $u$ (denoted as $h_u^L$ and $h_u^U$), which is considered as a point in the $d$-dimension Euclidean space. DPAM applies an unsupervised clustering algorithm, $f_{\text{cluster}}$, which partitions the embeddings of unlabeled nodes into $k'$ clusters and produces $k'$ corresponding cluster centers, where $k'$ is usually larger than $k$, the number of classes.

$$f_{\text{cluster}}(H^U) \implies \{\mathcal{K}_1, c_1, \mathcal{K}_2, c_2, \cdots, \mathcal{K}_{k'}, c_{k'}\} \tag{2}$$

where $\mathcal{K}_i$ is the $i$-th cluster and $c_i$ is the $i$-th cluster center. We use vanilla k-means in our implementation. We also compute the embedding center of each class in the training set

$$c_i^{\text{train}} = M(\{h_u^L \mid y_u \in \mathcal{C}_i\}). \tag{3}$$

Since we use k-means in our experiments, $M(\cdot)$ is simply the mean function. We next assign a pseudo-label $\tilde{y}_m$ to each cluster $\mathcal{K}_m$:

$$\tilde{y}_i = \arg\min_j \text{distance}(c_j^{\text{train}}, c_i). \tag{4}$$

We then combine clusters with the same pseudo-label $m$ as $\tilde{\mathcal{U}}_m$, and $\mathcal{U} = \bigcup_{m=1}^k \tilde{\mathcal{U}}_m$. On the other hand, the GNN model gives each node $u$ in $\mathcal{U}$ a prediction $\hat{y}_u$, and we put unlabeled nodes whose prediction is $m$ into the set $\mathcal{U}_m$, and $\mathcal{U} = \bigcup_{m=1}^k \mathcal{U}_m$.

**Dual Pseudo-tag Alignment Mechanism (DPAM)** The pseudo-labels produced by applying an unsupervised algorithm on the embeddings provide an alternative and potentially less biased prediction, which may compensate the bias introduced by the imbalanced training set. At the same time, the overall accuracy of the unsupervised algorithm is inferior to supervised methods, and thus it is sub-optimal to rely solely on the pseudo-labels from clustering. As a result, DPAM only keeps unlabeled nodes whose two labels aligns, i.e., those belong to the intersection of $\tilde{\mathcal{U}}_m$ and $\mathcal{U}_m$ for each $m \in \{1, 2, \cdots, k\}$; and each node in $\tilde{\mathcal{U}}_m \cap \mathcal{U}_m$ gets a pseudo-label $m$. Due to the space constraints, we defer the empirical studies on why DPAM works to Appendix D.1.

## 4.2 NODE RE-ORDERING

Now DPAM has selected a pool of candidate nodes: $\mathcal{Z} = \bigcup_{i=m}^k (\tilde{\mathcal{U}}_m \cap \mathcal{U}_m)$. In this section, we present Node-Reordering, a method that re-orders nodes in $\mathcal{Z}$ according to the closeness of each node to its class center. Node-Reordering combines the geometric ranking from the unsupervised method and confidence ranking from model prediction.

**Geometric and confidence rankings** Suppose $u \in \tilde{\mathcal{U}}_m \cap \mathcal{U}_m$, and let $h_u^U$ be the embedding of $u$. We measure the distance between node $u$ and its class center by

$$\delta_u = \text{distance}\left(h_u^U, c_m^{\text{train}}\right) \tag{5}$$

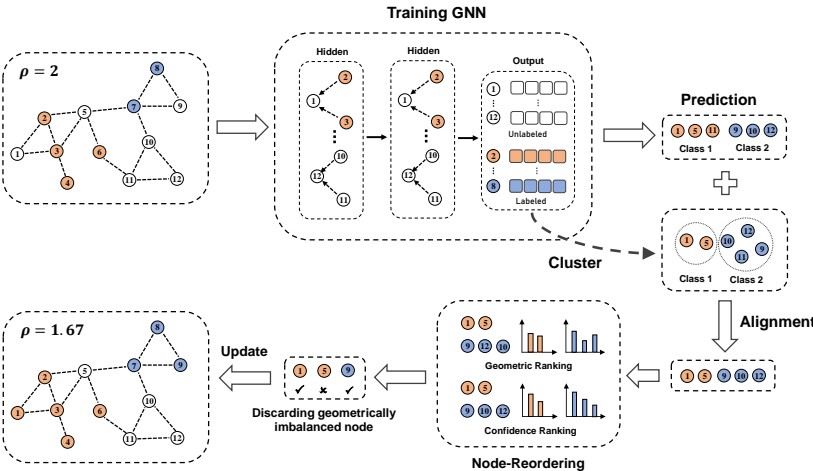

**Figure 1:** Overall pipeline of our UNREAL. Colored nodes denote labeled nodes. Parameters in the GNN Model and the classifier are trained together using the current training set.

where $c_m^{\text{train}}$ is the class center of class $m$ (see equation 3). For each class $m$, we sort nodes in $\tilde{\mathcal{U}}_m \cap \mathcal{U}_m$ in the increasing order of their distance to the class center, so we obtain $k$ sorted lists $\{\mathcal{S}_1, \mathcal{S}_2, \cdots, \mathcal{S}_k\}$, which we call *geometric rankings*.

On the other hand, for each node $u \in \tilde{\mathcal{U}}_m \cap \mathcal{U}_m$, we can get a classification *confidence* for the node from the output of the classifier as follow:

$$predictions = \mathsf{softmax}\,(logits), confidence = \mathsf{max}\,(predictions), \qquad (6)$$

Here, $logits$ is the output of the neural network, usually a $k$ (number of classes) dimensional vector. The pseudo-labels of $u$ from the classifier is the index of class with highest prediction probability and the corresponding probability is its confidence. We sort nodes in $\tilde{\mathcal{U}}_m \cap \mathcal{U}_m$ in the decreasing order of their confidence, and obtain another $k$ sorted lists $\{\mathcal{T}_1, \mathcal{T}_2, \cdots, \mathcal{T}_k\}$, which we call confidence rankings.

**Rank Biased Overlap** In the fields of information retrieval and recommendation systems, a fundamental task is to measure the similarity between two rankings. Rank Biased Overlap (RBO) (Webber et al., 2010) compares two ranked lists, and returns a numeric value between zero and one to quantify their similarity. A RBO value of zero indicates the lists are completely different, and a RBO of one means completely identical.

**Node-Reordering** For each class $m$, we calculate the RBO value between $\mathcal{S}_m$ and $\mathcal{T}_m$ and then use the RBO score as a weight and get the weighted combination of the two rankings. More specifically, we first compute $r_m = \mathrm{RBO}(\mathcal{S}_m, \mathcal{T}_m)$, and then compute

$$\mathcal{N}_m^{New} = \max\{r_m, 1 - r_m\} \cdot \mathcal{S}_m + \min\{r_m, 1 - r_m\} \cdot \mathcal{T}_m, \qquad (7)$$

We then select nodes according to the new ranking based on values in $\mathcal{N}_m^{New}$. Note that we always make the geometric rankings have the dominating influence in this step. Due to the space constraints, ablation analysis on Node-Reordering is presented in Appendix D.2.

## 4.3 GEOMETRIC IMBALANCE

In this section, we consider the issue of geometric imbalance (GI) in the embedding space, and define a simple and effective metric to measure GI.

**Geometric Imbalance**  In highly imbalanced scenarios, minority nodes often suffer from topology imbalance (Song et al., 2022; Chen et al., 2021), which means the node stays near the boundary between the minority class and a majority class. The geometric ranking and DPAM introduced above effectively alleviate this issue. However, when the class centers of a minority class and a majority class are very close in the embedding space, the problem may still exist: we rank nodes only based on their absolute distance to the centers, so the nodes on the boundary of two close classes may have high rankings. We refer to this issue as geometric imbalance in the embedding space. We present a visualization to illustrate geometric imbalance, which is in Figure 9 due to space constraints.

**Discarding geometrically imbalanced nodes (DGI)**  After identifying the GI problem, we define a simple and natural metric to measure the degree of GI. According to equation 5, $\delta_u$ refers to the distance between the embedding of $u$ and the center of the class to which $u$ is assigned (i.e., the closest class center among all classes). Similarly, we define $\beta_u$ as the distance between the embedding of $u$ and the second closest center to $u$. We have $\delta_u \leq \beta_u$ for all $u$, and intuitively, if $\delta_u \approx \beta_u$, then $u$ is likely to have high degree of GI. We thus define the metric for measuring GI as

$$\mathrm{GI}_u = \frac{\beta_u - \delta_u}{\delta_u}. \tag{8}$$

We refer to the metric as GI index. The GI issue is more seriously on node with smaller GI index. So we set a threshold and discard all nodes with GI index below the threshold. We empirically verify the effectiveness of DGI, and the results and analysis are provided in Appendix D.2.

## 4.4 SELECTING NEW NODES ITERATIVELY

As in self-training techniques, we select nodes to join the training set in several rounds, and in each round we retrain the model using the newly formed training set. In highly-imbalanced cases, we only add nodes from the minority classes. In this way, the label distribution of the training set is gradually smoothed, and the imbalance issues of minority nodes are alleviated, benefiting from the addition of high-quality new samples.

## 5 EXPERIMENT

### 5.1 EXPERIMENTAL SETUPS

**Datasets**  We validate the advantages of our method on five benchmark datasets(i.e. Cora, CiteSeer, PubMed, Amazon-Computers, and Flickr) under different imbalance scenarios, in which the step imbalance scheme given in (Zhao et al., 2021; Park et al., 2021; Song et al., 2022) is adopted to construct class imbalanced datasets. More specifically, we choose half of the classes as minority classes and convert randomly picked labeled nodes into unlabeled ones until the imbalance ratio of the training set reaches $\rho$. For Flickr, in the public split, the training set is already imbalanced, and thus we directly use this split and do not make any changes. For the three citation networks (Cora, CiteSeer, Pubmed), we use the standard splits from Yang et al. (2016) as our initial splits when the imbalance ratio is 10, 20. To create a larger imbalance ratio, 20 labeled nodes per class is not enough, and we use a random split as the initial split for creating an imbalance ratio of 50 and 100. The detailed experimental settings such as evaluation protocol and implementation details of our algorithm are described in Appendix F.

**Baselines**  We compare UNREAL with several classic techniques (cross-entropy loss with re-weighting (Japkowicz & Stephen, 2002), PC Softmax (Hong et al., 2021) and Balanced Softmax (Ren et al., 2020)) and state-of-the-art methods for imbalanced node classification, including GraphSMOTE (Zhao et al., 2021), GraphENS (Park et al., 2021), ReNode (Chen et al., 2021), and TAM (Song et al., 2022). Among them GraphSMOTE and GraphENS are representative over-sampling method for node classification, ReNode and TAM are loss function modification approaches. For TAM, we test its performances when combined with different base models, including GraphENS, ReNode, and Balanced softmax, following Song et al. (2022). The implementation details of baselines are described in Appendix F.5.

**Table 1:** Experimental results of our method UNREAL and other baselines on four class-imbalanced node classification benchmark datasets with $\rho = 10$. We report averaged balanced accuracy (bAcc.,%) and F1-score (%) with the standard errors over 5 repetitions on three representative GNN architectures.

| | Dataset | Cora | | CiteSeer | | PubMed | | Amazon-Computers | |
|---|---|---|---|---|---|---|---|---|---|
| | Imbalance Ratio ($\rho = 10$) | bAcc. | F1 | bAcc. | F1 | bAcc. | F1 | bAcc. | F1 |
| GCN | Vanilla | $62.82 \pm 1.43$ | $61.67 \pm 1.59$ | $38.72 \pm 1.88$ | $28.74 \pm 3.21$ | $65.64 \pm 1.72$ | $56.97 \pm 3.17$ | $80.01 \pm 0.71$ | $71.56 \pm 0.81$ |
| | Re-Weight | $65.36 \pm 1.15$ | $64.97 \pm 1.39$ | $44.69 \pm 1.78$ | $38.61 \pm 2.37$ | $69.06 \pm 1.84$ | $64.08 \pm 2.97$ | $80.93 \pm 1.30$ | $73.99 \pm 2.20$ |
| | PC Softmax | $68.04 \pm 0.82$ | $67.84 \pm 0.81$ | $50.18 \pm 0.55$ | $46.14 \pm 0.14$ | $72.46 \pm 0.80$ | $70.27 \pm 0.94$ | $81.54 \pm 0.76$ | $73.30 \pm 0.51$ |
| | BalancedSoftmax | $69.98 \pm 0.58$ | $68.68 \pm 0.55$ | $55.52 \pm 0.97$ | $53.74 \pm 1.42$ | $73.73 \pm 0.89$ | $71.53 \pm 1.06$ | $81.46 \pm 0.74$ | $\underline{74.31 \pm 0.51}$ |
| | GraphSMOTE | $66.39 \pm 0.56$ | $65.49 \pm 0.93$ | $44.87 \pm 1.12$ | $39.20 \pm 1.62$ | $67.91 \pm 0.64$ | $62.68 \pm 1.92$ | $79.48 \pm 0.47$ | $72.63 \pm 0.76$ |
| | Renode | $67.03 \pm 1.41$ | $67.16 \pm 1.67$ | $43.47 \pm 2.22$ | $37.52 \pm 3.10$ | $71.40 \pm 1.42$ | $67.27 \pm 2.96$ | $81.89 \pm 0.77$ | $73.13 \pm 1.60$ |
| | GraphENS | $70.89 \pm 0.71$ | $70.90 \pm 0.81$ | $56.57 \pm 0.98$ | $55.29 \pm 1.33$ | $72.13 \pm 1.04$ | $70.72 \pm 1.07$ | $\underline{82.40 \pm 0.39}$ | $74.26 \pm 1.05$ |
| | BalancedSoftmax+TAM | $69.94 \pm 0.45$ | $69.54 \pm 0.47$ | $56.73 \pm 0.71$ | $56.15 \pm 0.78$ | $74.62 \pm 0.97$ | $72.25 \pm 1.30$ | $82.36 \pm 0.67$ | $72.94 \pm 1.43$ |
| | Renode+TAM | $68.26 \pm 1.84$ | $68.11 \pm 1.97$ | $46.20 \pm 1.17$ | $39.96 \pm 2.76$ | $72.63 \pm 2.03$ | $68.28 \pm 3.30$ | $80.36 \pm 1.19$ | $72.51 \pm 0.68$ |
| | GraphENS+TAM | $\underline{71.69 \pm 0.36}$ | $\underline{72.14 \pm 0.51}$ | $\underline{58.01 \pm 0.68}$ | $\underline{56.32 \pm 1.03}$ | $\underline{74.14 \pm 1.42}$ | $\underline{72.42 \pm 1.39}$ | $81.02 \pm 0.99$ | $70.78 \pm 1.72$ |
| | **UNREAL** | $\mathbf{78.33 \pm 1.04}$ | $\mathbf{76.44 \pm 1.06}$ | $\mathbf{65.63 \pm 1.38}$ | $\mathbf{64.94 \pm 1.38}$ | $\mathbf{75.35 \pm 1.41}$ | $\mathbf{73.65 \pm 1.43}$ | $\mathbf{85.08 \pm 0.38}$ | $\mathbf{75.27 \pm 0.23}$ |
| | $\Delta$ | **+6.64** | **+4.30** | **+7.62** | **+8.62** | **+1.21** | **+1.23** | **+2.68** | **+0.96** |
| GAT | Vanilla | $62.33 \pm 1.56$ | $61.82 \pm 1.84$ | $38.84 \pm 1.13$ | $31.25 \pm 1.64$ | $64.60 \pm 1.64$ | $55.24 \pm 2.80$ | $79.04 \pm 1.60$ | $70.00 \pm 2.50$ |
| | Re-Weight | $66.87 \pm 0.97$ | $66.62 \pm 1.13$ | $45.47 \pm 2.35$ | $40.60 \pm 2.98$ | $68.10 \pm 2.85$ | $63.76 \pm 3.54$ | $80.38 \pm 0.66$ | $69.99 \pm 0.76$ |
| | PC Softmax | $66.69 \pm 0.79$ | $66.04 \pm 1.10$ | $50.78 \pm 1.66$ | $48.56 \pm 2.08$ | $72.88 \pm 0.83$ | $71.09 \pm 0.89$ | $79.43 \pm 0.94$ | $71.33 \pm 0.86$ |
| | BalancedSoftmax | $67.89 \pm 0.36$ | $67.96 \pm 0.41$ | $54.78 \pm 1.25$ | $51.83 \pm 2.11$ | $72.30 \pm 1.20$ | $69.30 \pm 1.79$ | $\underline{82.02 \pm 1.19}$ | $72.94 \pm 1.54$ |
| | GraphSMOTE | $66.71 \pm 0.32$ | $65.01 \pm 1.21$ | $45.68 \pm 0.93$ | $38.96 \pm 0.97$ | $67.43 \pm 1.23$ | $61.97 \pm 2.54$ | $79.38 \pm 1.97$ | $69.76 \pm 2.31$ |
| | Renode | $67.33 \pm 0.79$ | $68.08 \pm 1.16$ | $44.48 \pm 2.06$ | $37.93 \pm 2.87$ | $69.93 \pm 2.10$ | $65.27 \pm 2.90$ | $76.01 \pm 1.08$ | $66.72 \pm 1.42$ |
| | GraphENS | $\underline{70.45 \pm 1.25}$ | $69.87 \pm 1.32$ | $51.45 \pm 1.28$ | $47.98 \pm 2.08$ | $73.15 \pm 1.24$ | $71.90 \pm 1.03$ | $81.23 \pm 0.74$ | $71.23 \pm 0.42$ |
| | BalancedSoftmax+TAM | $69.16 \pm 0.27$ | $69.39 \pm 0.37$ | $56.30 \pm 1.25$ | $53.87 \pm 1.14$ | $73.50 \pm 1.24$ | $71.36 \pm 1.99$ | $75.54 \pm 2.09$ | $66.69 \pm 1.44$ |
| | Renode+TAM | $67.50 \pm 0.67$ | $68.06 \pm 0.96$ | $45.12 \pm 1.41$ | $39.29 \pm 1.79$ | $70.66 \pm 2.13$ | $66.94 \pm 3.54$ | $74.30 \pm 1.13$ | $66.13 \pm 1.75$ |
| | GraphENS+TAM | $70.15 \pm 0.18$ | $\underline{70.00 \pm 0.40}$ | $\underline{56.15 \pm 1.13}$ | $\underline{54.31 \pm 1.68}$ | $\underline{73.45 \pm 1.07}$ | $\underline{72.10 \pm 0.36}$ | $81.07 \pm 1.03$ | $71.27 \pm 1.98$ |
| | **UNREAL** | $\mathbf{78.91 \pm 0.59}$ | $\mathbf{75.99 \pm 0.47}$ | $\mathbf{64.10 \pm 1.49}$ | $\mathbf{63.44 \pm 1.47}$ | $\mathbf{74.68 \pm 1.43}$ | $\mathbf{72.78 \pm 0.89}$ | $\mathbf{85.62 \pm 0.44}$ | $\mathbf{75.34 \pm 0.99}$ |
| | $\Delta$ | **+8.46** | **+5.99** | **+7.80** | **+9.13** | **+1.23** | **+0.68** | **+3.60** | **+2.40** |
| SAGE | Vanilla | $61.82 \pm 0.97$ | $60.97 \pm 1.07$ | $43.18 \pm 0.52$ | $36.66 \pm 1.25$ | $68.68 \pm 1.51$ | $64.16 \pm 2.38$ | $72.36 \pm 2.39$ | $64.32 \pm 2.21$ |
| | Re-Weight | $63.94 \pm 1.07$ | $63.82 \pm 1.30$ | $46.17 \pm 1.32$ | $40.13 \pm 1.68$ | $69.89 \pm 1.60$ | $65.71 \pm 2.31$ | $76.08 \pm 1.14$ | $65.76 \pm 1.40$ |
| | PC Softmax | $65.79 \pm 0.70$ | $66.04 \pm 0.92$ | $50.66 \pm 0.99$ | $47.48 \pm 1.66$ | $71.49 \pm 0.94$ | $70.23 \pm 0.67$ | $74.63 \pm 3.01$ | $66.44 \pm 4.04$ |
| | BalancedSoftmax | $67.43 \pm 0.61$ | $67.66 \pm 0.69$ | $51.74 \pm 2.32$ | $49.01 \pm 3.16$ | $71.36 \pm 1.37$ | $69.66 \pm 1.81$ | $73.67 \pm 1.11$ | $65.23 \pm 2.44$ |
| | GraphSMOTE | $61.65 \pm 0.34$ | $60.97 \pm 0.98$ | $42.73 \pm 2.87$ | $35.18 \pm 1.75$ | $66.63 \pm 0.65$ | $61.97 \pm 2.54$ | $71.85 \pm 0.98$ | $68.92 \pm 0.73$ |
| | Renode | $66.84 \pm 1.78$ | $67.08 \pm 1.75$ | $48.65 \pm 1.37$ | $44.25 \pm 2.20$ | $71.37 \pm 1.33$ | $67.78 \pm 1.38$ | $77.37 \pm 0.74$ | $68.42 \pm 1.81$ |
| | GraphENS | $68.74 \pm 0.46$ | $68.34 \pm 0.33$ | $53.51 \pm 0.78$ | $51.42 \pm 1.19$ | $70.97 \pm 0.78$ | $70.00 \pm 1.22$ | $\underline{82.57 \pm 0.50}$ | $71.95 \pm 0.51$ |
| | BalancedSoftmax+TAM | $69.03 \pm 0.92$ | $69.03 \pm 0.97$ | $51.93 \pm 2.19$ | $48.67 \pm 3.25$ | $72.28 \pm 1.47$ | $71.02 \pm 1.31$ | $77.00 \pm 2.93$ | $70.85 \pm 2.28$ |
| | Renode+TAM | $67.28 \pm 1.11$ | $67.15 \pm 1.11$ | $48.39 \pm 1.76$ | $43.56 \pm 2.31$ | $71.25 \pm 1.07$ | $68.69 \pm 0.98$ | $74.87 \pm 2.25$ | $66.87 \pm 2.52$ |
| | GraphENS+TAM | $\underline{70.45 \pm 0.74}$ | $\underline{70.40 \pm 0.75}$ | $\underline{54.69 \pm 1.12}$ | $\underline{53.56 \pm 1.86}$ | $\underline{73.61 \pm 1.35}$ | $\underline{72.50 \pm 1.58}$ | $82.17 \pm 0.93$ | $\mathbf{72.46 \pm 1.00}$ |
| | **UNREAL** | $\mathbf{75.99 \pm 0.98}$ | $\mathbf{73.63 \pm 1.23}$ | $\mathbf{66.45 \pm 0.39}$ | $\mathbf{65.83 \pm 0.30}$ | $\mathbf{74.78 \pm 1.30}$ | $\mathbf{72.80 \pm 0.54}$ | $\mathbf{83.21 \pm 1.50}$ | $\underline{70.81 \pm 1.70}$ |
| | $\Delta$ | **+5.44** | **+3.23** | **+11.76** | **+12.77** | **+1.07** | **+0.30** | **+0.64** | **-1.65** |

## 5.2 MAIN RESULTS

**Experimental results under different imbalance ratios** In Table 1 and Table 2, we report the averaged balanced accuracy (bAcc.) and F1 score with standard errors for the baselines and UNREAL on four class-imbalanced node classification benchmark datasets under different imbalance ratios ($\rho = 10, 20$). The results clearly demonstrate the advantage of UNREAL. Our method consistently outperforms existing state-of-the-art approaches across four datasets, three base models and two imbalance ratios (except for GraphSAGE on Amazon-Computers with imbalance ratio 10). In many cases the margin is significant. To evaluate the performance on very skewed label distribution, we also test in more imbalanced settings ($\rho = 50, 100$), and similarly, our method outperforms all other methods consistently and often by a notable margin. We remark that since GraphSMOTE (Zhao et al., 2021) synthesizes nodes within the minority class, it is not applicable when there is only one node in some classes, which is the case when $\rho = 20, 50, 100$ in our setup. The results are presented in Appendix C.1.

**Experimental results for naturally imbalanced datasets** We also validate our model on a naturally imbalanced dataset, Flickr. The split of training set, validation set, and testing set follows (Zeng et al., 2019), which has an imbalance ratio roughly $\rho \approx 10.8$. We found that existing over-sampling methods use too much memory due to synthetic nodes generation, and cannot handle Flickr on a 3090 GPU with 24GB memory. This include GraphENS (Park et al., 2021), GraphSMOTE (Zhao et al., 2021) and ReNode (Chen et al., 2021). Due to the space constraints, we provide the experimental results in 8.

## 5.3 ABLATION ANALYSIS

In this section, we conduct ablation studies to analyze the benefit of each component in our method. From the results in Section 3, the necessity of unsupervised learning in the embedding space has been verified. Thus, in this section, DPAM is applied in all comparing methods. Here, we test the

**Table 2:** Experimental results of our method UNREAL and other baselines on four class-imbalanced node classification benchmark datasets with $\rho = 20$. We report averaged balanced accuracy (bAcc.,%) and F1-score (%) with the standard errors over 5 repetitions on three representative GNN architectures.

| Dataset | Cora | | CiteSeer | | PubMed | | Amazon-Computers | |
|---|---|---|---|---|---|---|---|---|
| Imbalance Ratio ($\rho = 20$) | bAcc. | F1 | bAcc. | F1 | bAcc. | F1 | bAcc. | F1 |
| **GCN** Vanilla | 53.20 ± 0.88 | 47.81 ± 1.23 | 35.32 ± 0.15 | 21.81 ± 0.12 | 61.13 ± 0.35 | 46.85 ± 0.76 | 72.34 ± 2.92 | 65.42 ± 3.00 |
| Re-Weight | 57.51 ± 1.05 | 54.63 ± 1.08 | 36.99 ± 1.79 | 27.33 ± 2.32 | 66.52 ± 2.42 | 58.22 ± 3.65 | 72.45 ± 2.06 | 65.85 ± 1.46 |
| PC Softmax | 61.74 ± 1.50 | 60.55 ± 1.97 | 42.53 ± 1.53 | 36.54 ± 1.13 | 68.26 ± 1.99 | 66.54 ± 1.87 | 73.84 ± 2.64 | 66.32 ± 2.97 |
| BalancedSoftmax | 64.06 ± 0.74 | 62.88 ± 0.86 | 47.29 ± 1.29 | 44.08 ± 1.71 | 69.71 ± 1.74 | 68.31 ± 1.71 | 76.92 ± 2.01 | 69.86 ± 1.99 |
| Renode | 59.40 ± 1.00 | 56.88 ± 1.52 | 38.25 ± 1.60 | 27.61 ± 2.25 | 67.45 ± 3.34 | 60.40 ± 5.74 | 74.15 ± 1.72 | 67.27 ± 0.92 |
| GraphENS | 67.30 ± 1.45 | 66.82 ± 1.40 | 46.39 ± 3.48 | 42.38 ± 4.14 | 71.37 ± 1.77 | 69.37 ± 1.69 | 75.41 ± 1.75 | 69.32 ± 1.58 |
| BalancedSoftmax+TAM | 64.75 ± 0.54 | 63.46 ± 0.72 | 48.52 ± 1.62 | 46.38 ± 1.79 | 69.95 ± 2.09 | 68.90 ± 1.86 | 77.09 ± 2.02 | 69.86 ± 1.76 |
| Renode+TAM | 59.88 ± 1.16 | 58.05 ± 1.66 | 41.11 ± 2.45 | 31.58 ± 2.62 | 68.53 ± 3.53 | 64.82 ± 4.32 | 73.46 ± 1.77 | 67.50 ± 1.18 |
| GraphENS+TAM | 66.94 ± 1.38 | 66.67 ± 1.42 | 48.80 ± 2.98 | 45.06 ± 4.16 | 71.92 ± 1.58 | 69.35 ± 1.88 | 75.78 ± 1.57 | 68.58 ± 1.78 |
| **UNREAL** | **77.02 ± 0.75** | **74.15 ± 0.87** | **55.81 ± 6.11** | **55.19 ± 6.23** | **73.06 ± 1.87** | **70.77 ± 1.96** | **85.69 ± 0.11** | **74.81 ± 0.68** |
| **Δ** | **+9.72** | **+7.33** | **+7.01** | **+8.81** | **+1.14** | **+1.40** | **+8.60** | **+4.95** |
| **GAT** Vanilla | 51.51 ± 0.53 | 46.59 ± 0.61 | 34.74 ± 0.16 | 22.00 ± 0.15 | 60.22 ± 0.47 | 46.03 ± 0.70 | 68.09 ± 2.96 | 60.08 ± 2.76 |
| Re-Weight | 58.68 ± 3.44 | 55.98 ± 3.97 | 36.78 ± 0.94 | 26.63 ± 1.61 | 63.47 ± 1.73 | 54.63 ± 3.25 | 71.44 ± 2.42 | 62.86 ± 1.94 |
| PC Softmax | 59.62 ± 1.41 | 58.77 ± 1.95 | 43.38 ± 2.01 | 37.76 ± 2.12 | 70.81 ± 1.41 | 70.25 ± 1.30 | 71.16 ± 1.15 | 62.26 ± 0.87 |
| BalancedSoftmax | 62.05 ± 1.62 | 61.14 ± 1.71 | 47.89 ± 1.25 | 44.84 ± 1.35 | 69.91 ± 1.68 | 67.43 ± 1.73 | 72.91 ± 1.93 | 62.79 ± 0.98 |
| Renode | 59.52 ± 2.28 | 57.16 ± 2.47 | 37.21 ± 2.01 | 27.09 ± 3.17 | 64.56 ± 1.65 | 55.87 ± 2.83 | 69.34 ± 2.35 | 59.02 ± 1.67 |
| GraphENS | 64.52 ± 2.05 | 62.52 ± 1.84 | 43.74 ± 3.81 | 37.47 ± 4.21 | 69.00 ± 2.67 | 65.54 ± 3.54 | 71.78 ± 2.30 | 61.83 ± 1.75 |
| BalancedSoftmax+TAM | 63.30 ± 0.99 | 62.81 ± 1.18 | 49.34 ± 1.29 | 46.92 ± 1.39 | 71.17 ± 2.09 | 68.85 ± 2.90 | 65.59 ± 2.86 | 58.12 ± 1.22 |
| Renode+TAM | 61.32 ± 2.18 | 59.19 ± 2.64 | 39.85 ± 2.20 | 30.63 ± 2.63 | 66.28 ± 3.24 | 58.99 ± 3.04 | 65.81 ± 2.57 | 56.73 ± 1.62 |
| GraphENS+TAM | 65.78 ± 1.62 | 63.80 ± 1.79 | 44.81 ± 2.66 | 39.47 ± 3.54 | 70.33 ± 2.33 | 67.00 ± 3.25 | 73.55 ± 2.04 | 64.03 ± 1.32 |
| **UNREAL** | **79.10 ± 0.71** | **76.21 ± 0.58** | **55.11 ± 5.00** | **53.67 ± 5.51** | **72.54 ± 1.52** | **70.54 ± 1.91** | **83.19 ± 0.66** | **74.39 ± 0.89** |
| **Δ** | **+13.22** | **+12.41** | **+6.75** | **+8.81** | **+1.37** | **+1.69** | **+9.64** | **+10.36** |
| **SAGE** Vanilla | 54.61 ± 1.21 | 50.95 ± 1.90 | 37.36 ± 1.03 | 27.49 ± 1.41 | 62.04 ± 1.34 | 54.18 ± 1.73 | 62.70 ± 2.87 | 55.39 ± 2.69 |
| Re-Weight | 57.37 ± 0.61 | 55.30 ± 0.72 | 37.69 ± 1.20 | 27.92 ± 2.01 | 65.01 ± 2.69 | 58.34 ± 2.19 | 68.31 ± 2.06 | 60.45 ± 2.40 |
| PC Softmax | 59.25 ± 0.74 | 58.55 ± 0.81 | 42.77 ± 1.82 | 40.08 ± 1.82 | 70.55 ± 1.19 | 67.60 ± 1.59 | 70.57 ± 2.86 | 62.73 ± 2.69 |
| BalancedSoftmax | 61.93 ± 1.26 | 60.89 ± 1.36 | 43.64 ± 1.33 | 38.31 ± 1.13 | 69.89 ± 1.40 | 68.12 ± 0.78 | 68.45 ± 2.92 | 62.12 ± 3.10 |
| Renode | 58.48 ± 0.97 | 55.39 ± 0.94 | 40.65 ± 2.36 | 31.78 ± 3.24 | 66.50 ± 2.63 | 58.72 ± 4.16 | 68.36 ± 1.54 | 61.60 ± 2.00 |
| GraphENS | 63.54 ± 0.91 | 62.20 ± 0.87 | 44.89 ± 2.51 | 40.48 ± 2.94 | 71.37 ± 1.77 | 69.37 ± 1.69 | 75.47 ± 2.20 | 67.49 ± 1.65 |
| BalancedSoftmax+TAM | 64.16 ± 0.94 | 63.63 ± 1.10 | 44.32 ± 2.36 | 40.17 ± 2.06 | 70.06 ± 1.46 | 69.54 ± 1.35 | 66.10 ± 2.37 | 59.22 ± 2.48 |
| Renode+TAM | 59.77 ± 2.20 | 57.98 ± 2.79 | 42.50 ± 0.93 | 35.11 ± 1.84 | 67.31 ± 2.73 | 60.63 ± 3.49 | 66.42 ± 2.32 | 58.62 ± 1.95 |
| GraphENS+TAM | 63.39 ± 1.36 | 61.66 ± 1.53 | 45.92 ± 1.96 | 41.97 ± 2.50 | 69.62 ± 2.57 | 66.85 ± 3.00 | 75.75 ± 2.30 | 68.86 ± 1.29 |
| **UNREAL** | **73.10 ± 1.60** | **69.92 ± 1.43** | **58.35 ± 4.58** | **57.51 ± 4.92** | **73.67 ± 0.58** | **71.15 ± 0.67** | **78.88 ± 2.16** | **69.00 ± 1.42** |
| **Δ** | **+8.94** | **+5.69** | **+12.43** | **+15.54** | **+2.30** | **+1.61** | **+3.13** | **+0.14** |

**Table 3:** Ablation analysis on different components

| Modules | Confidence ranking | Geometric ranking | Node-reordering | DGI | F1 |
|---|---|---|---|---|---|
| Cora+GCN ($\rho = 10$) | ✔ | ✗ | ✗ | ✗ | 73.93 ± 0.95 |
| | ✔ | ✗ | ✗ | ✔ | 72.74 ± 0.63 |
| | ✗ | ✔ | ✗ | ✗ | 75.85 ± 0.82 |
| | ✗ | ✔ | ✗ | ✔ | 75.34 ± 0.63 |
| | ✗ | ✗ | ✔ | ✗ | 75.00 ± 0.97 |
| | ✗ | ✗ | ✔ | ✔ | **76.44 ± 1.06** |
| CiteSeer+SAGE ($\rho = 20$) | ✔ | ✗ | ✗ | ✗ | 46.09 ± 4.08 |
| | ✔ | ✗ | ✗ | ✔ | 47.76 ± 1.06 |
| | ✗ | ✔ | ✗ | ✗ | 50.32 ± 3.75 |
| | ✗ | ✔ | ✗ | ✔ | 53.32 ± 3.75 |
| | ✗ | ✗ | ✔ | ✗ | **58.71 ± 3.21** |
| | ✗ | ✗ | ✔ | ✔ | 57.51 ± 4.92 |
| PubMed+GAT ($\rho = 50$) | ✔ | ✗ | ✗ | ✗ | 76.34 ± 0.39 |
| | ✔ | ✗ | ✗ | ✔ | 75.42 ± 0.39 |
| | ✗ | ✔ | ✗ | ✗ | 77.32 ± 0.21 |
| | ✗ | ✔ | ✗ | ✔ | 76.89 ± 1.43 |
| | ✗ | ✗ | ✔ | ✗ | 76.12 ± 2.63 |
| | ✗ | ✗ | ✔ | ✔ | **77.38 ± 0.39** |
| Computers+GAT ($\rho = 100$) | ✔ | ✗ | ✗ | ✗ | 70.86 ± 1.73 |
| | ✔ | ✗ | ✗ | ✔ | 68.86 ± 1.42 |
| | ✗ | ✔ | ✗ | ✗ | 72.32 ± 2.43 |
| | ✗ | ✔ | ✗ | ✔ | 73.65 ± 0.67 |
| | ✗ | ✗ | ✔ | ✗ | 74.03 ± 2.53 |
| | ✗ | ✗ | ✔ | ✔ | **75.83 ± 0.74** |

performance of three different ranking methods, namely confidence ranking, geometric ranking, and Node-reordering (which combines the former two rankings with information retrieval techniques). Moreover, we test the effect of DGI, which aims to eliminate geometrically imbalanced nodes. As shown in Table 3, each component of our method can bring performance improvements. In particular, in three out of four settings in the table, Node-reordering+DGI achieves best F1 scores.

In all cases, using geometric ranking is better than confidence ranking, which empirically verifies our hypothesis that the prediction confidence scores might contain bias and be less reliable.

## 6 RELATED WORK

**Imbalanced learning** Most real-world data is naturally imbalanced. The major challenge in imbalanced scenarios is how to train a fair model which does not biased toward majority classes. There are several commonly used approaches for alleviating this problem. Ensemble learning (Freund & Schapire, 1997; Liu et al., 2008; Zhou et al., 2020; Wang et al., 2020a; Liu et al., 2020; Cai et al., 2021) combines the results of multiple weak classifiers. Data re-sampling methods (Chawla et al., 2002; Han et al., 2005; Smith et al., 2014; Sáez et al., 2015; Kang et al., 2019; Wang et al., 2021a) smooth the label distribution in the training set by synthesizing or duplicating minority class samples. A third class of approaches alleviate the imbalance problem by modifying the loss function, which give larger weights to minority classes or change the margins of different classes (Zhou & Liu, 2005; Tang et al., 2008; Cao et al., 2019; Tang et al., 2020; Xu et al., 2020; Ren et al., 2020; Wang et al., 2021b). Methods based on post-hoc correction compensate minority classes during the inference step, after model training is complete (Kang et al., 2019; Tian et al., 2020; Menon et al., 2020; Hong et al., 2021). Although these techniques have been widely applied on the i.i.d. data, it is not a trivial task to extend them to graph-structured data.

**Imbalanced learning in node classification** Recently, a series of researches (Shi et al., 2020; Wang et al., 2020c; Zhao et al., 2021; Liu et al., 2021; Qu et al., 2021; Chen et al., 2021; Park et al., 2021; Song et al., 2022) explicitly tackle the challenges brought by the topological structures of graph data when handling imbalanced node classification. GraphSMOTE (Zhao et al., 2021) synthesizes minority nodes in embedding space by interpolating two minority nodes using the SMOTE (Chawla et al., 2002) algorithm, and infers the neighborhoods of new nodes with link prediction algorithms. ImGAGN (Qu et al., 2021) generates the features of minority nodes with all of the minority nodes according to the learned weight matrix, and synthesizes the neighborhoods of new nodes by based on weights. Qu et al. (2021) only consider binary classification, and it is computationally expensive to build a generator for each class on multi-classification tasks. GraphENS (Park et al., 2021) works for multi-class node classification, which synthesizes the whole ego network for minority nodes by interpolating the ego networks of two nodes based on their similarity. Chen et al. (2021) identify topology imbalance as a main source of difficulty when handling imbalance on node classification tasks; they propose ReNode, which mitigates topology imbalance by adjusting the weights of nodes according to their distance to class boundaries. TAM (Song et al., 2022) adjusts the scores of different classes in the Softmax function based on local topology and label statistics. To obtain label information of unlabeled nodes, TAM trains the model using the original imbalanced training set and takes the model predictions as proxies for ground-truth labels.

## 7 CONCLUSION

In this work, we observe that selecting unlabeled nodes instead of generating synthetic nodes in oversampling based methods for imbalanced node classification is much simpler and more effective. We propose a novel iterative unlabeled nodes selection and retraining framework, which effectively select high-quality new samples from the unlabeled sets to smooth the label distribution of training set. Moreover, we propose to exploit the geometric structure in the node embedding space to compensate the bias in the model predictions. Extensive experimental results show that UNREAL consistently outperforms existing state-of-the-art approaches by large margins.

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

# A  NOTATION TABLE

**Table 4:** Elaborated notation table of this paper.

| | |
|---|---|
| ***Indices*** | |
| $n$ | The number of nodes,$|\mathcal{V}|$ |
| $f$ | The node feature dimension |
| $k$ | The number of different classes |
| $k'$ | The number of cluster centers in the embedding space |
| $d$ | The dimension of the embedding space, or the dimension of the last layer of GNNs |
| $T$ | Rounds to select nodes |
| ***Parameters*** | |
| $\mathcal{G}$ | An undirected and unweighted graph |
| $\mathcal{V}$ | The node set of $\mathcal{G}$ |
| $\mathcal{E}$ | The edge set of $\mathcal{G}$ |
| $\mathcal{X}$ | The feature matrix of $\mathcal{G}$, $\mathcal{X} \in \mathbb{R}^{n \times f}$ |
| $\mathcal{L}$ | The set of labeled nodes of $\mathcal{G}$ |
| $A$ | The adjacency matrix of $\mathcal{G}$, $A \in \{0,1\}^{n \times n}$ |
| $\mathcal{N}(v)$ | The set of 1-hop neighbors for node $v$ |
| $\mathcal{U}$ | The set of unlabeled nodes, $\mathcal{U} = \mathcal{V} - \mathcal{L}$ |
| $\mathcal{C}_i$ | The $i$ class of the labeled sets |
| $\rho$ | Imbalance ratio of a dataset, $\rho := \frac{\max_i(|\mathcal{C}_i|)}{\min_i(|\mathcal{C}_i|)}$ |
| $h_v^l$ | The feature of node $v$ in the $l$-th layer |
| $e_{v,u}$ | The edge weight between $v$ and $u$ |
| $\Phi^l$ | The parameter matrix of the $l$-th layer |
| $H^L$ | The embedding matrix of labeled nodes, $H^L \in \mathbb{R}^{|\mathcal{L}| \times d}$ |
| $H^U$ | The embedding matrix of unlabeled nodes, $H^U \in \mathbb{R}^{|\mathcal{U}| \times d}$ |
| $h_u^L$ | The embedding of a node $u$, if $u \in \mathcal{L}$ |
| $h_u^U$ | The embedding of a node $u$, if $u \in \mathcal{U}$ |
| $\mathcal{K}_i$ | The $i$-th cluster |
| $c_i$ | The $i$-th cluster center,the center of cluster $i$-th |
| $\tilde{y}_i$ | The pseudo-label of the cluster $\mathcal{K}_i$ |
| $\tilde{\mathcal{U}}_m$ | The combination of clusters with the same pseudo-label $m$ |
| $\hat{y}_u$ | The prediction of node $u$ in $\mathcal{U}$ given by GNN model |
| $\mathcal{U}_m$ | The combination of unlabeled nodes whose prediction given by the GNN model is $m$ |
| $\mathcal{Z}$ | The pool of candidate nodes after DPAM, $\mathcal{Z} = \bigcup\limits_{i=m}^{k} (\tilde{\mathcal{U}}_m \cap \mathcal{U}_m)$ |
| $c_m^{\text{train}}$ | The class center of class $m$ in the embedding space |
| $\mathcal{S}_i$ | The sorted lists of geometric rankings |
| $\mathcal{T}_i$ | The sorted lists of confidence rankings |
| $r_m$ | The similarity between two rankings, $r_m = \text{RBO}(\mathcal{S}_m, \mathcal{T}_m)$ |
| $\delta_u$ | The distance between the embedding of $u$ and the closest class center to $u$ |
| $\beta_u$ | The distance between the embedding of $u$ and the second closest class center to $u$ |
| $\gamma$ | Threshold of DGI |
| $p$ | Weight hyperparameter of RBO |
| $\alpha$ | The size threshold of nodes being added in each class per round |
| $\eta$ | Learning rate of GNN model |
| ***Functions*** | |
| $m_l$ | The message function of MPNNs |
| $\theta_l$ | The information aggregation function |
| $\psi_l$ | The node feature update function |
| $f_{\text{cluster}}$ | An unsupervised clustering algorithm for the embedding space |
| $M(\cdot)$ | The mean function |
| $f_{\text{g}}$ | GNN model |

# B  ADDITIONAL RESULTS OF PSEUDO-LABEL MISJUDGMENT AUGMENTATION PROBLEM

Here, we present the details and results of the experiment which are not reported in Section 3 due to the space constraints.

**Experimental setup**  We first conduct experiments to test the accuracy of pseudo label for unlabeled nodes on class-imbalanced graphs. ST based on different GNN structures are trained on four node classification benchmark datasets, Cora, CiteSeer, PubMed, Amazon-Computers. We process the four datasets with a traditional imbalanced distribution following Zhao et al. (2021); Park et al. (2021); Song et al. (2022). The imbalance ratio $\rho$ between the numbers of the most frequent class and the least frequent class is set as 1, 5, 10, 20, 50, 100. We fix architecture as the 2-layer GNN (i.e. GCN (Kipf & Welling, 2016), GAT (Veličković et al., 2017), GraphSAGE (Hamilton et al., 2017)) having 128 hidden dimensions and train models for 2000 epochs. We select the model by the validation accuracy. We test the accuracy of pseudo labels for unlabeled nodes which are newly added to the training set, more specifically, we separately examine 100 nodes that join the majority class and join the minority class. We repeat each experiment five times and report the average experiment results.

**Pseudo-label Misjudgment Augmentation Problem**  In different imbalanced scenarios for ST, the accuracy of the pseudo labels for the unlabeled nodes which are selected into the minority class and the majority class of the training set respectively are reported in Figure 2, 3, 4, 5 and Table 5. We can find that as $\rho$ becomes larger, the accuracy of pseudo labels for unlabeled nodes selected into the minority class becomes lower, in other words, the influence of the bias of the classifier becomes larger . This means that in an imbalanced scenario, the pseudo-labels given by the classifier are not credible.  Similarly, we also believe that even if the pseudo-label of a node is accurate, the confidence given by the classifier is skewed, which means that we will also possibly put the low-quality unlabeled nodes into the training set, and neglect high-quality unlabeled nodes.  For the unlabeled nodes selected into the majority class, we found that with the increasing degree of imbalance, accuracy of pseudo labels for unlabeled nodes is basically stable at a low level, which also better confirms the bias problem of the classifier.  More importantly, regardless of selecting majority class nodes or minority class nodes, UNREAL consistently outperforms ST.

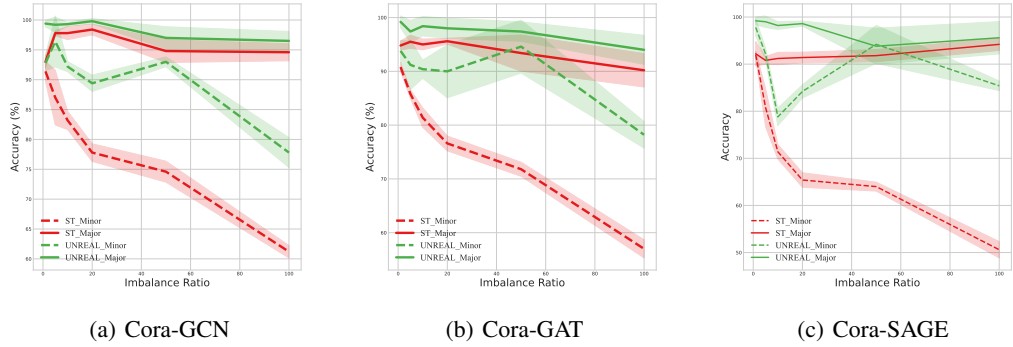

|     |     |     |
| :-: | :-: | :-: |
| (a) Cora-GCN | (b) Cora-GAT | (c) Cora-SAGE |

**Figure 2:** The experimental results on Cora under different imbalance scenarios ($\rho$ = 1, 5, 10, 20, 50, 100). We select 100 unlabeled nodes newly added to the training set through ST & UNREAL, and evaluate the performance of ST & UNREAL by testing the accuracy (%) with the standard errors of these nodes' pseudo labels. ST-Minor, UNREAL-Minor means that we only test unlabeled nodes that are selected into the minority class, and SL-Major, UNREAL-Major means that we only test unlabeled nodes that are selected into the majority class.

**The specific performance of ST**  ST is a classic technique in semi-supervised learning to enhance performance and robustness, e.g., Lee et al. (2013). However, as we have argued and verified above,

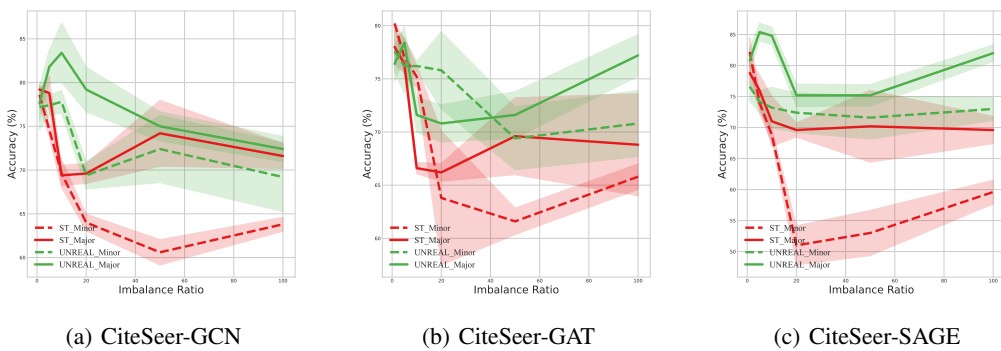

| (a) CiteSeer-GCN | (b) CiteSeer-GAT | (c) CiteSeer-SAGE |

**Figure 3:** The experimental results on CiteSeer under different imbalance scenarios ($\rho = 1, 5, 10, 20,$ 50, 100). We select 100 unlabeled nodes newly added to the training set through ST & UNREAL, and evaluate the performance of ST & UNREAL by testing the accuracy (%) with the standard errors of these nodes' pseudo labels. ST-Minor, UNREAL-Minor means that we only test unlabeled nodes that are selected into the minority class, and SL-Major, UNREAL-Major means that we only test unlabeled nodes that are selected into the majority class.

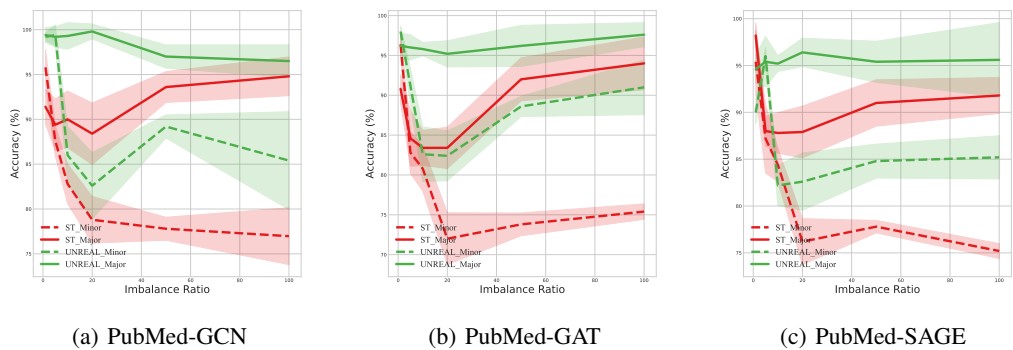

| (a) PubMed-GCN | (b) PubMed-GAT | (c) PubMed-SAGE |

**Figure 4:** The experimental results on PubMed under different imbalance scenarios ($\rho = 1, 5, 10, 20,$ 50, 100). We select 100 unlabeled nodes newly added to the training set through ST & UNREAL, and evaluate the performance of ST & UNREAL by testing the accuracy (%) with the standard errors of these nodes' pseudo labels. ST-Minor, UNREAL-Minor means that we only test unlabeled nodes that are selected into the minority class, and SL-Major, UNREAL-Major means that we only test unlabeled nodes that are selected into the majority class.

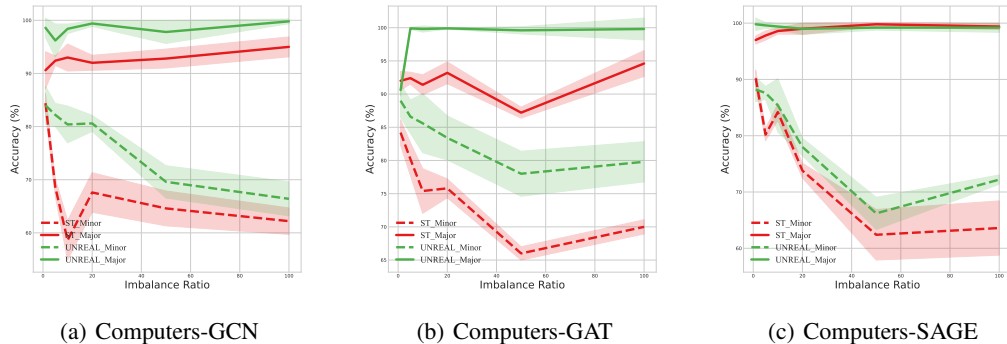

|  (a) Computers-GCN | (b) Computers-GAT | (c) Computers-SAGE |

**Figure 5:** The experimental results on Amazon-Computers under different imbalance scenarios ($\rho$ = 1, 5, 10, 20, 50, 100). We select 100 unlabeled nodes newly added to the training set through ST & UNREAL, and evaluate the performance of ST & UNREAL by testing the accuracy (%) with the standard errors of these nodes' pseudo labels. ST-Minor, UNREAL-Minor means that we only test unlabeled nodes that are selected into the minority class, and SL-Major, UNREAL-Major means that we only test unlabeled nodes that are selected into the majority class.

**Table 5:** Experimental results of ST and UNREAL on four class-imbalanced node classification benchmark datasets with $\rho = 1, 5, 10, 20, 50, 100$. We select 100 unlabeled nodes newly added to the training set through ST & UNREAL, and evaluate the performance of ST & UNREAL by testing the accuracy (%) with the standard errors of the pseudo labels for the seleted nodes, **minor** means that we only evaluate unlabeled nodes thich are selected into the minority class, and **major** means that we only evaluate unlabeled nodes which are selected into the majority class. We report average results over 5 repetitions on three representative GNN architectures.

| | Dataset | Cora | | CiteSeer | | PubMed | | Amazon-Computers | |
|---|---|---|---|---|---|---|---|---|---|
| | Method | Self-Training | UNREAL | Self-Training | UNREAL | Self-Training | UNREAL | Self-Training | UNREAL |
| GCN | $\rho = 1(minor)$ | $91.40 \pm 1.67$ | $92.80 \pm 1.30$ | $78.60 \pm 1.14$ | $77.20 \pm 2.58$ | $95.80 \pm 1.92$ | $99.20 \pm 0.44$ | $84.40 \pm 1.81$ | $84.00 \pm 3.60$ |
| | $\rho = 5(minor)$ | $87.00 \pm 6.59$ | $96.40 \pm 4.27$ | $74.60 \pm 0.83$ | $77.40 \pm 1.20$ | $87.60 \pm 1.67$ | $99.40 \pm 1.14$ | $68.60 \pm 1.67$ | $82.20 \pm 2.16$ |
| | $\rho = 10(minor)$ | $83.20 \pm 1.54$ | $92.20 \pm 0.85$ | $69.60 \pm 1.60$ | $77.80 \pm 1.29$ | $82.80 \pm 2.20$ | $86.00 \pm 3.19$ | $58.60 \pm 5.72$ | $80.40 \pm 3.46$ |
| | $\rho = 20(minor)$ | $77.80 \pm 1.48$ | $89.40 \pm 1.37$ | $64.00 \pm 0.96$ | $69.40 \pm 1.57$ | $78.80 \pm 2.60$ | $82.60 \pm 3.69$ | $67.60 \pm 5.72$ | $80.60 \pm 1.51$ |
| | $\rho = 50(minor)$ | $74.60 \pm 1.76$ | $93.00 \pm 0.82$ | $60.60 \pm 1.45$ | $72.40 \pm 3.84$ | $77.80 \pm 1.28$ | $89.20 \pm 1.29$ | $64.60 \pm 3.28$ | $69.60 \pm 3.04$ |
| | $\rho = 100(minor)$ | $61.20 \pm 1.04$ | $77.80 \pm 2.50$ | $63.80 \pm 0.80$ | $69.20 \pm 3.96$ | $76.97 \pm 3.17$ | $85.40 \pm 5.45$ | $62.20 \pm 4.49$ | $66.40 \pm 3.20$ |
| | $\rho = 1(major)$ | $97.00 \pm 1.00$ | $99.40 \pm 0.54$ | $79.20 \pm 0.01$ | $77.60 \pm 1.94$ | $91.40 \pm 1.94$ | $96.80 \pm 0.83$ | $90.60 \pm 3.50$ | $98.60 \pm 1.81$ |
| | $\rho = 5(major)$ | $97.80 \pm 0.44$ | $99.20 \pm 1.28$ | $78.80 \pm 1.92$ | $81.80 \pm 1.91$ | $89.40 \pm 2.88$ | $95.60 \pm 1.14$ | $92.40 \pm 0.89$ | $96.20 \pm 3.11$ |
| | $\rho = 10(major)$ | $97.80 \pm 1.09$ | $99.30 \pm 0.34$ | $69.40 \pm 1.14$ | $83.40 \pm 3.40$ | $90.00 \pm 3.16$ | $94.20 \pm 1.48$ | $93.00 \pm 2.54$ | $98.40 \pm 0.89$ |
| | $\rho = 20(major)$ | $98.40 \pm 0.89$ | $99.80 \pm 0.44$ | $69.60 \pm 1.14$ | $79.20 \pm 2.55$ | $88.40 \pm 3.43$ | $95.80 \pm 0.83$ | $92.00 \pm 1.41$ | $99.40 \pm 0.54$ |
| | $\rho = 50(major)$ | $94.80 \pm 1.92$ | $97.00 \pm 1.87$ | $74.20 \pm 3.77$ | $72.00 \pm 1.66$ | $93.60 \pm 1.72$ | $96.20 \pm 1.30$ | $92.80 \pm 1.78$ | $97.80 \pm 2.16$ |
| | $\rho = 100(major)$ | $94.60 \pm 1.43$ | $96.50 \pm 1.58$ | $71.60 \pm 7.43$ | $72.40 \pm 1.44$ | $94.80 \pm 2.16$ | $96.40 \pm 1.81$ | $95.00 \pm 1.87$ | $99.80 \pm 0.44$ |
| GAT | $\rho = 1(minor)$ | $90.80 \pm 0.83$ | $93.80 \pm 1.92$ | $80.20 \pm 0.04$ | $77.60 \pm 1.94$ | $96.40 \pm 1.14$ | $98.00 \pm 0.70$ | $84.20 \pm 1.92$ | $89.00 \pm 2.54$ |
| | $\rho = 5(minor)$ | $85.80 \pm 0.81$ | $91.20 \pm 4.60$ | $77.00 \pm 1.83$ | $76.20 \pm 2.58$ | $82.80 \pm 2.62$ | $91.20 \pm 2.24$ | $80.20 \pm 2.48$ | $86.60 \pm 2.50$ |
| | $\rho = 10(minor)$ | $81.40 \pm 1.81$ | $90.40 \pm 1.69$ | $75.20 \pm 1.03$ | $76.20 \pm 0.44$ | $80.80 \pm 2.81$ | $82.60 \pm 3.43$ | $75.40 \pm 3.36$ | $85.60 \pm 4.44$ |
| | $\rho = 20(minor)$ | $76.60 \pm 1.38$ | $90.00 \pm 9.92$ | $63.80 \pm 6.30$ | $75.80 \pm 3.63$ | $72.00 \pm 3.25$ | $82.40 \pm 3.13$ | $75.80 \pm 1.42$ | $83.40 \pm 3.31$ |
| | $\rho = 50(minor)$ | $71.80 \pm 1.31$ | $94.60 \pm 4.92$ | $61.60 \pm 1.25$ | $69.40 \pm 2.96$ | $73.80 \pm 1.43$ | $88.60 \pm 1.27$ | $66.00 \pm 1.00$ | $78.00 \pm 3.39$ |
| | $\rho = 100(minor)$ | $57.00 \pm 1.69$ | $78.20 \pm 2.47$ | $65.80 \pm 1.20$ | $70.80 \pm 3.11$ | $75.40 \pm 0.97$ | $91.00 \pm 3.43$ | $70.00 \pm 1.07$ | $79.80 \pm 3.03$ |
| | $\rho = 1(major)$ | $94.80 \pm 0.83$ | $99.20 \pm 1.09$ | $78.00 \pm 1.58$ | $76.40 \pm 1.62$ | $90.80 \pm 1.78$ | $96.20 \pm 1.64$ | $92.00 \pm 1.41$ | $90.60 \pm 0.89$ |
| | $\rho = 5(major)$ | $95.50 \pm 1.22$ | $97.40 \pm 2.07$ | $76.40 \pm 0.89$ | $78.40 \pm 1.53$ | $84.60 \pm 3.50$ | $96.00 \pm 1.58$ | $92.40 \pm 0.89$ | $99.90 \pm 0.15$ |
| | $\rho = 10(major)$ | $95.00 \pm 1.00$ | $98.40 \pm 1.81$ | $66.60 \pm 0.54$ | $71.60 \pm 1.64$ | $83.40 \pm 2.19$ | $95.80 \pm 0.83$ | $91.40 \pm 1.51$ | $99.80 \pm 0.46$ |
| | $\rho = 20(major)$ | $95.60 \pm 0.54$ | $98.00 \pm 1.87$ | $66.20 \pm 0.83$ | $70.80 \pm 1.76$ | $83.40 \pm 2.60$ | $95.20 \pm 1.64$ | $93.20 \pm 1.64$ | $99.90 \pm 0.12$ |
| | $\rho = 50(major)$ | $93.40 \pm 3.36$ | $97.40 \pm 1.81$ | $69.60 \pm 3.62$ | $71.60 \pm 2.19$ | $92.00 \pm 5.70$ | $96.20 \pm 2.58$ | $87.20 \pm 0.83$ | $99.60 \pm 0.54$ |
| | $\rho = 100(major)$ | $90.20 \pm 3.11$ | $94.00 \pm 2.70$ | $68.80 \pm 5.80$ | $77.20 \pm 1.97$ | $94.00 \pm 3.31$ | $97.60 \pm 1.51$ | $94.60 \pm 1.94$ | $99.80 \pm 1.64$ |
| SAGE | $\rho = 1(minor)$ | $92.00 \pm 0.70$ | $97.80 \pm 1.78$ | $82.20 \pm 2.28$ | $76.60 \pm 2.40$ | $95.40 \pm 3.36$ | $90.00 \pm 1.22$ | $90.20 \pm 1.48$ | $88.20 \pm 2.16$ |
| | $\rho = 5(minor)$ | $80.80 \pm 4.05$ | $92.20 \pm 1.32$ | $74.00 \pm 1.71$ | $74.20 \pm 1.78$ | $87.20 \pm 3.67$ | $96.00 \pm 0.90$ | $80.20 \pm 1.09$ | $87.60 \pm 1.14$ |
| | $\rho = 10(minor)$ | $71.40 \pm 1.54$ | $78.8 \pm 1.82$ | $68.80 \pm 1.03$ | $73.20 \pm 3.27$ | $84.40 \pm 1.79$ | $82.20 \pm 2.13$ | $84.20 \pm 1.48$ | $85.40 \pm 4.72$ |
| | $\rho = 20(minor)$ | $65.40 \pm 1.54$ | $84.20 \pm 1.39$ | $51.60 \pm 3.16$ | $72.40 \pm 3.20$ | $76.20 \pm 2.45$ | $82.60 \pm 3.06$ | $73.80 \pm 1.30$ | $78.00 \pm 1.55$ |
| | $\rho = 50(minor)$ | $64.00 \pm 0.95$ | $94.20 \pm 8.04$ | $53.00 \pm 3.65$ | $71.60 \pm 3.46$ | $77.80 \pm 0.67$ | $84.80 \pm 1.81$ | $62.40 \pm 4.49$ | $66.20 \pm 2.86$ |
| | $\rho = 100(minor)$ | $50.60 \pm 1.74$ | $85.40 \pm 1.02$ | $59.60 \pm 1.93$ | $73.00 \pm 1.87$ | $75.20 \pm 0.79$ | $85.20 \pm 2.30$ | $63.60 \pm 4.82$ | $72.20 \pm 0.83$ |
| | $\rho = 1(major)$ | $92.20 \pm 2.58$ | $99.20 \pm 0.83$ | $78.80 \pm 1.92$ | $80.80 \pm 1.97$ | $98.20 \pm 1.30$ | $94.60 \pm 1.51$ | $97.00 \pm 0.71$ | $99.80 \pm 1.09$ |
| | $\rho = 5(major)$ | $90.80 \pm 0.83$ | $99.00 \pm 1.22$ | $76.00 \pm 2.54$ | $85.40 \pm 1.45$ | $88.00 \pm 1.58$ | $95.40 \pm 2.70$ | $97.80 \pm 0.83$ | $99.60 \pm 0.54$ |
| | $\rho = 10(major)$ | $91.20 \pm 1.30$ | $98.20 \pm 0.83$ | $71.00 \pm 4.00$ | $84.80 \pm 1.39$ | $87.80 \pm 2.16$ | $95.20 \pm 0.84$ | $98.60 \pm 0.54$ | $99.40 \pm 0.54$ |
| | $\rho = 20(major)$ | $91.40 \pm 1.14$ | $98.60 \pm 0.54$ | $69.60 \pm 1.14$ | $75.20 \pm 1.80$ | $87.90 \pm 2.77$ | $96.40 \pm 1.51$ | $99.00 \pm 1.00$ | $99.00 \pm 1.00$ |
| | $\rho = 50(major)$ | $91.80 \pm 1.30$ | $93.80 \pm 3.83$ | $70.20 \pm 5.80$ | $75.20 \pm 1.74$ | $91.00 \pm 4.47$ | $95.40 \pm 2.19$ | $99.80 \pm 0.44$ | $99.20 \pm 0.44$ |
| | $\rho = 100(major)$ | $94.20 \pm 1.30$ | $95.60 \pm 3.43$ | $69.60 \pm 2.19$ | $82.00 \pm 1.35$ | $91.80 \pm 1.92$ | $95.60 \pm 3.97$ | $99.40 \pm 0.54$ | $99.20 \pm 0.83$ |

for highly imbalanced data, ST is unlikely to achieve optimal performance as biased and untrustworthy predictions may bring low-quality nodes into the training set in the early stage. Our key idea to remedy this is to exploit the geometric structural information in the embedding space. In this sec-

tion, we empirically verify the informativeness of geometric structures by comparing UNREAL with pure self-training schemes.

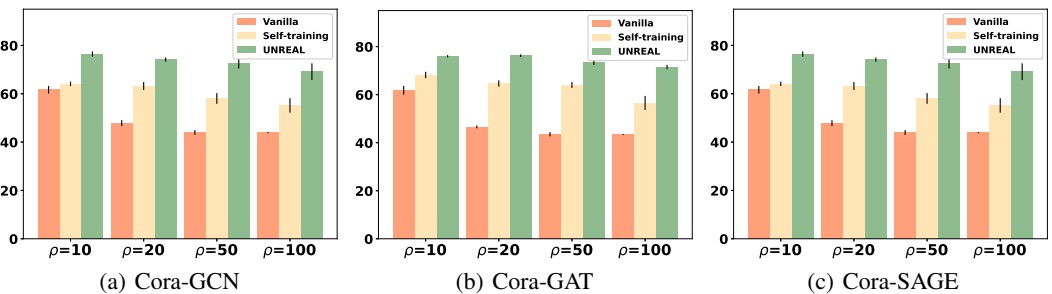

**Figure 6:** The experimental results on Cora under different imbalance scenarios ($\rho = 10, 20, 50, 100$). We compare the F1-score (%) with the standard errors of ST and UNREAL.

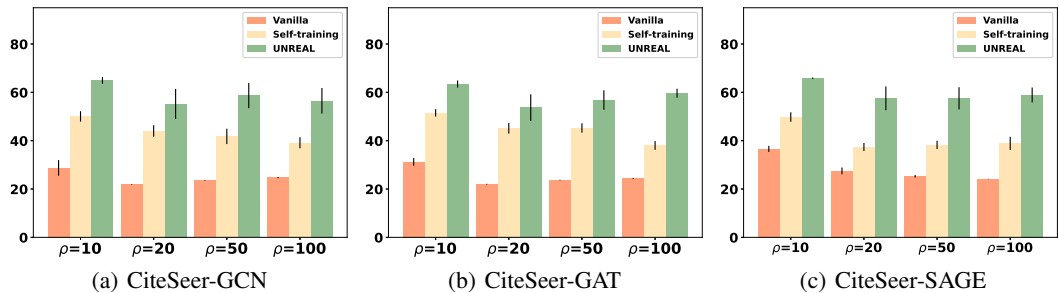

**Figure 7:** The experimental results on CiteSeer under different imbalance scenarios ($\rho = 10, 20, 50, 100$). We compare the F1-score (%) with the standard errors of ST and UNREAL.

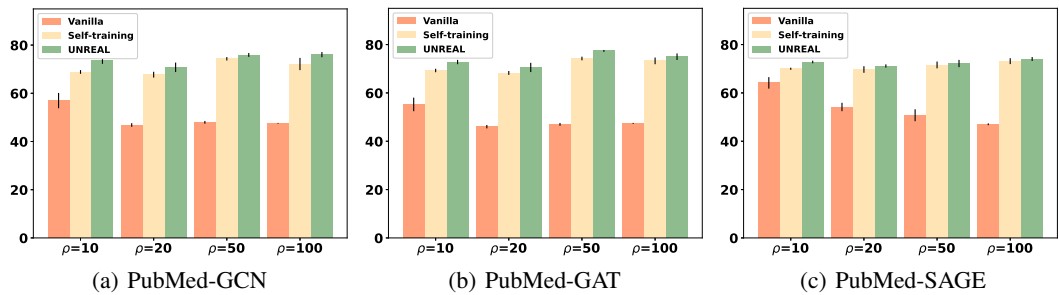

**Figure 8:** The experimental results on PubMed under different imbalance scenarios ($\rho = 10, 20, 50, 100$). We compare the F1-score (%) with the standard errors of ST and UNREAL.

**Results**  The size of added nodes in each round for each class is a hyperparameter, and we tune the hyperparameter based on the accuracy on the validation set. We repeat each experiment five times and report the average experiment results on the node classification benchmark datasets under different imbalance ratios in Figure 6, 7, 8. It can be observed that across different ratios, UN-REAL consistently outperforms self-training by a large margin, and as imbalance ratio increases, the gap of the F1 scores between ST and our method becomes larger. This shows that as the data imbalance issue become more severe, the performance of ST degrades more rapidly, which is likely due to noise introduced in early rounds.

## C  ADDITIONAL RESULTS IN DIFFERENT SCENARIOS

### C.1  MORE RESULTS ON HIGHER IMBALANCE RATIOS

In this section, we show the performance of UNREAL in highly-imbalanced scenarios by constructing training sets with $\rho = 50, 100$ on the benchmark datasets which is not presented in the main paper. The results are presented in Table 6 and Table 7. We can find that our model is more robust on highly-imbalanced datasets with different architectures, namely GCN (Kipf & Welling, 2016), GAT (Veličković et al., 2017), GraphSAGE (Hamilton et al., 2017)). It is shown that UNREAL can deal with different degrees of imbalance, and significantly outperforms other methods with a large margin. We observe from the Table 6 and Table 7 that the performance of of GraphENS (also GraphENS+TAM) degrades noticeably on highly-imbalanced datasets. In highly imbalanced scenarios, synthesizing or duplicating huge amounts of nodes based on rare minority nodes in the training set is prone to overfitting of minority classes. On the other hand, BalancedSoftmax+TAM achieves an overall better performance than GraphENS+TAM in those highly imbalanced scenarios.

### C.2  RESULTS ON FLICKR

The Flickr is naturally imbalanced and the training set in the public split is also imbalanced, so we directly evaluate the performance of all methods on the public split (Zeng et al., 2019). The results is presented in Table 8.

## D  ADDITIONAL ANALYSIS FOR EACH COMPONENTS OF UNREAL

### D.1  ADDITIONAL ANALYSIS FOR DPAM

In this section, we analyze why DPAM works. DPAM conducts an unsupervised algorithm to obtain pseudo-labels for each unlabeled node in the embedding space, and finally only unlabeled nodes whose pseudo-labels and classifier's predictions are aligned are put into the candidate pool, which effectively circumvents the bias problem of the classifier, such as pseudo-label misjudgment of unlabeled nodes, selecting low-quality nodes into training set based on the skewed condidence rankings. To quantify the performance of DPAM, we conduct the novel experiments below.

**Experimental setup**  We use DPAM to filter the unlabeled nodes of the whole graph, and test the accuracy of pseudo-labels (prediction of the classifier) of the aligned node set $\mathcal{U}_{in}$ and the discarded node set $\mathcal{U}_{out}$ respectively. DPAM based on different GNN structures are trained on two node classification benchmark datasets, Cora, Amazon-Computers. We process the two dataset with a traditional imbalanced distribution following Zhao et al. (2021); Park et al. (2021); Song et al. (2022). The imbalance ratio $\rho$ between the numbers of the most frequent class and the least frequent class is set as 1, 5, 10, 20, 50, 100. We fix architecture as the 2-layer GNN (i.e. GCN(Kipf & Welling, 2016), GAT(Veličković et al., 2017), GraphSAGE(Hamilton et al., 2017)) having 128 hidden dimensions and train models for 2000 epochs. We select the model by the validation accuracy. We observe the accuracy of pseudo labels for unlabeled nodes which are filtered out and absorbed into by DPAM respectively. We repeat each experiment five times and present the average experiment results.

**Result**  DPAM divides the unlabeled nodes of the whole graph into two parts, $\mathcal{U}_{in}, \mathcal{U}_{out}$. We verify the effect of DPAM by testing the accuracy of pseudo-labels for these two parts of nodes. We can observe that the accuracy of pseudo-labels for $\mathcal{U}_{in}$ and $\mathcal{U}_{out}$ differ greatly in different imbalanced scenarios. Usually the pseudo-label accuracy of $\mathcal{U}_{in}$ is high and the pseudo-label accuracy of $\mathcal{U}_{out}$ is lower, which means the effectiveness of DPAM. We can also observe that as $\rho$ increases, the accuracy of both decreases, which also reflects the model bias caused by the imbalanced label distribution.

### D.2  ADDITIONAL ANALYSIS FOR NODE-REORDERING AND DGI

In this section, we analyze why Node-Reordering and DGI works. With DPAM, we filter out a large part of untrustworthy nodes, and get a pool of candidate nodes. We try to carefully hunt for a part of high-quality nodes in the pool to add to the training set, which involves a priority issue. As we mentioned before, we have already verified in Section 3 that the prediction and confidence given by

**Table 6:** Experimental results of our method UNREAL and other baselines on four class-imbalanced node classification benchmark datasets with $\rho = 50$. We report averaged balanced accuracy (bAcc.,%) and F1-score (%) with the standard errors over 5 repetitions on three representative GNN architectures.

| Dataset | | Cora | | CiteSeer | | PubMed | | Amazon-Computers | |
|---|---|---|---|---|---|---|---|---|---|
| **Imbalance Ratio** ($\rho = 50$) | | bAcc. | F1 | bAcc. | F1 | bAcc. | F1 | bAcc. | F1 |
| **GCN** | Vanilla | 51.81 ± 0.62 | 43.98 ± 1.00 | 37.59 ± 0.17 | 23.54 ± 0.13 | 61.65 ± 0.34 | 47.95 ± 0.58 | 77.36 ± 3.41 | 69.68 ± 3.12 |
| | Re-Weight | 58.54 ± 2.39 | 54.13 ± 3.20 | 38.19 ± 1.28 | 27.43 ± 2.34 | 65.70 ± 1.59 | 56.35 ± 4.26 | 79.10 ± 2.44 | 71.40 ± 2.86 |
| | PC Softmax | 64.87 ± 2.23 | 62.01 ± 3.14 | 42.42 ± 2.19 | 38.83 ± 2.70 | 69.21 ± 0.59 | 69.40 ± 0.87 | 81.90 ± 1.63 | 74.34 ± 2.13 |
| | BalancedSoftmax | 65.94 ± 1.55 | 64.00 ± 2.05 | 47.62 ± 1.11 | 46.55 ± 1.46 | 70.40 ± 1.00 | 69.04 ± 0.66 | 82.97 ± 0.83 | 73.74 ± 1.27 |
| | Renode | 62.22 ± 1.76 | 61.18 ± 2.24 | 41.23 ± 1.66 | 33.66 ± 2.69 | 68.67 ± 1.21 | 63.05 ± 1.47 | 81.71 ± 0.99 | 72.55 ± 1.61 |
| | GraphENS | 63.47 ± 0.98 | 62.21 ± 1.65 | 48.17 ± 1.58 | 41.07 ± 2.34 | 69.63 ± 2.55 | 64.30 ± 3.51 | 81.63 ± 2.35 | 72.57 ± 2.33 |
| | BalancedSoftmax+TAM | 68.57 ± 1.58 | 67.25 ± 1.27 | 53.43 ± 2.42 | 51.74 ± 2.80 | 77.20 ± 1.45 | 74.86 ± 0.99 | 81.74 ± 2.30 | 73.85 ± 2.68 |
| | Renode+TAM | 63.93 ± 1.96 | 61.64 ± 2.71 | 48.17 ± 1.58 | 41.07 ± 2.34 | 69.63 ± 2.55 | 64.30 ± 3.51 | 80.55 ± 1.75 | 72.33 ± 1.63 |
| | GraphENS+TAM | 65.05 ± 1.11 | 62.11 ± 1.98 | 45.03 ± 1.34 | 42.65 ± 1.94 | 69.74 ± 0.78 | 70.82 ± 0.63 | 81.69 ± 2.22 | 72.09 ± 1.75 |
| | **UNREAL** | **75.62 ± 2.02** | **72.59 ± 2.13** | **59.97 ± 4.59** | **58.66 ± 5.20** | **78.55 ± 0.84** | **75.91 ± 0.81** | **85.54 ± 0.26** | **75.76 ± 0.13** |
| | **Δ** | **+7.05** | **+5.34** | **+6.54** | **+6.92** | **+1.35** | **+1.06** | **+2.57** | **+1.91** |
| **GAT** | Vanilla | 53.90 ± 0.63 | 45.53 ± 0.89 | 36.48 ± 0.08 | 23.68 ± 0.16 | 60.16 ± 0.47 | 46.99 ± 0.58 | 72.42 ± 2.17 | 64.41 ± 2.68 |
| | Re-Weight | 59.78 ± 1.92 | 56.69 ± 2.21 | 38.70 ± 2.23 | 29.38 ± 3.06 | 66.27 ± 0.68 | 57.34 ± 1.41 | 73.46 ± 3.07 | 67.00 ± 2.60 |
| | PC Softmax | 59.44 ± 2.62 | 58.06 ± 2.69 | 43.13 ± 1.56 | 37.04 ± 2.07 | 70.86 ± 0.44 | 70.96 ± 0.54 | 77.21 ± 2.90 | 69.17 ± 2.89 |
| | BalancedSoftmax | 64.71 ± 2.28 | 62.55 ± 2.61 | 51.89 ± 1.15 | 49.36 ± 1.52 | 70.94 ± 1.09 | 70.33 ± 0.99 | 77.49 ± 1.58 | 70.44 ± 2.33 |
| | Renode | 63.81 ± 1.72 | 60.63 ± 2.26 | 41.60 ± 2.30 | 33.94 ± 4.60 | 70.35 ± 1.26 | 67.43 ± 0.01 | 72.39 ± 2.75 | 65.23 ± 3.35 |
| | GraphENS | 64.52 ± 2.51 | 61.41 ± 3.15 | 45.23 ± 2.97 | 41.12 ± 4.23 | 69.66 ± 1.01 | 66.83 ± 0.94 | 78.36 ± 2.74 | 70.44 ± 2.51 |
| | BalancedSoftmax+TAM | 68.05 ± 1.03 | 66.07 ± 1.14 | 54.28 ± 0.79 | 52.77 ± 0.97 | 75.65 ± 1.11 | 74.02 ± 1.44 | 78.86 ± 1.53 | 70.71 ± 2.04 |
| | Renode+TAM | 64.40 ± 1.83 | 63.48 ± 2.83 | 43.54 ± 1.54 | 35.80 ± 2.43 | 71.23 ± 2.04 | 66.61 ± 4.31 | 76.07 ± 2.70 | 68.43 ± 2.68 |
| | GraphENS+TAM | 65.33 ± 2.67 | 65.34 ± 2.53 | 48.00 ± 1.46 | 48.14 ± 1.43 | 71.50 ± 1.26 | 72.58 ± 1.07 | 80.02 ± 2.32 | 72.38 ± 2.47 |
| | **UNREAL** | **77.07 ± 0.83** | **73.44 ± 1.05** | **57.70 ± 4.35** | **56.81 ± 4.67** | **79.41 ± 0.29** | **77.38 ± 0.39** | **86.06 ± 0.45** | **77.55 ± 0.71** |
| | **Δ** | **+9.02** | **+7.37** | **+3.42** | **+4.04** | **+3.76** | **+3.36** | **+6.04** | **+5.17** |
| **SAGE** | Vanilla | 53.02 ± 0.83 | 45.58 ± 1.30 | 38.81 ± 0.89 | 25.28 ± 0.51 | 61.41 ± 1.01 | 50.46 ± 2.47 | 56.53 ± 2.12 | 48.52 ± 2.75 |
| | Re-Weight | 58.03 ± 0.81 | 54.32 ± 0.99 | 38.49 ± 1.34 | 30.41 ± 1.82 | 62.41 ± 0.90 | 51.37 ± 2.62 | 70.36 ± 2.21 | 61.52 ± 2.73 |
| | PC Softmax | 62.33 ± 1.62 | 59.97 ± 1.98 | 41.79 ± 1.19 | 36.90 ± 0.84 | 69.58 ± 1.09 | 67.13 ± 0.95 | 73.53 ± 2.02 | 66.12 ± 3.19 |
| | BalancedSoftmax | 64.57 ± 0.77 | 62.22 ± 0.82 | 41.84 ± 1.72 | 40.09 ± 1.04 | 70.43 ± 0.38 | 68.99 ± 0.99 | 73.27 ± 2.30 | 68.30 ± 1.77 |
| | Renode | 61.35 ± 1.86 | 58.88 ± 2.53 | 40.37 ± 2.33 | 32.57 ± 3.62 | 67.54 ± 3.05 | 59.77 ± 5.30 | 70.46 ± 3.45 | 62.30 ± 4.40 |
| | GraphENS | 63.95 ± 0.96 | 62.63 ± 2.12 | 41.99 ± 1.54 | 37.44 ± 2.43 | 66.07 ± 1.12 | 61.63 ± 1.85 | 76.21 ± 2.84 | 68.10 ± 2.56 |
| | BalancedSoftmax+TAM | 65.97 ± 0.71 | 65.53 ± 0.88 | 52.89 ± 1.65 | 49.92 ± 1.83 | 71.11 ± 0.75 | 71.73 ± 0.79 | 73.12 ± 1.41 | 66.45 ± 1.04 |
| | Renode+TAM | 62.79 ± 0.47 | 61.05 ± 0.82 | 43.04 ± 1.30 | 36.97 ± 1.92 | 71.79 ± 1.33 | 67.80 ± 2.45 | 74.55 ± 2.95 | 66.06 ± 2.16 |
| | GraphENS+TAM | 65.98 ± 1.37 | 64.84 ± 1.13 | 49.54 ± 1.79 | 49.48 ± 1.70 | 73.24 ± 1.32 | 73.73 ± 1.14 | 80.75 ± 1.22 | 72.31 ± 0.95 |
| | **UNREAL** | **76.04 ± 1.30** | **72.99 ± 1.25** | **58.70 ± 4.10** | **57.53 ± 4.59** | **75.27 ± 1.26** | **72.16 ± 1.50** | **82.03 ± 0.77** | **72.98 ± 0.52** |
| | **Δ** | **+10.06** | **+7.46** | **+5.81** | **+7.61** | **+2.03** | **-1.57** | **+1.28** | **+0.67** |

**Table 7:** Experimental results of our method UNREAL and other baselines on four class-imbalanced node classification benchmark datasets with $\rho = 100$. We report averaged balanced accuracy (bAcc.,%) and F1-score (%) with the standard errors over 5 repetitions on three representative GNN architectures.

| Dataset | | Cora | | CiteSeer | | PubMed | | Amazon-Computers | |
|---|---|---|---|---|---|---|---|---|---|
| **Imbalance Ratio** ($\rho = 100$) | | bAcc. | F1 | bAcc. | F1 | bAcc. | F1 | bAcc. | F1 |
| **GCN** | Vanilla | 51.62 ± 0.20 | 43.91 ± 0.25 | 38.83 ± 0.26 | 24.71 ± 0.25 | 61.28 ± 0.12 | 47.55 ± 0.16 | 76.09 ± 3.79 | 69.32 ± 3.49 |
| | Re-Weight | 59.11 ± 1.06 | 54.04 ± 1.36 | 42.67 ± 2.06 | 33.17 ± 3.40 | 67.14 ± 2.71 | 55.24 ± 5.36 | 81.53 ± 2.20 | 71.45 ± 2.05 |
| | PC Softmax | 63.75 ± 1.02 | 61.19 ± 1.43 | 38.34 ± 0.71 | 33.65 ± 1.42 | 70.85 ± 0.44 | 70.26 ± 0.63 | 82.22 ± 1.99 | 72.38 ± 2.52 |
| | BalancedSoftmax | 63.03 ± 1.57 | 61.28 ± 1.77 | 48.49 ± 1.20 | 46.59 ± 1.34 | 70.77 ± 1.88 | 68.88 ± 1.74 | 83.33 ± 3.35 | 74.34 ± 2.74 |
| | Renode | 60.76 ± 2.53 | 58.09 ± 3.00 | 43.41 ± 2.07 | 33.69 ± 2.76 | 67.63 ± 2.77 | 61.70 ± 4.84 | 82.13 ± 1.73 | 71.79 ± 1.85 |
| | GraphENS | 63.00 ± 1.30 | 62.33 ± 1.67 | 45.99 ± 2.26 | 37.23 ± 3.40 | 68.65 ± 1.00 | 62.17 ± 1.60 | 83.37 ± 2.17 | 73.96 ± 1.98 |
| | BalancedSoftmax+TAM | 69.44 ± 0.59 | 67.10 ± 0.88 | 52.60 ± 0.69 | 51.21 ± 0.84 | 73.73 ± 1.10 | 73.72 ± 0.83 | 83.70 ± 2.17 | 75.39 ± 1.43 |
| | Renode+TAM | 64.19 ± 1.46 | 60.90 ± 1.56 | 44.78 ± 1.51 | 35.90 ± 2.61 | 70.53 ± 0.75 | 64.30 ± 1.79 | 82.32 ± 2.19 | 73.09 ± 1.75 |
| | GraphENS+TAM | 60.40 ± 4.42 | 57.77 ± 4.02 | 42.72 ± 2.54 | 39.40 ± 2.57 | 70.73 ± 1.96 | 72.50 ± 1.87 | 81.29 ± 1.52 | 71.66 ± 1.75 |
| | **UNREAL** | **72.82 ± 3.55** | **69.12 ± 3.45** | **57.66 ± 1.96** | **56.50 ± 1.12** | **78.73 ± 0.88** | **76.03 ± 1.08** | **84.30 ± 0.30** | **76.06 ± 0.32** |
| | **Δ** | **+3.38** | **+2.02** | **+5.06** | **+5.29** | **+5.00** | **+2.31** | **+0.60** | **+0.67** |
| **GAT** | Vanilla | 51.58 ± 0.32 | 43.37 ± 0.21 | 37.91 ± 0.28 | 23.49 ± 0.21 | 62.07 ± 0.17 | 47.39 ± 0.20 | 72.66 ± 2.97 | 64.87 ± 3.46 |
| | Re-Weight | 58.28 ± 1.88 | 54.47 ± 2.35 | 38.13 ± 1.55 | 29.60 ± 3.02 | 67.41 ± 2.69 | 58.06 ± 5.07 | 77.10 ± 3.26 | 68.35 ± 2.71 |
| | PC Softmax | 63.74 ± 2.01 | 59.76 ± 2.19 | 45.07 ± 1.13 | 39.21 ± 2.29 | 69.68 ± 1.29 | 69.44 ± 1.29 | 79.72 ± 1.52 | 70.78 ± 1.45 |
| | BalancedSoftmax | 63.19 ± 1.35 | 61.03 ± 1.46 | 46.03 ± 2.11 | 43.38 ± 2.24 | 71.45 ± 1.23 | 69.40 ± 1.20 | 79.15 ± 2.08 | 69.68 ± 2.13 |
| | Renode | 60.04 ± 2.21 | 58.04 ± 2.66 | 42.40 ± 2.97 | 34.09 ± 0.04 | 68.54 ± 2.11 | 65.63 ± 3.15 | 75.34 ± 1.65 | 69.99 ± 1.60 |
| | GraphENS | 63.93 ± 2.70 | 61.77 ± 3.38 | 44.43 ± 1.90 | 39.26 ± 2.55 | 66.07 ± 1.12 | 64.14 ± 3.28 | 81.63 ± 2.08 | 71.20 ± 2.75 |
| | BalancedSoftmax+TAM | 64.96 ± 3.23 | 62.91 ± 3.96 | 52.75 ± 1.29 | 50.69 ± 1.83 | 73.38 ± 0.77 | 72.45 ± 0.88 | 80.86 ± 2.52 | 72.93 ± 2.95 |
| | Renode+TAM | 63.45 ± 1.41 | 61.51 ± 1.95 | 41.55 ± 1.39 | 36.13 ± 2.87 | 71.53 ± 2.35 | 68.11 ± 4.28 | 78.60 ± 1.90 | 70.35 ± 2.80 |
| | GraphENS+TAM | 62.52 ± 0.95 | 61.65 ± 1.19 | 45.79 ± 1.31 | 44.80 ± 1.14 | 69.09 ± 1.61 | | 83.33 ± 0.83 | 72.81 ± 1.22 |
| | **UNREAL** | **75.42 ± 0.91** | **71.50 ± 0.89** | **60.35 ± 1.87** | **59.63 ± 1.86** | **77.88 ± 1.31** | **74.98 ± 1.35** | **85.33 ± 0.19** | **75.83 ± 0.74** |
| | **Δ** | **+10.46** | **+8.59** | **+7.60** | **+8.94** | **+4.50** | **+2.53** | **+2.00** | **+3.02** |
| **SAGE** | Vanilla | 52.65 ± 0.24 | 43.79 ± 0.47 | 36.63 ± 0.09 | 24.12 ± 0.09 | 62.29 ± 0.25 | 47.02 ± 0.38 | 55.94 ± 2.37 | 47.21 ± 2.73 |
| | Re-Weight | 59.42 ± 2.88 | 55.26 ± 4.40 | 36.24 ± 1.37 | 27.07 ± 2.88 | 63.33 ± 0.75 | 55.11 ± 1.62 | 70.76 ± 3.35 | 62.09 ± 3.30 |
| | PC Softmax | 64.01 ± 1.15 | 60.74 ± 1.68 | 44.74 ± 1.41 | 37.61 ± 1.69 | 72.62 ± 1.42 | 70.95 ± 1.70 | 75.96 ± 2.44 | 69.12 ± 2.90 |
| | BalancedSoftmax | 63.43 ± 2.12 | 62.30 ± 2.27 | 49.33 ± 1.12 | 44.58 ± 1.64 | 70.68 ± 0.92 | 69.15 ± 0.84 | 74.66 ± 0.86 | 66.28 ± 1.92 |
| | Renode | 62.42 ± 0.90 | 60.08 ± 1.19 | 39.61 ± 2.66 | 30.13 ± 3.86 | 67.11 ± 1.12 | 61.09 ± 3.50 | 73.73 ± 2.26 | 64.47 ± 2.39 |
| | GraphENS | 63.09 ± 0.97 | 61.20 ± 1.74 | 42.03 ± 1.88 | 36.71 ± 2.99 | 69.71 ± 1.87 | 63.47 ± 3.87 | 81.33 ± 1.66 | 72.83 ± 1.76 |
| | BalancedSoftmax+TAM | 66.58 ± 1.53 | 64.56 ± 2.49 | 53.33 ± 1.06 | 50.15 ± 1.45 | 72.59 ± 2.06 | 72.22 ± 2.08 | 78.01 ± 1.06 | 71.02 ± 1.08 |
| | Renode+TAM | 62.06 ± 2.08 | 60.72 ± 3.32 | 42.08 ± 1.88 | 33.19 ± 3.45 | 69.95 ± 1.01 | 65.99 ± 2.28 | 74.81 ± 3.29 | 67.48 ± 3.32 |
| | GraphENS+TAM | 65.95 ± 2.25 | 63.88 ± 1.78 | 51.03 ± 1.51 | 50.49 ± 1.88 | 73.58 ± 2.01 | 72.44 ± 1.77 | 81.72 ± 1.08 | 72.31 ± 1.98 |
| | **UNREAL** | **73.47 ± 2.31** | **68.30 ± 2.11** | **59.77 ± 2.98** | **58.92 ± 3.07** | **77.11 ± 0.59** | **74.03 ± 0.81** | **82.92 ± 2.94** | **73.11 ± 2.57** |
| | **Δ** | **+6.89** | **+3.74** | **+6.44** | **+8.43** | **+3.53** | **+1.59** | **+1.20** | **+0.28** |

**Table 8:** Experimental results of our method UNREAL and other baselines on Flickr . We report averaged balanced accuracy (bAcc.,%) and F1-score (%) with the standard errors over 5 repetitions on three representative GNN architectures.

| Dataset (Flickr) | GCN | | GAT | | SAGE | |
|---|---|---|---|---|---|---|
| **Imbalance Ratio**($\rho \approx 10.80$) | bAcc. | F1 | bAcc. | F1 | bAcc. | F1 |
| Cross Entropy | $24.62 \pm 0.07$ | $24.53 \pm 0.11$ | $25.87 \pm 0.30$ | $25.32 \pm 0.44$ | $25.29 \pm 0.18$ | $24.16 \pm 0.27$ |
| Re-Weight | $28.31 \pm 1.64$ | $24.06 \pm 1.16$ | $\mathbf{30.66 \pm 0.76}$ | $27.12 \pm 0.34$ | $27.39 \pm 1.84$ | $22.62 \pm 1.04$ |
| PC Softmax | $\underline{29.21 \pm 2.16}$ | $\underline{25.81 \pm 1.75}$ | $30.20 \pm 0.46$ | $\underline{27.24 \pm 0.37}$ | $25.40 \pm 2.49$ | $21.08 \pm 1.73$ |
| BalancedSoftmax | $27.61 \pm 0.61$ | $23.70 \pm 0.77$ | $26.01 \pm 2.81$ | $23.50 \pm 3.07$ | $28.24 \pm 2.10$ | $24.98 \pm 1.59$ |
| GraphSMOTE | OOM | OOM | OOM | OOM | OOM | OOM |
| Renode | OOM | OOM | OOM | OOM | OOM | OOM |
| GraphENS | OOM | OOM | OOM | OOM | OOM | OOM |
| BalancedSoftmax+TAM | $27.06 \pm 1.03$ | $23.97 \pm 0.60$ | $28.24 \pm 0.99$ | $25.52 \pm 0.89$ | $\underline{29.79 \pm 0.37}$ | $\underline{27.56 \pm 0.25}$ |
| Renode+TAM | OOM | OOM | OOM | OOM | OOM | OOM |
| GraphENS+TAM | OOM | OOM | OOM | OOM | OOM | OOM |
| **UNREAL** | $\mathbf{30.76 \pm 0.27}$ | $\mathbf{30.60 \pm 0.29}$ | $\underline{29.45 \pm 0.72}$ | $\mathbf{28.21 \pm 0.76}$ | $\mathbf{53.68 \pm 0.63}$ | $\mathbf{54.01 \pm 1.34}$ |
| $\Delta$ | **+1.55** | **+4.79** | **-1.21** | **+0.97** | **+23.89** | **+26.45** |

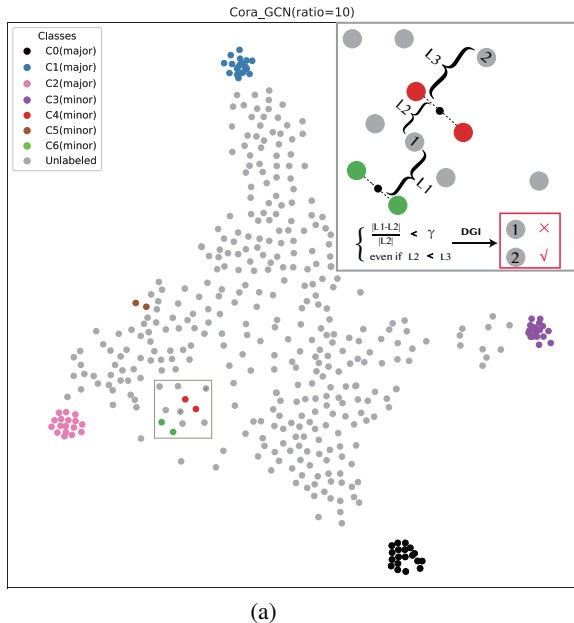

(a)

**Figure 9:** An elaboration of Geometric Imbalance and DGI, we use T-SNE to visualize the all embeddings of nodes in the training set and the part of embeddings of unlabeled nodes.

**Table 9:** Experimental results of DPAM effectiveness on **Cora** with $\rho = 1, 5, 10, 20, 50, 100$. We observe the accuracy (%) of the pseudo-label (prediction of the classifier) of the aligned node set $\mathcal{U}_{in}$ and the discarded node set $\mathcal{U}_{out}$ respectively. We report averaged results with the standard errors over 5 repetitions on three representative GNN architectures. **All**, **Labeled**, **Unlabeled** represent the size of whole nodes, labeled nodes, and unlabeled nodes on the graph. **Align**, **Out**, **Align-True**, **Out-Ture** represent the size of $\mathcal{U}_{in}, \mathcal{U}_{out}$, nodes with accurate pseudo-labels of $\mathcal{U}_{in}, \mathcal{U}_{out}$ respectively.

| | Dataset | All | Labled | Unlabled | Align | Align-True | Accuracy(%) | Out | Out-True | Accuracy(%) |
|---|---|---|---|---|---|---|---|---|---|---|
| GCN | $\rho = 1$ | 2708 | 140 | 2568 | 2072.00 ± 10.29 | 1391.00 ± 22.56 | **67.11 ± 1.17** | 496.00 ± 10.29 | 233.80 ± 16.66 | **47.17 ± 3.74** |
| | $\rho = 5$ | 2708 | 92 | 2616 | 2122.80 ± 18.93 | 1392.00 ± 34.21 | **65.58 ± 1.57** | 493.20 ± 18.73 | 186.80 ± 13.08 | **37.86 ± 1.75** |
| | $\rho = 10$ | 2708 | 86 | 2622 | 2134.60 ± 23.42 | 1326.40 ± 24.23 | **62.14 ± 1.67** | 487.40 ± 23.43 | 181.60 ± 18.24 | **37.32 ± 3.13** |
| | $\rho = 20$ | 2708 | 83 | 2625 | 2149.60 ± 17.67 | 1310.20 ± 86.72 | **60.97 ± 3.50** | 475.40 ± 17.67 | 169.80 ± 21.47 | **35.64 ± 3.44** |
| | $\rho = 50$ | 2708 | 203 | 2505 | 1860.80 ± 31.15 | 1059.40 ± 58.77 | **56.90 ± 2.62** | 644.20 ± 31.14 | 225.80 ± 10.70 | **35.05 ± 3.79** |
| | $\rho = 100$ | 2708 | 403 | 2305 | 1820.40 ± 12.42 | 1001.60 ± 21.60 | **55.02 ± 3.99** | 484.60 ± 23.99 | 151.40 ± 20.74 | **31.78 ± 2.37** |
| GAT | $\rho = 1$ | 2708 | 140 | 2568 | 2072.00 ± 37.18 | 1412.40 ± 37.31 | **68.16 ± 1.41** | 496.00 ± 20.89 | 239.40 ± 11.37 | **48.29 ± 2.15** |
| | $\rho = 5$ | 2708 | 92 | 2616 | 2141.40 ± 26.36 | 1433.00 ± 59.82 | **66.90 ± 2.09** | 474.60 ± 26.36 | 195.20 ± 24.68 | **41.02 ± 3.27** |
| | $\rho = 10$ | 2708 | 86 | 2622 | 2132.60 ± 29.94 | 1377.40 ± 49.61 | **64.58 ± 1.60** | 489.40 ± 29.95 | 185.80 ± 12.28 | **37.97 ± 1.13** |
| | $\rho = 20$ | 2708 | 83 | 2625 | 2150.60 ± 37.35 | 1344.60 ± 54.17 | **62.16 ± 1.64** | 462.40 ± 33.28 | 178.00 ± 5.05 | **38.60 ± 2.12** |
| | $\rho = 1$ | 2708 | 140 | 2568 | 1892.40 ± 37.18 | 1080.80 ± 31.86 | **57.52 ± 1.52** | 612.60 ± 37.17 | 271.20 ± 6.30 | **44.35 ± 1.86** |
| | $\rho = 1$ | 2708 | 403 | 2305 | 1934.60 ± 19.65 | 1038.20 ± 21.08 | **53.66 ± 0.83** | 370.40 ± 37.17 | 147.53 ± 3.20 | **39.83 ± 1.36** |
| SAGE | $\rho = 1$ | 2708 | 140 | 2568 | 1944.00 ± 25.77 | 973.40 ± 32.26 | **51.27 ± 3.36** | 624.00 ± 25.77 | 237.00 ± 13.28 | **36.11 ± 4.07** |
| | $\rho = 5$ | 2708 | 92 | 2616 | 2004.40 ± 35.50 | 1038.20 ± 22.53 | **51.80 ± 3.73** | 611.60 ± 35.50 | 203.80 ± 7.15 | **33.40 ± 1.85** |
| | $\rho = 10$ | 2708 | 86 | 2622 | 2041.60 ± 32.48 | 1039.00 ± 41.32 | **50.89 ± 1.88** | 580.40 ± 32.48 | 189.20 ± 2.35 | **32.56 ± 4.25** |
| | $\rho = 20$ | 2708 | 83 | 2625 | 2040.20 ± 30.94 | 1002.20 ± 66.97 | **48.95 ± 2.66** | 578.80 ± 30.95 | 186.60 ± 18.00 | **32.18 ± 1.57** |
| | $\rho = 50$ | 2708 | 203 | 2505 | 1789.40 ± 30.56 | 870.20 ± 24.33 | **48.63 ± 1.03** | 715.60 ± 30.56 | 242.40 ± 16.77 | **33.87 ± 1.18** |
| | $\rho = 100$ | 2708 | 403 | 2305 | 1859.00 ± 192.42 | 914.41 ± 23.65 | **49.26 ± 2.59** | 446.00 ± 21.24 | 138.87 ± 6.32 | **31.15 ± 2.43** |

**Table 10:** Experimental results of DPAM effectiveness on **Amazon-Computers** with $\rho = 1, 5, 10, 20, 50, 100$. We observe the accuracy (%) of the pseudo-label (prediction of the classifier) of the aligned node set $\mathcal{U}_{in}$ and the discarded node set $\mathcal{U}_{out}$ respectively. We report averaged results with the standard errors over 5 repetitions on three representative GNN architectures. **All**, **Labeled**, **Unlabeled** represent the size of whole nodes, labeled nodes, and unlabeled nodes on the graph. **Align**, **Out**, **Align-True**, **Out-Ture** represent the size of $\mathcal{U}_{in}, \mathcal{U}_{out}$, nodes with accurate pseudo-labels of $\mathcal{U}_{in}, \mathcal{U}_{out}$ respectively.

| | Dataset | All | Labeled | Unlabeled | Align | Align-True | Accuracy(%) | Out | Out-True | Accuracy(%) |
|---|---|---|---|---|---|---|---|---|---|---|
| GCN | $\rho = 1$ | 13752 | 200 | 13552 | 11977.60 ± 108.09 | 9603.80 ± 93.34 | **80.08 ± 3.07** | 1554.40 ± 08.23 | 676.60 ± 141.11 | **43.58 ± 2.83** |
| | $\rho = 5$ | 13752 | 120 | 13632 | 11593.60 ± 73.16 | 9172.80 ± 87.32 | **79.06 ± 1.17** | 2308.40 ± 173.54 | 544.40 ± 66.26 | **30.74 ± 9.09** |
| | $\rho = 10$ | 13752 | 110 | 13642 | 11822.40 ± 13.43 | 8786.60 ± 55.48 | **74.24 ± 0.83** | 1807.60 ± 109.34 | 495.00 ± 100.37 | **27.24 ± 4.30** |
| | $\rho = 20$ | 13752 | 105 | 13647 | 11866.60 ± 17.34 | 8698.20 ± 188.13 | **73.40 ± 1.39** | 1780.40 ± 67.36 | 521.00 ± 60.76 | **29.20 ± 2.41** |
| | $\rho = 50$ | 13752 | 255 | 13497 | 11843.20 ± 168.20 | 8994.40 ± 175.24 | **75.94 ± 0.75** | 1653.80 ± 138.11 | 474.20 ± 50.72 | **28.68 ± 2.16** |
| | $\rho = 100$ | 13752 | 505 | 13247 | 9159.00 ± 192.42 | 7352.90 ± 61.23 | **81.41 ± 4.59** | 4088.00 ± 93.99 | 1129.60 ± 75.74 | **28.67 ± 4.77** |
| GAT | $\rho = 1$ | 13752 | 200 | 13552 | 12008.00 ± 101.93 | 9984.20 ± 308.03 | **83.44 ± 4.13** | 1544.80 ± 101.94 | 580.40 ± 190.49 | **43.33 ± 1.32** |
| | $\rho = 5$ | 13752 | 120 | 13632 | 11570.80 ± 136.11 | 8715.00 ± 86.33 | **75.33 ± 0.54** | 2061.20 ± 136.13 | 477.00 ± 97.07 | **25.39 ± 1.33** |
| | $\rho = 10$ | 13752 | 110 | 13642 | 8947.60 ± 13.40 | 6680.40 ± 177.54 | **75.85 ± 6.07** | 4694.40 ± 134.74 | 591.80 ± 13.74 | **15.94 ± 2.97** |
| | $\rho = 20$ | 13752 | 105 | 13647 | 10245.80 ± 68.00 | 7300.80 ± 64.89 | **71.42 ± 1.80** | 3401.20 ± 69.76 | 370.60 ± 43.87 | **18.52 ± 0.09** |
| | $\rho = 50$ | 13752 | 255 | 13497 | 10133.60 ± 31.56 | 7772.00 ± 155.87 | **77.17 ± 2.85** | 3363.40 ± 10.42 | 457.20 ± 108.19 | **19.28 ± 1.43** |
| | $\rho = 100$ | 13752 | 505 | 13247 | 11377.00 ± 63.32 | 9122.20 ± 96.70 | **80.46 ± 1.01** | 1910.00 ± 63.32 | 458.20 ± 41.04 | **24.78 ± 2.04** |
| SAGE | $\rho = 1$ | 13752 | 200 | 13552 | 10815.20 ± 86.50 | 7131.40 ± 72.83 | **65.94 ± 0.28** | 2736.80 ± 86.50 | 965.40 ± 56.42 | **35.26 ± 1.31** |
| | $\rho = 5$ | 13752 | 120 | 13632 | 10627.80 ± 78.33 | 6728.00 ± 53.24 | **63.25 ± 0.36** | 3004.20 ± 78.03 | 978.20 ± 59.93 | **32.55 ± 1.49** |
| | $\rho = 10$ | 13752 | 110 | 13642 | 10475.00 ± 118.41 | 6015.00 ± 41.14 | **57.43 ± 4.01** | 3167.00 ± 18.41 | 1064.40 ± 52.71 | **33.59 ± 6.23** |
| | $\rho = 20$ | 13752 | 105 | 13647 | 10653.20 ± 87.35 | 5998.40 ± 69.35 | **56.30 ± 4.01** | 2993.80 ± 87.35 | 886.20 ± 73.25 | **29.57 ± 1.77** |
| | $\rho = 50$ | 13752 | 255 | 13497 | 11044.80 ± 129.14 | 6760.80 ± 50.26 | **61.22 ± 3.42** | 2442.20 ± 28.48 | 879.00 ± 91.45 | **35.71 ± 1.78** |
| | $\rho = 100$ | 13752 | 505 | 13247 | 9175.20 ± 32.53 | 6475.60 ± 80.88 | **72.07 ± 1.96** | 4071.80 ± 32.63 | 1218.60 ± 14.70 | **34.43 ± 1.08** |

the classifier are biased, resulting in low accuracy of the pseudo-labels for nodes selected by ST in highly imbalanced scenarios. We can get the geometric ranking according to the distance between the unlabeled nodes and the class centers in the embedding space. Considering the influence of classifier bias on confidence ranking, we believe that geometric ranking is more credible in the early rounds. At the same time, we take into account the suboptimal nature of the unsupervised algorithm. We believe that with the rounds of UNREAL increases, the label distribution of the training set is gradually balanced, and the confidence given by the classifier is more reliable. Node-reordering considers both geometric ranking and confidence ranking, specifically, obtain the similarity between them to get a weight to reorder the priority of the nodes. To quantify the performance of Node-Reordering and DGI, we conduct the novel experiments below.

**Experimental setup** We conduct experiments to test the accuracy of pseudo labels for unlabeled nodes on class-imbalanced graphs. All model combinations based on different GNN structures are trained on two node classification benchmark datasets, Cora, Amaon-Computers. We process the two dataset with a traditional imbalanced distribution following Zhao et al. (2021); Park et al. (2021); Song et al. (2022). The imbalance ratio $\rho$ between the numbers of the most frequent class and the

least frequent class is set as 1, 5, 10, 20, 50, 100. We fix architecture as the 2-layer GNN (i.e. GCN(Kipf & Welling, 2016), GAT(Veličković et al., 2017), GraphSAGE(Hamilton et al., 2017)) having 128 hidden dimensions and train models for 2000 epochs. We select the model by the validation accuracy. We observe the accuracy of pseudo labels for unlabeled nodes which are newly added to the minority class of training set. We repeat each experiment five times and present the average experiment results.

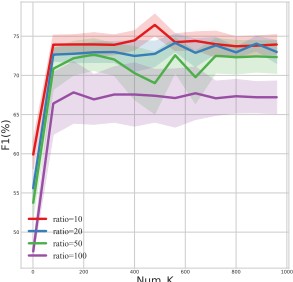
(a) Sensitivity performance of $k'$ of K-Means

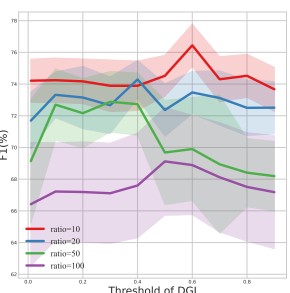
(b) Sensitivity performance of the threshold $\gamma$ of DGI

**Figure 10:** Sensitivity analysis on Cora based on GCN. The two images show the performance change as clusters' size $k'$ of K-Means and the threshold $\gamma$ of DGI increases respectively.

**Result**  As shown in Table 11 and Table 12, we verify the effectiveness of each component of UNREAL by testing the accuracy of the nodes' pseudo-labels selected by different model combinations, DPAM+Confidence ranking(with or without DGI), DPAM+Geometric ranking(with or without DGI), DPAM+Node-Reordering(with or without DGI). It can be observed that in different imbalanced scenarios, each component of UNREAL (Node-reordering & DGI) plays an important role, and the performance outperforms the other model combinations significantly.

## E    HYPERPARAMETER SENSITIVITY ANALYSIS OF UNREAL

We investigate the sensitivity of performance to clusters' size $k'$ of K-Means algorithm and the threshold $\gamma$ of DGI in Figure 10. We observe the performance gradually stabilize when $k'$ have extremely high values, on the other hand, when $k'$ is extremely low values, the performance of UN-REAL drops largely. We believe that when $k'$ is too small, the pseudo-labels given by unsupervised algorithms will have more errors. Also, we observe the performance gradually stabilize when $\gamma$ have extremely low values. We believe this is because that the DGI screening is too strict, which will lead to the loss of some high-quality nodes. On the other hand, extremely large $\gamma$ will introduce many noise into the training set.

## F    DETAILS OF THE EXPERIMENTAL SETUP

Here, we introduce the method of imbalanced datasets construction, evaluation protocol, and the details of our algorithm and baseline methods.

### F.1    IMBALANCED DATASETS CONSTRUCTION

The detailed descriptions of the datasets are shown in Table 13. For each citation dataset, for $\rho = 10, 20$, we follow the "public" split, and randomly convert minority class nodes to unlabeled nodes until the dataset reaches imbalanced ratio $\rho$. For $\rho = 50, 100$, since there are not enough nodes per class in the public split training set, we choose randomly selected nodes as training samples, and for validation and test sets we still follow the public split. For the co-purchased networks Amazon-Computers, we randomly select nodes as training set in each replicated experiment, and construct a random validation set with 30 nodes in each class, and treat the remaining nodes as testing set.

**Table 11:** Analyzed experimental results of Node-Reordering and DGI on **Cora** with $\rho = 1, 5, 10, 20, 50, 100$. We select 100 unlabeled nodes newly added to the minority class of training set through different method combinations, and evaluate the validity of Node-Reordering & DGI by testing the accuracy (%) with the standard errors of the pseudo labels for these nodes. We report averaged results over 5 repetitions on three representative GNN architectures.

| Dataset | | Cora | | | | | |
|---|---|---|---|---|---|---|---|
| **Imbalance Ratio** $(\rho)$ | | $\rho = 1$ | $\rho = 5$ | $\rho = 10$ | $\rho = 20$ | $\rho = 50$ | $\rho = 100$ |
| GCN | DPAM+Confidence Ranking | $61.40 \pm 2.73$ | $62.40 \pm 2.59$ | $60.20 \pm 1.02$ | $58.40 \pm 1.05$ | $57.60 \pm 1.86$ | $58.40 \pm 2.15$ |
| | DPAM+Geometric Ranking | $64.00 \pm 3.67$ | $61.20 \pm 2.89$ | $61.20 \pm 2.54$ | $63.60 \pm 1.31$ | $55.60 \pm 2.31$ | $47.80 \pm 2.87$ |
| | DPAM+Node-Reordering | $89.65 \pm 3.23$ | $86.98 \pm 0.21$ | $88.32 \pm 0.83$ | $85.32 \pm 2.98$ | $90.87 \pm 2.31$ | $71.60 \pm 2.91$ |
| | DPAM+Confidence Ranking+DGI | $71.00 \pm 5.47$ | $75.40 \pm 2.15$ | $68.20 \pm 1.25$ | $69.40 \pm 1.28$ | $67.80 \pm 2.75$ | $66.60 \pm 0.16$ |
| | DPAM+Geometric Ranking+DGI | $69.60 \pm 3.78$ | $73.80 \pm 0.45$ | $64.80 \pm 1.26$ | $64.20 \pm 1.91$ | $57.00 \pm 1.57$ | $69.00 \pm 1.71$ |
| | **DPAM+Node-Reordering+DGI(UNREAL)** | $\mathbf{92.80 \pm 1.30}$ | $\mathbf{96.40 \pm 4.27}$ | $\mathbf{92.20 \pm 0.85}$ | $\mathbf{89.40 \pm 1.37}$ | $\mathbf{93.00 \pm 0.82}$ | $\mathbf{77.80 \pm 2.50}$ |
| GAT | DPAM+Confidence Ranking | $61.60 \pm 4.26$ | $64.00 \pm 2.07$ | $62.60 \pm 3.47$ | $57.80 \pm 1.65$ | $58.20 \pm 1.07$ | $60.60 \pm 0.79$ |
| | DPAM+Geometric Ranking | $64.00 \pm 2.78$ | $67.80 \pm 3.76$ | $65.00 \pm 4.30$ | $52.00 \pm 1.02$ | $65.20 \pm 2.58$ | $40.80 \pm 2.63$ |
| | DPAM+Node-Reordering | $91.79 \pm 0.23$ | $90.45 \pm 5.78$ | $84.32 \pm 3.45$ | $88.34 \pm 0.23$ | $90.32 \pm 0.43$ | $75.34 \pm 1.54$ |
| | DPAM+Confidence Ranking+DGI | $69.80 \pm 2.77$ | $72.80 \pm 3.94$ | $72.40 \pm 1.13$ | $67.60 \pm 1.59$ | $71.60 \pm 9.12$ | $64.00 \pm 1.74$ |
| | DPAM+Geometric Ranking+DGI | $73.60 \pm 4.82$ | $74.00 \pm 5.47$ | $68.40 \pm 1.62$ | $57.20 \pm 2.17$ | $68.00 \pm 1.17$ | $62.00 \pm 1.53$ |
| | **DPAM+Node-Reordering+DGI(UNREAL)** | $\mathbf{93.80 \pm 1.92}$ | $\mathbf{91.20 \pm 4.60}$ | $\mathbf{90.40 \pm 1.69}$ | $\mathbf{90.00 \pm 9.92}$ | $\mathbf{94.60 \pm 4.92}$ | $\mathbf{78.20 \pm 2.47}$ |
| SAGE | DPAM+Confidence Ranking | $54.80 \pm 4.96$ | $53.00 \pm 2.46$ | $51.80 \pm 1.97$ | $43.60 \pm 2.57$ | $46.20 \pm 0.53$ | $41.60 \pm 1.14$ |
| | DPAM+Geometric Ranking | $53.60 \pm 2.78$ | $45.40 \pm 1.75$ | $40.60 \pm 0.26$ | $52.60 \pm 2.47$ | $47.40 \pm 4.27$ | $44.80 \pm 2.84$ |
| | DPAM+Node-Reordering | $90.69 \pm 0.21$ | $86.90 \pm 0.56$ | $86.45 \pm 3.21$ | $88.34 \pm 2.43$ | $75.34 \pm 4.20$ | $76.43 \pm 1.43$ |
| | DPAM+Confidence Ranking+DGI | $66.20 \pm 5.78$ | $59.00 \pm 3.04$ | $63.80 \pm 1.52$ | $54.60 \pm 1.64$ | $60.60 \pm 1.37$ | $57.40 \pm 2.26$ |
| | DPAM+Geometric Ranking+DGI | $61.60 \pm 3.71$ | $61.80 \pm 5.21$ | $54.00 \pm 7.31$ | $53.60 \pm 1.38$ | $63.00 \pm 1.23$ | $45.20 \pm 1.96$ |
| | **DPAM+Node-Reordering+DGI(UNREAL)** | $\mathbf{97.80 \pm 1.78}$ | $\mathbf{92.20 \pm 1.32}$ | $\mathbf{90.80 \pm 1.82}$ | $\mathbf{89.20 \pm 1.39}$ | $\mathbf{94.20 \pm 8.04}$ | $\mathbf{85.40 \pm 1.02}$ |

**Table 12:** Analyzed experimental results of Node-Reordering and DGI on **Amazon-Computers** with $\rho = 1, 5, 10, 20, 50, 100$. We select 100 unlabeled nodes newly added to the minority class of training set through different method combinations, and evaluate the validity of Node-Reordering & DGI by testing the accuracy (%) with the standard errors of the pseudo labels for these nodes. We report averaged results over 5 repetitions on three representative GNN architectures.

| Dataset | | Amazon-Computers | | | | | |
|---|---|---|---|---|---|---|---|
| **Imbalance Ratio** $(\rho)$ | | $\rho = 1$ | $\rho = 5$ | $\rho = 10$ | $\rho = 20$ | $\rho = 50$ | $\rho = 100$ |
| GCN | DPAM+Confidence Ranking | $75.40 \pm 2.50$ | $70.20 \pm 3.03$ | $74.88 \pm 3.11$ | $68.20 \pm 4.20$ | $63.60 \pm 2.30$ | $61.40 \pm 1.51$ |
| | DPAM+Geometric Ranking | $76.00 \pm 1.41$ | $74.80 \pm 4.71$ | $76.80 \pm 2.28$ | $65.80 \pm 3.27$ | $64.80 \pm 3.70$ | $65.60 \pm 3.98$ |
| | DPAM+Node-Reordering | $82.80 \pm 2.38$ | $79.60 \pm 3.64$ | $78.20 \pm 0.26$ | $74.00 \pm 3.23$ | $65.20 \pm 1.87$ | $66.00 \pm 2.82$ |
| | DPAM+Confidence Ranking+DGI | $76.40 \pm 2.07$ | $67.20 \pm 4.32$ | $75.80 \pm 2.38$ | $66.20 \pm 3.70$ | $62.80 \pm 0.12$ | $59.20 \pm 1.30$ |
| | DPAM+Geometric Ranking+DGI | $78.20 \pm 0.83$ | $80.00 \pm 1.22$ | $76.40 \pm 1.67$ | $66.00 \pm 2.44$ | $64.20 \pm 3.83$ | $66.20 \pm 2.38$ |
| | **DPAM+Node-Reordering+DGI(UNREAL)** | $\mathbf{84.40 \pm 3.60}$ | $\mathbf{82.20 \pm 2.16}$ | $\mathbf{80.40 \pm 3.46}$ | $\mathbf{80.60 \pm 1.51}$ | $\mathbf{69.60 \pm 3.04}$ | $\mathbf{66.40 \pm 3.20}$ |
| GAT | DPAM+Confidence Ranking | $84.60 \pm 2.40$ | $79.20 \pm 1.78$ | $73.00 \pm 2.12$ | $74.80 \pm 2.16$ | $65.00 \pm 1.73$ | $68.60 \pm 1.40$ |
| | DPAM+Geometric Ranking | $86.00 \pm 3.80$ | $79.80 \pm 2.94$ | $74.80 \pm 3.42$ | $75.00 \pm 2.91$ | $70.80 \pm 2.16$ | $69.40 \pm 1.10$ |
| | DPAM+Node-Reordering | $87.40 \pm 2.30$ | $80.60 \pm 3.04$ | $80.40 \pm 2.19$ | $79.00 \pm 3.67$ | $75.00 \pm 1.22$ | $73.40 \pm 2.52$ |
| | DPAM+Confidence Ranking+DGI | $84.20 \pm 1.64$ | $79.40 \pm 2.07$ | $76.40 \pm 6.50$ | $76.00 \pm 2.34$ | $66.00 \pm 0.12$ | $72.00 \pm 1.84$ |
| | DPAM+Geometric Ranking+DGI | $83.80 \pm 1.09$ | $80.20 \pm 1.09$ | $76.20 \pm 2.28$ | $77.80 \pm 2.58$ | $71.60 \pm 0.89$ | $69.00 \pm 1.16$ |
| | **DPAM+Node-Reordering+DGI(UNREAL)** | $\mathbf{89.00 \pm 2.54}$ | $\mathbf{86.60 \pm 2.50}$ | $\mathbf{85.60 \pm 4.44}$ | $\mathbf{83.40 \pm 3.31}$ | $\mathbf{78.00 \pm 3.39}$ | $\mathbf{79.80 \pm 3.03}$ |
| SAGE | DPAM+Confidence Ranking | $85.20 \pm 3.38$ | $80.20 \pm 6.26$ | $84.8 \pm 0.83$ | $77.60 \pm 0.89$ | $61.00 \pm 0.70$ | $65.40 \pm 2.65$ |
| | DPAM+Geometric Ranking | $86.00 \pm 0.70$ | $81.20 \pm 2.16$ | $83.40 \pm 1.14$ | $78.00 \pm 1.22$ | $61.40 \pm 0.54$ | $65.00 \pm 1.72$ |
| | DPAM+Node-Reordering | $86.00 \pm 1.58$ | $83.20 \pm 3.27$ | $84.60 \pm 0.54$ | $79.20 \pm 1.92$ | $61.80 \pm 0.44$ | $67.80 \pm 1.03$ |
| | DPAM+Confidence Ranking+DGI | $86.40 \pm 2.07$ | $81.60 \pm 3.20$ | $83.40 \pm 1.14$ | $\mathbf{79.20 \pm 0.44}$ | $61.20 \pm 0.44$ | $70.40 \pm 3.59$ |
| | DPAM+Geometric Ranking+DGI | $87.00 \pm 2.12$ | $80.80 \pm 2.48$ | $84.20 \pm 1.30$ | $78.20 \pm 1.48$ | $61.20 \pm 0.47$ | $68.20 \pm 1.72$ |
| | **DPAM+Node-Reordering+DGI(UNREAL)** | $\mathbf{88.20 \pm 2.16}$ | $\mathbf{87.60 \pm 1.14}$ | $\mathbf{85.40 \pm 4.72}$ | $78.00 \pm 1.55$ | $\mathbf{66.20 \pm 2.86}$ | $\mathbf{72.20 \pm 0.83}$ |

**Table 13:** Summary of the datasets used in our experiments.

| Dataset | Nodes | Edges | Features | Classes |
|---|---|---|---|---|
| Cora | 2,708 | 5,429 | 1,433 | 7 |
| Citeseer | 3,327 | 4,732 | 3,703 | 6 |
| Pubmed | 19,717 | 44,338 | 500 | 3 |
| Amazon-Computers | 13,752 | 491,722 | 767 | 10 |
| Flickr | 89,250 | 899,756 | 500 | 7 |

For Flickr, we follow the dataset split from Zeng et al. (2019). The details of label distribution in training set of the five imbalanced benchmark datasets are in Table 14, and the label distribution of full graph is provided in Table 15.

## F.2 DETAILS OF GNNS

We evaluate our method with three classic GNN architectures, namely GCN (Kipf & Welling, 2016), GAT (Veličković et al., 2017), and GraphSAGE (Hamilton et al., 2017). GNN consists of $L = 1, 2, 3$ layers and each GNN layer is followed by a BatchNorm layer (momentum=0.99) and a PRelu activation (He et al., 2015). For GAT, we adopt multi-head attention with 8 heads. We search for the best model on the validation set. The choices of hidden unit size for each layer are 64, 128, 256.

## F.3 EVALUATION PROTOCOL

We adopt Adam (Kingma & Ba, 2014) optimizer with initial learning rate 0.01 or 0.005. We follow (Song et al., 2022) to devise a scheduler, which cuts the learning rate by half if there is no decrease on validation loss for 100 consecutive epochs. All learnable parameters in the model adopt weight decay with rate 0.0005. For the first training iteration, we train the model for 200 epochs using the original training set for Cora, CiteSeer, PubMed or Amazon-Computers. For Flickr, we train for 2000 epochs in the first iteration. We train models for 2000 epochs in the rest of the iteration with the above optimizer and scheduler. The best models are selected based on validation accuracy. Early stopping is used with patience set to 300.

## F.4 IMPLEMENTATION DETAILS

In UNREAL, we employ the vanilla K-means algorithm as the unsupervised clustering method. The number of clusters $K$ is chosen from $\{100, 300, 500, 700, 900\}$ for Cora, CiteSeer, PubMed and Amaon-Computers. For Flickr, $K$ is selected among $\{1000, 2000, 3000, 5000\}$. For Cora, CiteSeer, PubMed, and Amazon-Computers, the number of training round $T$ is tuned among $\{40, 60, 80, 100\}$. For Flickr, $T$ is tuned among $\{40, 50, 60, 70\}$. We also introduce a hyperparameter $\alpha$, which is the upper bound on the number of nodes being added per class per round. The tuning range of $\alpha$ is $\{4, 6, 8, 10\}$ for Cora, CiteSeer, Amazon-Computers and $\{64, 72, 80\}$ for PubMed. For Flickr the value of $\alpha$ is selected among $\{30, 40, 50, 60\}$. The weight parameters $p$ in RBO is selected among $\{0.5, 0.75, 0.98\}$, and the threshold in DGI is tuned among $\{0.25, 0.5, 0.75, 1.00\}$. For Flickr, we only add minority nodes to the training set in all iterations, which means that we set $\alpha = 0$ for majority classes in Flickr.

## F.5 BASELINES

For GraphSMOTE (Zhao et al., 2021), we use the branched algorithms whose edge predictions are discrete-valued, which have achieved superior performance over other variants in most experiments. For the ReNode method (Chen et al., 2021), we search hyperparameters among lower bound of cosine annealing $w_{min} \in \{0.25, 0.5, 0.75\}$ and upper bound of the cosine annealing $w_{max} \in \{1.25, 1.5, 1.75\}$ following Chen et al. (2021). PageRank teleport probability is fixed as $\alpha = 0.15$, which is the default setting in the released codes. For TAM (Song et al., 2022), we search the best hyperparameters among the coefficient of ACM term $\alpha \in \{1.25, 1.5, 1.75\}$, the coefficient of ADM term $\beta \in \{0.125, 0.25, 0.5\}$, and the minimum temperature of class-wise temperature $\phi \in \{0.8, 1.2\}$ following Song et al. (2022). The sensitivity to imbalance ratio of class-wise temperature $\delta$ is fixed as 0.4 for all main experiments. Following (Song et al., 2022), we adopt warmup for 5 iterations since we utilize model prediction for unlabeled nodes.

## F.6 CONFIGURATION

All the algorithms and models are implemented in Python and PyTorch Geometric. Experiments are conducted on a server with an NVIDIA 3090 GPU (24 GB memory) and an Intel(R) Xeon(R) Silver 4210R CPU @ 2.40GHz.

**Table 14:** Label distributions in the training sets

| Dataset | $\mathcal{C}_0$ | $\mathcal{C}_1$ | $\mathcal{C}_2$ | $\mathcal{C}_3$ | $\mathcal{C}_4$ | $\mathcal{C}_5$ | $\mathcal{C}_6$ | $\mathcal{C}_7$ | $\mathcal{C}_8$ | $\mathcal{C}_9$ |
|---|---|---|---|---|---|---|---|---|---|---|
| Cora ($\rho$=10) | 20 (23.26%) | 20 (23.26%) | 20 (23.26%) | 20 (23.26%) | 2 (23.26%) | 2 (23.26%) | 2 (23.26%) | - | - | - |
| Cora ($\rho$=20) | 20 (24.10%) | 20 (24.10%) | 20 (24.10%) | 20 (24.10%) | 1 (1.19%) | 1 (1.19%) | 1 (1.19%) | - | - | - |
| Cora ($\rho$=50) | 50 (24.63%) | 50 (24.63%) | 50 (24.63%) | 50 (24.63%) | 1 (0.49%) | 1 (0.49%) | 1 (0.49%) | - | - | - |
| Cora ($\rho$=100) | 100 (24.81%) | 100 (24.81%) | 100 (24.81%) | 100 (24.81%) | 1 (0.25%) | 1 (0.25%) | 1 (0.25%) | - | - | - |
| CiteSeer ($\rho$=10) | 20 (30.30%) | 20 (30.30%) | 20 (30.30%) | 2 (30.30%) | 2 (3.03%) | 2 (3.03%) | - | - | - | - |
| CiteSeer ($\rho$=20) | 20 (31.75%) | 20 (31.75%) | 20 (31.75%) | 1 (1.59%) | 1 (1.59%) | 1 (1.59%) | - | - | - | - |
| CiteSeer ($\rho$=50) | 50 (32.68%) | 50 (32.68%) | 50 (32.68%) | 1 (0.65%) | 1 (0.65%) | 1 (0.65%) | - | - | - | - |
| CiteSeer ($\rho$=100) | 100 (33.00%) | 100 (33.00%) | 100 (33.00%) | 1 (0.33%) | 1 (0.33%) | 1 (0.33%) | - | - | - | - |
| PubMed ($\rho$=10) | 20 (47.62%) | 20 (47.62%) | 2 (4.76%) | - | - | - | - | - | - | - |
| PubMed ($\rho$=20) | 20 (48.78%) | 20 (48.78%) | 1 (2.44%) | - | - | - | - | - | - | - |
| PubMed ($\rho$=50) | 50 (49.50%) | 50 (49.50%) | 1 (0.99%) | - | - | - | - | - | - | - |
| PubMed ($\rho$=100) | 100 (49.75%) | 100 (49.75%) | 1 (0.50%) | - | - | - | - | - | - | - |
| Amazon-Computers ($\rho$=10) | 20 (18.18%) | 20 (18.18%) | 20 (18.18%) | 20 (18.18%) | 20 (18.18%) | 2 (1.82%) | 2 (1.82%) | 2 (1.82%) | 2 (1.82%) | 2 (1.82%) |
| Amzon-Computers ($\rho$=20) | 20 (19.05%) | 20 (19.05%) | 20 (19.05%) | 20 (19.05%) | 20 (19.05%) | 1 (0.95%) | 1 (0.95%) | 1 (0.95%) | 1 (0.95%) | 1 (0.95%) |
| Amazon-Computers ($\rho$=50) | 50 (19.61%) | 50 (19.61%) | 50 (19.61%) | 50 (19.61%) | 50 (19.61%) | 1 (0.39%) | 1 (0.39%) | 1 (0.39%) | 1 (0.39%) | 1 (0.39%) |
| Amazon-Computers ($\rho$=100) | 100 (19.80%) | 100 (19.80%) | 100 (19.80%) | 100 (19.80%) | 100 (19.80%) | 1 (0.20%) | 1 (0.20%) | 1 (0.20%) | 1 (0.20%) | 1 (0.20%) |
| Flickr ($\rho \approx 10.80$) | 2628 (5.89%) | 4321 (9.68%) | 3164 (7.09%) | 2431 (5.45%) | 11525 (25.83%) | 1742 (3.90%) | 18814 (42.16%) | - | - | - |

**Table 15:** Label distributions on the whole graphs

| Dataset | $\mathcal{C}_0$ | $\mathcal{C}_1$ | $\mathcal{C}_2$ | $\mathcal{C}_3$ | $\mathcal{C}_4$ | $\mathcal{C}_5$ | $\mathcal{C}_6$ | $\mathcal{C}_7$ | $\mathcal{C}_8$ | $\mathcal{C}_9$ |
|---|---|---|---|---|---|---|---|---|---|---|
| Cora | 351 | 217 | 418 | 818 | 426 | 298 | 180 | - | - | - |
| CiteSeer | 264 | 590 | 668 | 701 | 696 | 508 | - | - | - | - |
| PubMed | 4103 | 7739 | 7835 | - | - | - | - | - | - | - |
| Amazon-Computers | 436 | 2142 | 1414 | 542 | 5158 | 308 | 487 | 818 | 2156 | 291 |
| Flickr | 5264 | 8506 | 6413 | 4903 | 22966 | 3479 | 37719 | - | - | - |

# G  MAIN ALGORITHM

---

**Algorithm 1** *UNREAL*

---

**Input:** Imbalanced dataset $(\mathcal{G} = (\mathcal{V}, \mathcal{E}, \mathcal{L}_0), y)$, feature matrix $\mathcal{X}$, adjacency matrix $\mathcal{A}$, unlabeled set $\mathcal{U} = \mathcal{V} - \mathcal{L}_0$, rounds $T$ to select nodes, the size threshold $\alpha$ of nodes being added in each class per round, weight hyperparameter $p$ of RBO, threshold $\gamma$ of DGI, learning rate $\eta$, the size $k'$ of the clusters, GNN model $f_g$, clustering algorithm $f_{\text{cluster}}$, and the mean function $M(\cdot)$.

1: **for** $i = 0$ to round $T$ **do**
2:     Train $f_g$ based on the current training set $\mathcal{L}_i := \{\mathcal{C}_1, \cdots, \mathcal{C}_k\}$.
3:     Obtain node embedding matrix of the labeled node set and unlabeled node set $H^L \in \mathbb{R}^{|\mathcal{L}| \times d}, H^U \in \mathbb{R}^{|\mathcal{U}| \times d}$, prediction $\hat{y}$ and confidence $r$ from classifier.
4:     **% Step 1: Dual Pseudo-tag Alignment Mechanism(DPAM)**
5:     $f_{\text{cluster}}(H^U) \Longrightarrow \{\mathcal{K}_1, c_1, \mathcal{K}_2, c_2, \cdots, \mathcal{K}_{k'}, c_{k'}\}$
6:     $c_i^{\text{train}} = M(\{h_u^L \mid y_u \in \mathcal{C}_i\})$
7:     Assign a label $\tilde{y}_m$ to each cluster $\mathcal{K}_m$: $\tilde{y}_m = \arg\min_j \text{distance}(c_j^{\text{train}}, c_m)$.
8:     Combine clusters with the same pseudo-label $m$ as $\tilde{\mathcal{U}}_m$, and $\mathcal{U} = \bigcup_{m=1}^{k} \tilde{\mathcal{U}}_m$.
9:     Put unlabeled nodes whose prediction in $\hat{y}$ is $m$ into the set $\mathcal{U}_m$, and $\mathcal{U} = \bigcup_{m=1}^{k} \mathcal{U}_m$.
10:    **% Step 2: Node-reordering**
11:    For each $u \in \tilde{\mathcal{U}}_m \cap \mathcal{U}_m$: $\delta_u = \text{distance }(h_u^L, c_m^{\text{train}})$.
12:    Obtain geometric rankings $\{\mathcal{S}_1, \mathcal{S}_2, \cdots, \mathcal{S}_k\}$ based on $\delta$; and confidence rankings $\{\mathcal{T}_1, \mathcal{T}_2, \cdots, \mathcal{T}_k\}$ based on $r$.
13:    For each $m$, $\mathcal{N}_m^{New} = \max\{r_m, 1 - r_m\} \cdot \mathcal{S}_m + \min\{r_m, 1 - r_m\} \cdot \mathcal{T}_m$.
14:    Select nodes based on the rank of their values in $\mathcal{N}_m^{New}$.
15:    **% Step 3:Discarding geometrically imbalanced nodes (DGI)**
16:    Obtain the distance between the embedding of $u$ and the second closest center to $u$ as $\beta_u$, compute GI index of node $u$ as $\frac{\beta_u - \delta_u}{\delta_u}$.
17:    **if** $\frac{\beta_u - \delta_u}{\delta_u} < \gamma$ **then**
18:        Discard node $u$.
19:    **else**
20:        Select it to training set.
21:    **end if**
22:    Update the training set $\mathcal{L}_i$.
23: **end for**

---

