# OpenReview forum: "UNREAL: Unlabeled Nodes Retrieval and Labeling for Heavily-imbalanced Node Classification"
_ICLR.cc/2023/Conference — Submitted to ICLR 2023_

### Official Review · Reviewer_9Evz · 2022-10-23

**Confidence:** 4
**Correctness:** 3
**Technical Novelty And Significance:** 3
**Empirical Novelty And Significance:** 3
**Recommendation:** 6

**Clarity, Quality, Novelty And Reproducibility:**

Clarity: the paper needs more clarification and modification, especially in Section 3.

Quality and novelty: the proposed method UNREAL is the first method to use unlabeled nodes to deal with class imbalanced node classification. However, I do not think it is very suitable for the target problem.

Reproducibility: the authors provide code for reproducibility.


**Strength And Weaknesses:**

Strength:
1. The paper introduces the target problem in class-imbalanced node classification clearly. The motivation is convincing and strong.
2. The authors use figures to illustrate the pipeline of the proposed approach UNREAL, which helps readers understand the approach.
3. The authors elaborate experiment details in the paper including experimental settings and datasets. It enhances reproducibility of the paper.



Weaknesses:
1. The proposed approach UNREAL does not solve the class-imbalanced problem in the real world. UNREAL iteratively selects unlabeled nodes from the minority classes during training. In practice, however, the class-imbalanced problem in the test set may be also severe. For instance, suppose we have $20n$ normal accounts and $n$ malicious accounts as the training set for malicious account detection. Meanwhile, the number of malicious accounts in the test set may be also limited (e.g., $2n$). In this case, the new training set formed by UNREAL is still class imbalanced. On the contrary, synthetic over-sampling algorithms (e.g., GraphSMOTE) can overcome this problem and may have better performance.
2. In Section 3.2, Node-Reordering sorts nodes in the decreasing order of their confidence. The authors should provide a clear definition of the confidence. However, it is not provided either in Section 3 or in the main algorithm.
3. According to Table 3, using geometric ranking is empirically better than confidence ranking. So why do we need to involve confidence ranking in Node-Reordering instead of solely using geometric ranking? In other words, does the weighted combination of the two rankings really benefit the performance? The authors may answer the concern by explaining in Section 3.2 and conducting experiments in Section 4.
4. The proposed approach UNREAL involves a few hyperparameters such as $k’$ in $f_{\text{cluster}}$ and the threshold $\gamma$ of DGI. The authors should conduct experiments to investigate the sensitivity of UNREAL to hyperparameters.
5. The main algorithm in Appendix C should be consistent with the description of UNREAL in the main paper. In Section 3.1, the unlabeled nodes are clustered into $k’$ clusters and $k’$ is usually larger than $k$. So $k’$ should be an input of the main algorithm and line 5 in the main algorithm is incorrect. In addition, Section 3.4 implies that UNREAL only adds nodes from the minority classes, which does not appear in the main algorithm.
6. Please pay attention to grammar. I found many typos in the paper. For instance, in Related Work, “compensates” should be “compensate”, and “adjust” should be “adjusts”.


**Summary Of The Paper:**

This paper is targeted at class-imbalanced node classification. The authors propose UNREAL, an over-sampling-based approach which iteratively adds unlabeled nodes into the training set. UNREAL consists of the Dual Pseudo-tag Alignment Mechanism (DPAM), Node-Reordering, and Discarding Geometrically Imbalanced nodes (DGI). The authors conduct comprehensive experiments over five datasets and the results demonstrate that the proposed UNREAL outperforms recent baselines with different GNN backbones.

**Summary Of The Review:**

This paper is not good enough for this conference. The paper needs more clarification to enhance the rationale of the proposed method. In addition, the authors need to conduct more experiments to verify the superiority of UNREAL.

---

> ### Author Response · Authors · 2022-11-19
> **The reply of the other comments**
>
> **Q2: The authors should provide a clear definition of the confidence.**
>
> Thank you for the suggestion. We have added the definition of confidence in section 4 in the revision.
> For each node $u$, the classification $\textit{confidence}$ is the prediction probability of the assigned pseudo-label in the softmax classifier:
>
> predictions = Softmax(logits) and confidence = max(predictions)
>
>
>
> **Q3: So why do we need to involve confidence ranking in Node-Reordering instead of solely using geometric ranking?
> In other words, does the weighted combination of the two rankings really benefit the performance?**
>
> Thanks for the question. The confidence ranking is unreliable in early rounds of predictions, since the imbalance label distribution causes bias in the prediction. This is experimentally verified in Section 3 of the revised paper. On the other hand, geometric ranking assigns pseudo-labels using an unsupervised method, which is more immune to label imbalance. However, the classification accuracy of unsupervised clustering is inferior to supervised classifiers. So, as the label distribution of the training set gradually becomes more balanced in later rounds, intuitively we should rely more on confidence ranking (which is from a supervised classifier).
>
> To better quantify the effectiveness of different rankings, we have provided ablation analysis in our submission (Table 3). In the revision, we conduct more experiments on this.
>
>  * **Experimental setup** We conduct experiments to test the accuracy of pseudo-labels for unlabeled nodes on class-imbalanced graphs.
>  We process the two datasets following the method in (Park et al., 2021, Zhao et al., 2021). The imbalance ratio is set to 1, 5, 10, 20, 50, 100. We compare the effectiveness of different rankings, with or without DGI. We report the pseudo-label accuracy of 100 nodes that join the minority class. Here we list the results on Cora, where UNREAL uses GCN as the base model. More experimental results are provided in Appendix D.
>
>
> | Method \ Imbalance ratio    | 1                | 5                | 10               | 20               | 50               | 100              |
> |-----------------------------|------------------|------------------|------------------|------------------|------------------|------------------|
> | DPAM+Confidence Ranking     | 61.40 $\pm$ 2.73 | 62.40 $\pm$ 2.59 | 60.20 $\pm$ 1.02 | 58.40 $\pm$ 1.05 | 57.60 $\pm$ 1.86 | 58.40 $\pm$ 2.15 |
> | DPAM+Geometric Ranking      | 64.00 $\pm$ 3.67 | 61.20 $\pm$ 2.89 | 61.20 $\pm$ 2.54 | 63.60 $\pm$ 1.31 | 55.60 $\pm$ 2.31 | 47.80 $\pm$ 2.87 |
> | DPAM+Node-Reordering        | 89.65 $\pm$ 3.23 | 86.98 $\pm$ 0.21 | 88.32 $\pm$ 0.83 | 85.32 $\pm$ 2.98 | 90.87 $\pm$ 2.31 | 71.60 $\pm$ 2.91 |
> | DPAM+Confidence Ranking+DGI | 71.00 $\pm$ 5.47 | 75.40 $\pm$ 2.15 | 68.20 $\pm$ 1.25 | 69.40 $\pm$ 1.28 | 67.80 $\pm$ 2.75 | 66.60 $\pm$ 0.16 |
> | DPAM+Geometric Ranking+DGI  | 69.60 $\pm$ 3.78 | 73.80 $\pm$ 0.45 | 64.80 $\pm$ 1.26 | 64.20 $\pm$ 1.91 | 57.00 $\pm$ 1.57 | 69.00 $\pm$ 1.71 |
> | DPAM+Node-Reordering+DGI    | 92.80 $\pm$ 1.30 | 96.40 $\pm$ 4.27 | 92.20 $\pm$ 0.85 | 89.40 $\pm$ 1.37 | 93.00 $\pm$ 0.82 | 77.80 $\pm$ 2.50 |
>
>
>
> **Q4: The authors should conduct experiments to investigate the sensitivity of UNREAL to hyperparameters.**
>
> Thank you for your suggestion. We have added the results of hyperparameter sensitivity analysis in Appendix E.
>
>
>
> **Q5: So k' should be an input of the main algorithm and line 5 in the main algorithm is incorrect.**
>
> Thank you for the comment. We have corrected this in the revision.
>
>
>
>
> **Q6:Please pay attention to grammar. I found many typos in the paper. For instance, in Related Work, “compensates” should be “compensate”, and “adjust” should be “adjusts”.**
>
> Thanks for your comments. We have corrected these grammatical errors in the revised version of the paper.

---

> > ### Comment · Reviewer_9Evz · 2022-11-21
> > **Reply to the authors' feedback**
> >
> > As for grammar issues, I mentioned a few in my primary comments. It seems that some other typos are still existing after revising. Please go through the whole paper again and check it out carefully.

---

> > > ### Author Response · Authors · 2022-12-02
> > > **Thanks for your feedback !**
> > >
> > > We have checked the full text  carefully and recorded all typos, and we will correct them in time when we can update the paper.

---

> ### Author Response · Authors · 2022-11-19
> **Thank you for your constructive comments and suggestions!**
>
> Thank you for your constructive comments and suggestions! The response to your comments are listed below :
>
> **Q1: The proposed approach UNREAL does not solve the class-imbalanced problem in the real world.**
>
> Thanks for your comment. This work is mainly focused on semi-supervised setting. In semi-supervised learning, there are lots of unlabelled data available, and thus your question raised here is not of much concern. The reason is that even if the imbalance ratio is extremely high among unlabelled nodes, the number of unlabelled nodes is still enough for our method to work. To verify the applicability of our method in this scenario, here we provide experimental results on two naturally imbalanced datasets, namely Flickr and ogbn-arxiv, in which the imbalance ratio in the training set and in the rest of the graph are both very large. The label distributions of the two datasets are listed below:
>
> Flickr, ratio=10.8
> | Class   | Whole graph | Training set | Testing set |
> |---------------------------|-------------|--------------|-------------|
> | C0                        | 5264        | 2628         | 1354        |
> | C1                        | 8506        | 4321         | 2062        |
> | C2                        | 6413        | 3164         | 1041        |
> | C3                        | 4903        | 2431         | 1247        |
> | C4                        | 22966       | 11525        | 5684        |
> | C5                        | 3479        | 1742         | 877         |
> | C6                        | 37719       | 18814        | 9448        |
>
>
> Ogbn-arxiv,ratio=476
> | Class   | Whole graph | Training set | Testing set | Class                       | Whole graph | Training set | Testing set |
> |-----------------------------|-------------|--------------|-------------|-----------------------------|-------------|--------------|-------------|
> | C0                          | 565         | 437          | 54          | C20                         | 2076        | 1495         | 313         |
> | C1                          | 687         | 382          | 187          | C21                         | 393         | 304          | 51          |
> | C2                          | 4839        | 3604         | 733         | C22                         | 1903        | 1268         | 386         |
> | C3                          | 2080        | 1014         | 654         | C23                         | 2834        | 1539         | 808         |
> | C4                          | 5832        | 2864         | 1869        | C24                         | 22187       | 6989         | 10740       |
> | C5                          | 4958        | 2933         | 1246        | C25                         | 1257        | 457          | 475         |
> | C6                          | 1618        | 703          | 622         | C26                         | 4605        | 2834         | 1041        |
> | C7                          | 589         | 380          | 134         | C27                         | 4801        | 1661         | 2066        |
> | C8                          | 6232        | 4056         | 1250        | C28                         | 21406       | 16284        | 2849        |
> | C9                          | 2820        | 2245         | 345         | C29                         | 416         | 239          | 120         |
> | C10                         | 7869        | 5182         | 1455        | C30                         | 11814       | 4334         | 4631        |
> | C11                         | 750         | 391          | 239         | C31                         | 2828        | 1350         | 892         |
> | C12                         | 79          | 21           | 5           | C32                         | 411         | 270          | 83          |
> | C13                         | 2358        | 1290         | 628         | C33                         | 1271        | 926          | 220         |
> | C14                         | 597         | 433          | 71          | C34                         | 7867        | 5436         | 1414        |
> | C15                         | 403         | 248          | 87          | C35                         | 127         | 25           | 36          |
> | C16                         | 27321       | 9948         | 10471       | C36                         | 3524        | 2506         | 627         |
> | C17                         | 515         | 202          | 203         | C37                         | 2369        | 1615         | 481         |
> | C18                         | 749         | 402          | 209         | C38                         | 1507        | 1100         | 214         |
> | C19                         | 2877        | 1873         | 419         | C39                          | 2009        | 1551         | 269         |

---

> > ### Author Response · Authors · 2022-11-19
> > **Supplement to Question 1 （Due to Space limitations）**
> >
> > The results on F1 scores for the two benchmarks are list in the two tables below
> >
> > | Flickr / F1(\%) | GCN              | GAT              | SAGE             |
> > |--------|------------------|------------------|------------------|
> >  | Vanilla | 24.53 $\pm$ 0.11 | 25.32 $\pm$ 0.44 | 24.16 $\pm$ 0.27 |
> >  | GraphSmote | OOM              | OOM              | OOM              |
> > | GraphENS | OOM              | OOM              | OOM              |
> >  | BalancedSoftmax+TAM      | 23.97 $\pm$ 0.60 | 25.52 $\pm$ 0.89 | 27.56 $\pm$ 0.25 |
> >  | Ours   | 30.60 $\pm$ 0.29 | 28.21 $\pm$ 0.76 | 54.01  $\pm$ 1.34|
> >
> >
> >
> >
> >
> > | Ogbn-arxiv / F1(\%) | GCN              | GAT              | SAGE             |
> > |--------|------------------|------------------|------------------|
> >  | Vanilla | 49.60 $\pm$ 0.14 | 49.23 $\pm$ 0.33 | 49.43 $\pm$ 0.29 |
> >  | GraphSmote | OOM              | OOM              | OOM              |
> > | GraphENS | OOM              | OOM              | OOM              |
> >  | BalancedSoftmax+TAM      | 49.64 $\pm$ 0.12 | 49.50 $\pm$ 0.29 | 49.67 $\pm$ 0.42 |
> >  | Ours   | 49.80 $\pm$ 0.21 | 50.32 $\pm$ 0.12 | 50.14 $\pm$ 0.45 |
> >
> >
> > Clearly, on naturally imbalanced datasets, in which the testing sets are highly imbalanced, our method still outperforms baselines. We note that some existing methods for dealing with imbalanced node classification cannot handle Flicker and Ogbn-arxiv on a NVIDIA 3090 GPU (24 GB memory) due to excessive memory consumption.

---

> > > ### Comment · Reviewer_9Evz · 2022-11-21
> > > **Reply to the authors' feedback**
> > >
> > > Thanks for your detailed feedback. I think you solve most of my questions; therefore, I would love to increase the score to 5.
> > >
> > > I still have a concern that this paper fails to provide the results of some baselines (e.g., GraphSMOTE and GraphENS) over the real-world datasets (i.e., Flickr and Ogbn-arxiv) due to OOM. However, we already have a number of relatively smaller real-world datasets with the class imbalanced issue, such as Coauthor CS ($\rho=35.05$) and Amazon Computers ($\rho=17.73$) [1]. You may involve one or more such datasets and present the results of UNREAL as well as all the baselines, which can make the experiments much more convincing.
> > >
> > > [1] Shchur, Oleksandr, Maximilian Mumme, Aleksandar Bojchevski, and Stephan Günnemann. "Pitfalls of graph neural network evaluation." arXiv preprint arXiv:1811.05868 (2018).

---

> > > > ### Author Response · Authors · 2022-12-02
> > > > **Supplementary to the experimental results of natural imbalanced datasets（Due to Space limitations）**
> > > >
> > > > We report averaged balanced accuracy (bAcc.,%) and F1-score (%) with the standard errors over 5 repetitions on three representative GNN architectures as follows:
> > > >
> > > >
> > > > | Dataset(Computers-random) | Based  on GCN                          | Based on GAT                          | Based on GraphSAGE             |
> > > > |---------------------------|------|------|------|
> > > > | $\rho\approx$17.76        | bAcc                          \| F1                           | bAcc   \|  F1                   | bAcc  \|          F1 |
> > > > | Vanilla                   | 89.96 $\pm$ 0.33 \| 87.33 $\pm$ 0.14             | 88.20 $\pm$ 0.20 \| 87.54 $\pm$ 0.08 |88.04 $\pm$ 0.64  \| 86.76 $\pm$ 0.72 |
> > > > | Re-weight                 | 91.18 $\pm$ 0.05 \| 86.40 $\pm$ 0.42             | 90.48 $\pm$ 0.17 \| 87.82 $\pm$ 0.22 |89.62 $\pm$ 0.25  \| 86.18 $\pm$ 0.53 |
> > > > | PC Softmax                | 91.07 $\pm$ 0.12 \| 85.90 $\pm$ 0.18             | 90.40 $\pm$ 0.11 \| 86.80 $\pm$ 0.44 |90.07 $\pm$ 0.24  \| 85.57 $\pm$ 0.56 |
> > > > | BalancedSoftmax           | 91.27 $\pm$ 0.08 \| 86.26 $\pm$ 0.30             | 90.36 $\pm$ 0.16 \| 86.89 $\pm$ 0.30 |90.02 $\pm$ 0.22  \| 85.86 $\pm$ 0.47 |
> > > > | GraphSMOTE                | 90.90 $\pm$ 1.11 \| 85.61 $\pm$ 0.14             | 89.87 $\pm$ 0.21 \| 86.76 $\pm$ 0.31 |89.43 $\pm$ 0.67  \| 85.43 $\pm$ 1.01 |
> > > > | Renode                    | 91.03 $\pm$ 0.09 \| 86.92 $\pm$ 0.27             | 90.51 $\pm$ 0.20 \| $\underline{87.94 \pm 0.25}$ |90.25 $\pm$ 0.16  \| $\underline{86.84 \pm 0.40}$ |
> > > > | GraphENS                  | 90.58 $\pm$ 0.10 \| 85.16 $\pm$ 0.18             | 90.71 $\pm$ 0.11 \| 85.91 $\pm$ 0.40 |90.08 $\pm$ 0.30  \| 84.35 $\pm$ 0.41 |
> > > > | BalancedSoftmax+TAM       | $\underline{91.46 \pm 0.07}$ \| $\underline{87.49 \pm 0.24}$ | $\underline{90.83 \pm 0.12}$ \| 87.33 $\pm$ 0.29 |90.39 $\pm$ 0.07  \| 85.75 $\pm$ 0.14 |
> > > > | GraphENS+TAM              | 90.66 $\pm$ 0.11 \| 86.40 $\pm$ 0.17             | 90.68 $\pm$ 0.15                 \| 86.02 $\pm$ 0.19 |$\underline{90.43 \pm 0.17}$  \| 85.03 $\pm$ 0.21 |
> > > > | $\textbf{UNREAL}$         | $\textbf{92.76 $\pm$ 0.38}$ \| $\textbf{88.89 $\pm$ 0.32}$  | $\textbf{92.07 $\pm$ 0.29}$ \| $\textbf{89.24 $\pm$ 0.08}$ |$\textbf{92.04 $\pm$ 0.31}$  \| $\textbf{88.76 $\pm$ 0.43}$ |
> > > >
> > > >
> > > > | Dataset(CS-random)  | Based on GCN                             | Based on GAT                       | Based on GraphSAGE        |
> > > > |-------|------|------|----|
> > > > | $\rho\approx$35.05  | bAcc \| F1      | bAcc\|  F1  | bAcc \| F1 |
> > > > | Vanilla             | 89.73 $\pm$ 0.02              \| 90.76 $\pm$ 0.03            | 88.37 $\pm$ 0.13 \| 89.10 $\pm$ 0.15 |89.71 $\pm$ 0.12  \| 90.83 $\pm$ 0.05 |
> > > > | Re-weight           | 90.41 $\pm$ 0.07 \| 90.78 $\pm$ 0.07            | 89.00 $\pm$ 0.19 \| 89.10 $\pm$ 0.17 | 90.54 $\pm$ 0.08  \| 91.21 $\pm$ 0.05 |
> > > > | PC Softmax          | $\underline{90.91 \pm 0.05}$ \| 90.64 $\pm$ 0.06            | 89.07 $\pm$ 0.07 \| 88.80 $\pm$ 0.16 |90.44 $\pm$ 0.20  \| 90.56 $\pm$ 0.36 |
> > > > | BalancedSoftmax     | 90.72 $\pm$ 0.10 \| 90.54 $\pm$ 0.12            | 89.61 $\pm$ 0.29 \| 89.16 $\pm$ 0.29 |90.83 $\pm$ 0.12  \| 91.07 $\pm$ 0.11 |
> > > > | GraphSMOTE          | 90.30 $\pm$ 0.31 \| 90.61 $\pm$ 0.65            | 89.53 $\pm$ 0.61 \| 88.96 $\pm$ 0.48 |90.03 $\pm$ 0.65  \| 90.76 $\pm$ 0.78 |
> > > > | Renode              | 90.03 $\pm$ 0.21 \| 90.32 $\pm$ 0.23            | 88.51 $\pm$ 0.80 \| 88.54 $\pm$ 0.65 |90.25 $\pm$ 0.56  \| 90.14 $\pm$ 0.32 |
> > > > | GraphENS            | 89.55 $\pm$ 0.06 \| 89.42 $\pm$ 0.06            | 89.95 $\pm$ 0.31 \| 89.58 $\pm$ 0.20 |90.14 $\pm$ 0.26  \| 90.10 $\pm$ 0.28 |
> > > > | BalancedSoftmax+TAM | 90.97 $\pm$ 0.12 \| $\underline{90.89 \pm 0.10}$              | 89.86 $\pm$ 0.25  \| 89.36 $\pm$ 0.26 |$\underline{90.96  \pm 0.08}$  \| $\underline{91.24 \pm 0.08}$ |
> > > > | GraphENS+TAM        | 89.53 $\pm$ 0.03 \| 89.60 $\pm$ 0.04           |$\underline{90.02 \pm 0.25}$ \|$ \underline{89.80 \pm 0.20}$|90.33 $\pm$ 0.09  \| 90.39 $\pm$ 0.17 |
> > > > | $\textbf{UNREAL}$   | $\textbf{91.53 $\pm$ 0.43}$ \| $\textbf{91.42 $\pm$ 0.32}$ | $\textbf{90.89 $\pm$ 0.29}$ \| $\textbf{90.64 $\pm$ 0.65}$ |$\textbf{92.43 $\pm$ 0.67}$  \| $\textbf{92.76 $\pm$ 0.68}$ |
> > > >
> > > >
> > > >
> > > > We can observe that UNREAL still achieves state-of-the-art on naturally imbalanced datasets (both training and testing sets are imbalanced).

---

> > > > > ### Comment · Reviewer_9Evz · 2022-12-05
> > > > > **Reply to the authors' feedback**
> > > > >
> > > > > Thanks for providing experiment results over two extra datasets. I am glad to increase the score to 6. Please consider including the new results in the paper or supplementary materials.

---

> > > > ### Author Response · Authors · 2022-12-02
> > > > **Thank you for your constructive comments and suggestions!**
> > > >
> > > > Thank you for your concern about the performance of UNREAL on naturally imbalanced datasets. In order to verify the performance of UNREAL and all baselines on natural imbalanced datasets better, we conduct more experiments on Coauthor-CS($\rho$=35.05) and Amazon-Computers($\rho$=17.73),
> > > > we construct the imbalanced training set by random sampling, and repeat the experiment five times. The label distributions of the training set, testing set and the whole graph are as follows:
> > > >
> > > > | Class/Node size（Coauthor-CS） | C0  | C1  | C2    |C3  | C4  | C5  | C6 | C7 | C8  | C9   | C10  | C11| C12| C13| C14 |
> > > > |------------------------------|-----|-----|-------|----|-------|----|---- |----|-----|------|------|----|----|----|----|
> > > > | The whole graph              | 708 | 462 | 2050  |429| 1394 |2193|371| 924| 775 | 118  | 1444 | 2033| 420| 4136| 876|
> > > > | Training set                 | 70  | 46  | 205   | 42 | 139  | 219 | 37 | 92   | 77    |  11   | 144     | 203   | 42   | 413   | 87   |
> > > > | Testing set                  | 608 | 386 | 1815  | 357 |1225|  1944 | 304 | 802   | 668    |  77  | 1270    | 1800   | 348    |  3693   | 759   |
> > > >
> > > >
> > > > | Class/Node size（Amazon-Computers） | C0  | C1  | C2   | C3   | C4  | C5  | C6 | C7 | C8  | C9  |
> > > > |-----------------------------------|-----|-----|------|------|-------|----|---- |----|-----|-----|
> > > > | The whole graph                   | 436 | 2142 | 1414 | 542  | 5158 |308 |487 | 818 | 2156 | 291 |
> > > > | Training set                      | 43  | 214  | 141  | 54   | 515  | 30 | 48 | 81  | 215    | 29  |
> > > > | Testing set                       | 363 | 1898 | 1243 | 458  |  4613 |248 |409   | 707    | 1911  | 232|

---

### Official Review · Reviewer_D8cW · 2022-10-25

**Confidence:** 5
**Correctness:** 3
**Technical Novelty And Significance:** 3
**Empirical Novelty And Significance:** 3
**Recommendation:** 6

**Clarity, Quality, Novelty And Reproducibility:**

* For clarity, the paper is well-written and easy-to-follow.
* The proposed idea is novel with significant improvements over baseline methods, but the technical quality could be improved by conducting more comprehensive experiments and analysis.
* The paper provides some implementation details but does not mention if they will release the implementation.


**Strength And Weaknesses:**

Strengths
* Interesting and novel idea of DPAM and reranking for node retrieval.
* Surprisingly good accuracy. The improvements over baseline methods are significant, especially for the cases with higher imbalance ratios.
* The paper is well-written and easy-to-follow.

Weaknesses
* Lack of experiments with large-scale graphs, such as OGB benchmarks, where the imbalance could naturally exist.
* Some analysis could be more extensive. For example, I wonder why for each dataset the authors only report the analysis for one imbalance ratio (and different across datasets). The space is actually enough to show all if using more columns.
* Some concrete examples and qualitative analysis could be valuable to validate the motivations and justify the high improvements over baseline methods.


**Summary Of The Paper:**

In this paper, the authors propose an over-sampling based approach to tackle the situation of extremely skewed label distributions. Instead of generating synthetic minority nodes, as well as features and local topology, the authors consider unlabeled nodes, thereby eliminating the challenge of generating synthetic features and neighborhoods. Specifically, they conduct geometric ranking based on node embedding. The experiments conducted on benchmark datasets demonstrate that the proposed method significantly outperforms several baseline methods with different imbalance ratios.

**Summary Of The Review:**

In sum, I would recommend “6: marginally above the acceptance threshold”. The proposed idea is novel with significant gains, but it would be great to have more serious validation and analysis on concrete examples and large-scale datasets.

---

> ### Author Response · Authors · 2022-11-19
> **Thank you for your constructive comments and suggestions!**
>
> Thank you for your constructive comments and suggestions! The response to your comments are listed below :
>
> **Q1: Lack of experiments with large-scale graphs, such as OGB benchmarks, where the imbalance could naturally exist.**
>
> Thank you for the comment. In the paper, we have verified the effectiveness of UNREAL on Flicker, in which imbalance naturally exist (with imbalance ratio=10.8). The results are listed below:
> | Method / F1(\%) | GCN              | GAT              | SAGE             |
> |--------|------------------|------------------|------------------|
>  | Vanilla | 24.53 $\pm$ 0.11 | 25.32 $\pm$ 0.44 | 24.16 $\pm$ 0.27 |
>  | PC Softmax| 25.81  $\pm$ 1.75| 27.24 $\pm$ 0.37 | 21.08 $\pm$ 1.73 |
>  | GraphSmote | OOM              | OOM              | OOM              |
> | GraphENS | OOM              | OOM              | OOM              |
>  | BalancedSoftmax+TAM      | 23.97 $\pm$ 0.60 | 25.52 $\pm$ 0.89 | 27.56 $\pm$ 0.25 |
>  | Ours   | 30.60 $\pm$ 0.29 | 28.21 $\pm$ 0.76 | 54.01  $\pm$ 1.34|
>
>
> Here we provides the performance of UNREAL on Ogbn-arxiv, which is also a naturally imbalanced dataset (with imbalance ratio=476). Our model still achieves improved performance compared to baselines.
>
> | Method / F1(\%) | GCN              | GAT              | SAGE             |
> |--------|------------------|------------------|------------------|
>  | Vanilla | 49.60 $\pm$ 0.14 | 49.23 $\pm$ 0.33 | 49.43 $\pm$ 0.29 |
>  | GraphSmote | OOM              | OOM              | OOM              |
> | GraphENS | OOM              | OOM              | OOM              |
>  | BalancedSoftmax+TAM      | 49.64 $\pm$ 0.12 | 49.50 $\pm$ 0.29 | 49.67 $\pm$ 0.42 |
>  | Ours   | 49.80 $\pm$ 0.21 | 50.32 $\pm$ 0.12 | 50.14 $\pm$ 0.45 |
>
> We note that some existing methods for dealing with imbalanced node classification cannot handle Flicker and Ogbn-arxiv on a NVIDIA 3090 GPU (24 GB memory) due to excessive memory consumption.
>
> **Q2: Some analysis could be more extensive. For example, I wonder why for each dataset the authors only report the analysis for one imbalance ratio (and different across datasets).
> The space is actually enough to show all if using more columns.**
>
>
> Thanks for your comments. In our original submission, we have reported results for imbalance ratio 10, 20, 50, 100 on all datasets and all three base models. However, for the ablation study (table 3), we indeed only report the analysis for one imbalance ratio (and different across datasets).
> Here, we report the ablation results for different imbalance ratios on Cora with base model GCN.  As shown in the Table , each component of our method can bring performance improvements. In particular, in three out of four settings in the table, Node-reordering+DGI achieves best F1 scores.
> In all cases, using geometric ranking is better than confidence ranking, which empirically verifies our hypothesis  that the prediction confidence scores might contain bias and be less reliable.
>
> | Cora(GCN) / Imbalance ratio   | 10               | 20               | 50               | 100              |
> |----|------------------|------------------|------------------|------------------|
> | DPAM+Confidence Ranking     | 73.93 $\pm$ 0.95 | 67.40 $\pm$ 2.59 | 66.20 $\pm$ 0.32 | 63.23 $\pm$ 4.08 |
> | DPAM+Geometric Ranking      | 75.85 $\pm$ 0.82 | 70.20 $\pm$ 2.89 | 69.32 $\pm$ 1.23 | 65.60 $\pm$ 1.32 |
> | DPAM+Node-Reordering        | 75.00 $\pm$ 0.97 | 73.21 $\pm$ 0.97 | 70.32 $\pm$ 0.43 | 67.32 $\pm$ 0.32 |
> | DPAM+Confidence Ranking+DGI | 72.74 $\pm$ 0.63 | 69.40 $\pm$ 2.15 | 68.32 $\pm$ 0.75 | 64.57 $\pm$ 0.65 |
> | DPAM+Geometric Ranking+DGI  | 75.00 $\pm$ 0.97 | 71.32 $\pm$ 0.64 | 70.21 $\pm$ 0.13 | 66.98 $\pm$ 0.13 |
> | DPAM+Node-Reordering+DGI    | 76.44 $\pm$ 1.06 | 74.15 $\pm$ 0.65 | 72.59 $\pm$ 2.13 | 69.12 $\pm$ 3.45 |
>
>
>
>
> **Q3: Some concrete examples and qualitative analysis could be valuable to validate the motivations and justify the high improvements over baseline methods,
> or more comprehensive experiments and analysis.**
>
> Thanks for your comments. In the revision, we have added more refined experimental results. A more comprehensive comparison between standard self-training method and UNREAL is added, which shows the ineffectiveness of self-training in handling imbalanced node classification. More ablation analysis on different components are provided. Please see Appendix B and Appendix D of the revised paper for the details.
>
>
>
>
> **Q4: The paper provides some implementation details but does not mention if they will release the implementation.**
>
> The source code was provided in the supplementary material in our original submission.

---

> ### Author Response · Authors · 2022-12-02
> **Supplementary to the experimental results of natural imbalanced datasets（Due to Space limitations）**
>
> We report averaged balanced accuracy (bAcc.,%) and F1-score (%) with the standard errors over 5 repetitions on three representative GNN architectures as follows:
>
>
> | Dataset(Computers-random) | Based  on GCN                          | Based on GAT                          | Based on GraphSAGE             |
> |---------------------------|------|------|------|
> | $\rho\approx$17.76        | bAcc                          \| F1                           | bAcc   \|  F1                   | bAcc  \|          F1 |
> | Vanilla                   | 89.96 $\pm$ 0.33 \| 87.33 $\pm$ 0.14             | 88.20 $\pm$ 0.20 \| 87.54 $\pm$ 0.08 |88.04 $\pm$ 0.64  \| 86.76 $\pm$ 0.72 |
> | Re-weight                 | 91.18 $\pm$ 0.05 \| 86.40 $\pm$ 0.42             | 90.48 $\pm$ 0.17 \| 87.82 $\pm$ 0.22 |89.62 $\pm$ 0.25  \| 86.18 $\pm$ 0.53 |
> | PC Softmax                | 91.07 $\pm$ 0.12 \| 85.90 $\pm$ 0.18             | 90.40 $\pm$ 0.11 \| 86.80 $\pm$ 0.44 |90.07 $\pm$ 0.24  \| 85.57 $\pm$ 0.56 |
> | BalancedSoftmax           | 91.27 $\pm$ 0.08 \| 86.26 $\pm$ 0.30             | 90.36 $\pm$ 0.16 \| 86.89 $\pm$ 0.30 |90.02 $\pm$ 0.22  \| 85.86 $\pm$ 0.47 |
> | GraphSMOTE                | 90.90 $\pm$ 1.11 \| 85.61 $\pm$ 0.14             | 89.87 $\pm$ 0.21 \| 86.76 $\pm$ 0.31 |89.43 $\pm$ 0.67  \| 85.43 $\pm$ 1.01 |
> | Renode                    | 91.03 $\pm$ 0.09 \| 86.92 $\pm$ 0.27             | 90.51 $\pm$ 0.20 \| $\underline{87.94 \pm 0.25}$ |90.25 $\pm$ 0.16  \| $\underline{86.84 \pm 0.40}$ |
> | GraphENS                  | 90.58 $\pm$ 0.10 \| 85.16 $\pm$ 0.18             | 90.71 $\pm$ 0.11 \| 85.91 $\pm$ 0.40 |90.08 $\pm$ 0.30  \| 84.35 $\pm$ 0.41 |
> | BalancedSoftmax+TAM       | $\underline{91.46 \pm 0.07}$ \| $\underline{87.49 \pm 0.24}$ | $\underline{90.83 \pm 0.12}$ \| 87.33 $\pm$ 0.29 |90.39 $\pm$ 0.07  \| 85.75 $\pm$ 0.14 |
> | GraphENS+TAM              | 90.66 $\pm$ 0.11 \| 86.40 $\pm$ 0.17             | 90.68 $\pm$ 0.15                 \| 86.02 $\pm$ 0.19 |$\underline{90.43 \pm 0.17}$  \| 85.03 $\pm$ 0.21 |
> | $\textbf{UNREAL}$         | $\textbf{92.76 $\pm$ 0.38}$ \| $\textbf{88.89 $\pm$ 0.32}$  | $\textbf{92.07 $\pm$ 0.29}$ \| $\textbf{89.24 $\pm$ 0.08}$ |$\textbf{92.04 $\pm$ 0.31}$  \| $\textbf{88.76 $\pm$ 0.43}$ |
>
>
> | Dataset(CS-random)  | Based on GCN                             | Based on GAT                       | Based on GraphSAGE        |
> |-------|------|------|----|
> | $\rho\approx$35.05  | bAcc \| F1      | bAcc\|  F1  | bAcc \| F1 |
> | Vanilla             | 89.73 $\pm$ 0.02              \| 90.76 $\pm$ 0.03            | 88.37 $\pm$ 0.13 \| 89.10 $\pm$ 0.15 |89.71 $\pm$ 0.12  \| 90.83 $\pm$ 0.05 |
> | Re-weight           | 90.41 $\pm$ 0.07 \| 90.78 $\pm$ 0.07            | 89.00 $\pm$ 0.19 \| 89.10 $\pm$ 0.17 | 90.54 $\pm$ 0.08  \| 91.21 $\pm$ 0.05 |
> | PC Softmax          | $\underline{90.91 \pm 0.05}$ \| 90.64 $\pm$ 0.06            | 89.07 $\pm$ 0.07 \| 88.80 $\pm$ 0.16 |90.44 $\pm$ 0.20  \| 90.56 $\pm$ 0.36 |
> | BalancedSoftmax     | 90.72 $\pm$ 0.10 \| 90.54 $\pm$ 0.12            | 89.61 $\pm$ 0.29 \| 89.16 $\pm$ 0.29 |90.83 $\pm$ 0.12  \| 91.07 $\pm$ 0.11 |
> | GraphSMOTE          | 90.30 $\pm$ 0.31 \| 90.61 $\pm$ 0.65            | 89.53 $\pm$ 0.61 \| 88.96 $\pm$ 0.48 |90.03 $\pm$ 0.65  \| 90.76 $\pm$ 0.78 |
> | Renode              | 90.03 $\pm$ 0.21 \| 90.32 $\pm$ 0.23            | 88.51 $\pm$ 0.80 \| 88.54 $\pm$ 0.65 |90.25 $\pm$ 0.56  \| 90.14 $\pm$ 0.32 |
> | GraphENS            | 89.55 $\pm$ 0.06 \| 89.42 $\pm$ 0.06            | 89.95 $\pm$ 0.31 \| 89.58 $\pm$ 0.20 |90.14 $\pm$ 0.26  \| 90.10 $\pm$ 0.28 |
> | BalancedSoftmax+TAM | 90.97 $\pm$ 0.12 \| $\underline{90.89 \pm 0.10}$              | 89.86 $\pm$ 0.25  \| 89.36 $\pm$ 0.26 |$\underline{90.96  \pm 0.08}$  \| $\underline{91.24 \pm 0.08}$ |
> | GraphENS+TAM        | 89.53 $\pm$ 0.03 \| 89.60 $\pm$ 0.04           |$\underline{90.02 \pm 0.25}$ \|$ \underline{89.80 \pm 0.20}$|90.33 $\pm$ 0.09  \| 90.39 $\pm$ 0.17 |
> | $\textbf{UNREAL}$   | $\textbf{91.53 $\pm$ 0.43}$ \| $\textbf{91.42 $\pm$ 0.32}$ | $\textbf{90.89 $\pm$ 0.29}$ \| $\textbf{90.64 $\pm$ 0.65}$ |$\textbf{92.43 $\pm$ 0.67}$  \| $\textbf{92.76 $\pm$ 0.68}$ |
>
>
>
> We can observe that UNREAL still achieves state-of-the-art on naturally imbalanced datasets (both training and testing sets are imbalanced).

---

> ### Author Response · Authors · 2022-12-02
> **More experiments about UNREAL and other baselines on naturally imbalanced datasets**
>
> In order to verify the performance of UNREAL and all baselines on natural imbalanced datasets better, we conduct more experiments on Coauthor-CS($\rho$=35.05) and Amazon-Computers($\rho$=17.73). Since we can't update the manuscript now，we would like to present here. We construct the imbalanced training set by random sampling, and repeat the experiment five times. The label distributions of the training set, testing set and the whole graph are as follows:
>
> | Class/Node size（Coauthor-CS） | C0  | C1  | C2    |C3  | C4  | C5  | C6 | C7 | C8  | C9   | C10  | C11| C12| C13| C14 |
> |------------------------------|-----|-----|-------|----|-------|----|---- |----|-----|------|------|----|----|----|----|
> | The whole graph              | 708 | 462 | 2050  |429| 1394 |2193|371| 924| 775 | 118  | 1444 | 2033| 420| 4136| 876|
> | Training set                 | 70  | 46  | 205   | 42 | 139  | 219 | 37 | 92   | 77    |  11   | 144     | 203   | 42   | 413   | 87   |
> | Testing set                  | 608 | 386 | 1815  | 357 |1225|  1944 | 304 | 802   | 668    |  77  | 1270    | 1800   | 348    |  3693   | 759   |
>
>
> | Class/Node size（Amazon-Computers） | C0  | C1  | C2   | C3   | C4  | C5  | C6 | C7 | C8  | C9  |
> |-----------------------------------|-----|-----|------|------|-------|----|---- |----|-----|-----|
> | The whole graph                   | 436 | 2142 | 1414 | 542  | 5158 |308 |487 | 818 | 2156 | 291 |
> | Training set                      | 43  | 214  | 141  | 54   | 515  | 30 | 48 | 81  | 215    | 29  |
> | Testing set                       | 363 | 1898 | 1243 | 458  |  4613 |248 |409   | 707    | 1911  | 232|

---

### Official Review · Reviewer_fiXK · 2022-10-27

**Confidence:** 4
**Correctness:** 3
**Technical Novelty And Significance:** 2
**Empirical Novelty And Significance:** 2
**Recommendation:** 5

**Clarity, Quality, Novelty And Reproducibility:**

This work does not provide source code for result reproducibility. The paper write-up could be improved. Experiments may need further clarification and explanations.

**Strength And Weaknesses:**

Pros:

1.	Some recent baselines are considered as compared methods in evaluation.
2.	The results of the main experiment also show the effectiveness of the model.
3.	The author's ablation experiments are conducted by arranging and combining "Confidence ranking", "Geometric ranking", "Node reordering" and "DGI" four modules on different datasets and encoders.
4.	Experiments in Sec 4.2 show the ability of the new model to handle imbalanced node classification problem under different ratios.

Cons:

1. The author states that "As far as we know, UNREAL is the first method to use unlabelled nodes rather than synthetic ones in over-sampling approaches to deal with class imbalanced node classification." in Section 1.  However, we believe that this is a method of learning pseudo labels for graph node classification tasks. There have been many efforts working on this topic, and thus the contribution of "using unlabelled nodes rather than synthetic ones." may be over-claimed.
2. As Section 3.3 states, "Identify the problem of geometric node imbalance (GI) and define a new metric to measure GI." Does this dimension has been validated through experimental results or theoretical analysis? I see that the author just referred to a previous article. A meaningful experiment and the theoretical foundation of the "metric" created for this problem are also lacking. Overall, this section may not be solid enough.


3. It is suggested that the author should improve the paper organization to facilitate paper reading. For example, "labelled nodes" and "aggregation function" are in Section 2.1. In addition, there are too many notations in the article. It is recommended to organize a notation table.
4. We did not find the author's source code and experimental dataset link in the paper for result reproducibility.


**Summary Of The Paper:**

To address imbalanced label distributions in node classification tasks, the author suggested an over-sampling-based technique. To obtain unlabelled nodes rather than artificial ones, it uses unsupervised approaches in the embedding space. Additionally, new samples are simultaneously chosen using geometric confidence rankings.

**Summary Of The Review:**

Model effectiveness validated in this work when competing with recent baselines. However, the contribution of this work may be over-claimed. Results of some experiments need to be further clarified. Some key points have not been validated.

---

> ### Author Response · Authors · 2022-11-19
> **Thank you for your constructive comments and suggestions!**
>
> **Q1：There have been many efforts working on this topic, and thus the contribution of "using unlabelled nodes rather than synthetic ones." may be over-claimed.**
>
> Yes，there are quite a lot previous works, which use unlabeled information (i.e., self-training) to enhance performance and robustness of GNNs, and we have also emphasized this in our original submission. Here we just want to point out that, in the imbalanced node classification literature, currently there is no such effort, and our work is the first such attempt on this problem. More importantly, standard self-training (ST) along doesn't perform very well on this task. In this work, we proposed several techniques to enhance the performance, which are the main technical contributions of this work. We are sorry that our statement here may cause some misunderstanding and we have adjusted accordingly.
>
> We have conducted extensive experiments to demonstrate the inadequacy of ST. In the revision, we provided additional experiments and analyses on the comparison between ST and our method, which show that if the original training set is highly imbalanced, the pseudo-labels predicted in standard ST are very unreliable, especially in the early stages. Thus, the pseudo-labels selected based on the predictions could be too noisy. Please see Section 3 and Appendix B for more details on this.
>
>
>
> **Q2: Does Geometric Imbalance has been validated through experimental results or theoretical analysis? A meaningful experiment and the theoretical foundation of the "metric" created for this problem are also lacking. Overall, this section may not be solid enough.**
>
> * Thanks for the comment; we have provided additional ablation and qualitative analyses in the revision. In Node-reordering, we select nodes which are closest to the class centers in the embedding space. When two class centers are close in the embedding space, the predictions of nodes on the boundary may have low quality. But they could be very close to either centers and thus may rank high in node-reordering. DGI aims to eliminate such nodes. We added a visualization using T-SNE on this issue in the revision (Figure 9 in Appendix D).
>
> * We conduct additional experiments to verify the effectiveness of DGI. In particular, we conduct experiments to investigate how the accuracy of pseudo labels is improved by DGI. We record the accuracy of pseudo labels of unlabeled nodes which are newly added to the training set. We report the accuracy of 100 nodes that newly join the minority class. Below, we show the results on Cora (with GCN as the base model), and more experimental analysis and results are provided in Appendix D. We can see, after using DGI, the accuracy of pseudo-labels for increased significantly, which means that more high-quality nodes are added to the training set.
>
> | Method / Imbalance ratio    | 1                | 5                | 10               | 20               | 50               | 100              |
> |-----------------------------|------------------|------------------|------------------|------------------|------------------|------------------|
> | DPAM+Confidence Ranking     | 61.40 $\pm$ 2.73 | 62.40 $\pm$ 2.59 | 60.20 $\pm$ 1.02 | 58.40 $\pm$ 1.05 | 57.60 $\pm$ 1.86 | 58.40 $\pm$ 2.15 |
> | DPAM+Geometric Ranking      | 64.00 $\pm$ 3.67 | 61.20 $\pm$ 2.89 | 61.20 $\pm$ 2.54 | 63.60 $\pm$ 1.31 | 55.60 $\pm$ 2.31 | 47.80 $\pm$ 2.87 |
> | DPAM+Node-Reordering        | 89.65 $\pm$ 3.23 | 86.98 $\pm$ 0.21 | 88.32 $\pm$ 0.83 | 85.32 $\pm$ 2.98 | 90.87 $\pm$ 2.31 | 71.60 $\pm$ 2.91 |
> | DPAM+Confidence Ranking+DGI | 71.00 $\pm$ 5.47 | 75.40 $\pm$ 2.15 | 68.20 $\pm$ 1.25 | 69.40 $\pm$ 1.28 | 67.80 $\pm$ 2.75 | 66.60 $\pm$ 0.16 |
> | DPAM+Geometric Ranking+DGI  | 69.60 $\pm$ 3.78 | 73.80 $\pm$ 0.45 | 64.80 $\pm$ 1.26 | 64.20 $\pm$ 1.91 | 57.00 $\pm$ 1.57 | 69.00 $\pm$ 1.71 |
> | DPAM+Node-Reordering+DGI    | 92.80 $\pm$ 1.30 | 96.40 $\pm$ 4.27 | 92.20 $\pm$ 0.85 | 89.40 $\pm$ 1.37 | 93.00 $\pm$ 0.82 | 77.80 $\pm$ 2.50 |
>
>
>
>
> **Q3: It is suggested that the author should improve the paper organization to facilitate paper reading. For example, "labelled nodes" and "aggregation function" are in Section 2.1.
> In addition, there are too many notations in the article. It is recommended to organize a notation table.**
>
> Thanks for your suggestion. We revised the paper carefully and provided a summary notation in Appendix A.
>
>
>
>
>
> **Q4: We did not find the author's source code and experimental dataset link in the paper for result reproducibility.**
>
> The source code was provided in the supplementary material in our original submission.

---

> ### Author Response · Authors · 2022-12-02
> **Supplementary to the experimental results of natural imbalanced datasets（Due to Space limitations）**
>
> We report averaged balanced accuracy (bAcc.,%) and F1-score (%) with the standard errors over 5 repetitions on three representative GNN architectures as follows:
>
>
> | Dataset(Computers-random) | Based  on GCN                          | Based on GAT                          | Based on GraphSAGE             |
> |---------------------------|------|------|------|
> | $\rho\approx$17.76        | bAcc                          \| F1                           | bAcc   \|  F1                   | bAcc  \|          F1 |
> | Vanilla                   | 89.96 $\pm$ 0.33 \| 87.33 $\pm$ 0.14             | 88.20 $\pm$ 0.20 \| 87.54 $\pm$ 0.08 |88.04 $\pm$ 0.64  \| 86.76 $\pm$ 0.72 |
> | Re-weight                 | 91.18 $\pm$ 0.05 \| 86.40 $\pm$ 0.42             | 90.48 $\pm$ 0.17 \| 87.82 $\pm$ 0.22 |89.62 $\pm$ 0.25  \| 86.18 $\pm$ 0.53 |
> | PC Softmax                | 91.07 $\pm$ 0.12 \| 85.90 $\pm$ 0.18             | 90.40 $\pm$ 0.11 \| 86.80 $\pm$ 0.44 |90.07 $\pm$ 0.24  \| 85.57 $\pm$ 0.56 |
> | BalancedSoftmax           | 91.27 $\pm$ 0.08 \| 86.26 $\pm$ 0.30             | 90.36 $\pm$ 0.16 \| 86.89 $\pm$ 0.30 |90.02 $\pm$ 0.22  \| 85.86 $\pm$ 0.47 |
> | GraphSMOTE                | 90.90 $\pm$ 1.11 \| 85.61 $\pm$ 0.14             | 89.87 $\pm$ 0.21 \| 86.76 $\pm$ 0.31 |89.43 $\pm$ 0.67  \| 85.43 $\pm$ 1.01 |
> | Renode                    | 91.03 $\pm$ 0.09 \| 86.92 $\pm$ 0.27             | 90.51 $\pm$ 0.20 \| $\underline{87.94 \pm 0.25}$ |90.25 $\pm$ 0.16  \| $\underline{86.84 \pm 0.40}$ |
> | GraphENS                  | 90.58 $\pm$ 0.10 \| 85.16 $\pm$ 0.18             | 90.71 $\pm$ 0.11 \| 85.91 $\pm$ 0.40 |90.08 $\pm$ 0.30  \| 84.35 $\pm$ 0.41 |
> | BalancedSoftmax+TAM       | $\underline{91.46 \pm 0.07}$ \| $\underline{87.49 \pm 0.24}$ | $\underline{90.83 \pm 0.12}$ \| 87.33 $\pm$ 0.29 |90.39 $\pm$ 0.07  \| 85.75 $\pm$ 0.14 |
> | GraphENS+TAM              | 90.66 $\pm$ 0.11 \| 86.40 $\pm$ 0.17             | 90.68 $\pm$ 0.15                 \| 86.02 $\pm$ 0.19 |$\underline{90.43 \pm 0.17}$  \| 85.03 $\pm$ 0.21 |
> | $\textbf{UNREAL}$         | $\textbf{92.76 $\pm$ 0.38}$ \| $\textbf{88.89 $\pm$ 0.32}$  | $\textbf{92.07 $\pm$ 0.29}$ \| $\textbf{89.24 $\pm$ 0.08}$ |$\textbf{92.04 $\pm$ 0.32}$  \| $\textbf{88.76 $\pm$ 0.43}$ |
>
>
> | Dataset(CS-random)  | Based on GCN                             | Based on GAT                       | Based on GraphSAGE        |
> |-------|------|------|----|
> | $\rho\approx$35.05  | bAcc \| F1      | bAcc\|  F1  | bAcc \| F1 |
> | Vanilla             | 89.73 $\pm$ 0.02              \| 90.76 $\pm$ 0.03            | 88.37 $\pm$ 0.13 \| 89.10 $\pm$ 0.15 |89.71 $\pm$ 0.12  \| 90.83 $\pm$ 0.05 |
> | Re-weight           | 90.41 $\pm$ 0.07 \| 90.78 $\pm$ 0.07            | 89.00 $\pm$ 0.19 \| 89.10 $\pm$ 0.17 | 90.54 $\pm$ 0.08  \| 91.21 $\pm$ 0.05 |
> | PC Softmax          | $\underline{90.91 \pm 0.05}$ \| 90.64 $\pm$ 0.06            | 89.07 $\pm$ 0.07 \| 88.80 $\pm$ 0.16 |90.44 $\pm$ 0.20  \| 90.56 $\pm$ 0.36 |
> | BalancedSoftmax     | 90.72 $\pm$ 0.10 \| 90.54 $\pm$ 0.12            | 89.61 $\pm$ 0.29 \| 89.16 $\pm$ 0.29 |90.83 $\pm$ 0.12  \| 91.07 $\pm$ 0.11 |
> | GraphSMOTE          | 90.30 $\pm$ 0.31 \| 90.61 $\pm$ 0.65            | 89.53 $\pm$ 0.61 \| 88.96 $\pm$ 0.48 |90.03 $\pm$ 0.65  \| 90.76 $\pm$ 0.78 |
> | Renode              | 90.03 $\pm$ 0.21 \| 90.32 $\pm$ 0.23            | 88.51 $\pm$ 0.80 \| 88.54 $\pm$ 0.65 |90.25 $\pm$ 0.56  \| 90.14 $\pm$ 0.32 |
> | GraphENS            | 89.55 $\pm$ 0.06 \| 89.42 $\pm$ 0.06            | 89.95 $\pm$ 0.31 \| 89.58 $\pm$ 0.20 |90.14 $\pm$ 0.26  \| 90.10 $\pm$ 0.28 |
> | BalancedSoftmax+TAM | 90.97 $\pm$ 0.12 \| $\underline{90.89 \pm 0.10}$              | 89.86 $\pm$ 0.25  \| 89.36 $\pm$ 0.26 |$\underline{90.96  \pm 0.08}$  \| $\underline{91.24 \pm 0.08}$ |
> | GraphENS+TAM        | 89.53 $\pm$ 0.03 \| 89.60 $\pm$ 0.04           |$\underline{90.02 \pm 0.25}$ \|$ \underline{89.80 \pm 0.20}$|90.33 $\pm$ 0.09  \| 90.39 $\pm$ 0.17 |
> | $\textbf{UNREAL}$   | $\textbf{91.53 $\pm$ 0.43}$ \| $\textbf{91.42 $\pm$ 0.32}$ | $\textbf{90.89 $\pm$ 0.29}$ \| $\textbf{90.64 $\pm$ 0.65}$ |$\textbf{92.43 $\pm$ 0.67}$  \| $\textbf{92.76 $\pm$ 0.68}$ |
>
>
>
> We can observe that UNREAL still achieves state-of-the-art on naturally imbalanced datasets (both training and testing sets are imbalanced).

---

> ### Author Response · Authors · 2022-12-02
> **More experiments about UNREAL and other baselines on naturally imbalanced datasets**
>
> In order to verify the performance of UNREAL and all baselines on natural imbalanced datasets better, we conduct more experiments on Coauthor-CS($\rho$=35.05) and Amazon-Computers($\rho$=17.73). Since we can't update the manuscript now，we would like to present here. We construct the imbalanced training set by random sampling, and repeat the experiment five times. The label distributions of the training set, testing set and the whole graph are as follows:
>
> | Class/Node size（Coauthor-CS） | C0  | C1  | C2    |C3  | C4  | C5  | C6 | C7 | C8  | C9   | C10  | C11| C12| C13| C14 |
> |------------------------------|-----|-----|-------|----|-------|----|---- |----|-----|------|------|----|----|----|----|
> | The whole graph              | 708 | 462 | 2050  |429| 1394 |2193|371| 924| 775 | 118  | 1444 | 2033| 420| 4136| 876|
> | Training set                 | 70  | 46  | 205   | 42 | 139  | 219 | 37 | 92   | 77    |  11   | 144     | 203   | 42   | 413   | 87   |
> | Testing set                  | 608 | 386 | 1815  | 357 |1225|  1944 | 304 | 802   | 668    |  77  | 1270    | 1800   | 348    |  3693   | 759   |
>
>
> | Class/Node size（Amazon-Computers） | C0  | C1  | C2   | C3   | C4  | C5  | C6 | C7 | C8  | C9  |
> |-----------------------------------|-----|-----|------|------|-------|----|---- |----|-----|-----|
> | The whole graph                   | 436 | 2142 | 1414 | 542  | 5158 |308 |487 | 818 | 2156 | 291 |
> | Training set                      | 43  | 214  | 141  | 54   | 515  | 30 | 48 | 81  | 215    | 29  |
> | Testing set                       | 363 | 1898 | 1243 | 458  |  4613 |248 |409   | 707    | 1911  | 232|

---

### Author Response · Authors · 2022-12-02
**Common replies to all reviewers**

We would like to thank the reviewers for their efforts in reviewing our paper and their constructive suggestions on improving our manuscript. We have added more clarifications and new experiments to address reviewers' concerns in the updated manuscript. The main changes are listed below:


1. We update the experimental results on the natural unbalanced dataset Flickr in Table 8.  The latest experimental results of UNREAL based on GraphSAGE are much higher than the best results of the baselines (bacc: +23.89%, F1: +26.45%).

2. We reorganize some parts of the paper for clarity. We further clarify the motivation for different components in UNREAL in section 1. We elaborate on the pseudo-label misjudgment augmentation problem of Self-Training under imbalanced scenarios in Section 3.  We conduct comprehensive experimental studies to verifyy this problem. Due to space constraints, thel details and conclusions are deferred to Appendix B.


3. We correct some notations in Section 2.1, and summarize the notations in a table in Appendix A for convenience.


4. We provide more ablation studies on DPAM in Appendix D.1.


5. We add a more detailed definition of confidence and confidence ranking in Section 4.2.


6. We add more ablation analysis on Node-Reordering, which is presented in Appendix D.2.


7. We present a visualization to illustrate geometric imbalance (Figure 9).


8. We provide more empirical verification on the effectiveness of DGI; the results and analysis are provided in Appendix D.2.

We hope we have addressed all your concerns. If it is not the case, we will be happy to further answer any questions and improve the manuscript.

---

### Decision · Program_Chairs · 2023-01-20

**Decision:**

Reject

**Justification For Why Not Higher Score:**

Technical depth is a bit limited, even though the empirical results are encouraging.

**Justification For Why Not Lower Score:**

N/A

**Metareview: Summary, Strengths And Weaknesses:**

The paper proposes a technique for coping with label imbalance in graph-node classification. The basic idea is to perform a careful form of node oversampling, based on unlabelled nodes. Empirical results show the method works well on several benchmarks.

Reviewers generally appreciated the promising results from this method, and the motivation of coping with imbalance in node classification appears clear. There were some concerns regarding the novelty over prior work, and the potential for a more systematic treatment of some introduced ideas (e.g., the GI metric).

The response was quite detailed, and included many new results and some updates to the writeup. These are highly appreciated. The empirical results indeed seem very promising. At the same time, in our reading, we did feel that the technical content is somewhat limited: the paper involves a somewhat complicated multi-step procedure, and it is not immediate how brittle the method is in choices at each of these stages. Some more analysis, e.g., along the lines of Sec 3, could help in this regard; however in the current presentation the main body is somewhat ad-hoc in terms of techniques. We nonetheless sincerely encourage the authors to incorporate all reviewer feedback and work on a new submission for some future venue.